# Why Dedicated Critics: Eliminating Target Drift in Multi-Constraint RL

Yue Yang [1 2]   Chenghao Huang [1]   Hao Wang [1]

## Abstract

Lagrangian-based methodologies are one of the fundamental paradigms of safe reinforcement learning (RL) for constrained Markov decision processes, particularly when dealing with multi-constraint cases. While the specific details of the methodologies may differ, with some using a single estimator for the overall mixed penalty term of the constraints and others using separate estimators for the constraints, the fundamental question of the theoretical validity of the methodologies has remained largely unexplored. The present paper performs the first theoretical analysis of the methodologies and proves that the use of the mixed critic structure leads to the presence of a bias due to the target drift of the Lagrange multipliers. On the other hand, the use of the dedicated critic structure, where separate critics are used for the reward function and the constraint functions, does not suffer from this bias. The theoretical analysis is supported with experiments on a realistic power system environment with multiple constraints, where the dedicated critic structure succeeds in satisfying the constraints, whereas the mixed critic structure fails.

## 1. Introduction

Safe reinforcement learning in constrained Markov decision processes (Altman, 2021) has become prominent in real-world applications, such as robotics, power systems, autonomous vehicles, and healthcare. (Yan & Xu, 2020; Wang et al., 2020; Calascibetta et al., 2023; Zhang et al., 2020; Shi et al., 2023). Lagrangian-based approaches are among the most popular frameworks used to tackle these problems (Achiam et al., 2017; Ray et al., 2019b), where Lagrange multipliers are incorporated for the constraints and

[1]Department of Data Science and AI, Monash University, Melbourne, Australia [2]Maincode, Melbourne, Australia. Correspondence to: Yue Yang <yue.yang1@monash.edu>, Hao Wang <hao.wang2@monash.edu>.

*Proceedings of the 43rd International Conference on Machine Learning*, Seoul, South Korea. PMLR 306, 2026. Copyright 2026 by the author(s).

a mixed/augmented objective function is optimized. The advantages of this approach include the following: it converts a constrained problem to an unconstrained problem, allowing us to leverage powerful policy gradient and actor-critic approaches, and it enforces the constraints using dual variable updates, making it elegant and scalable for complex and high-dimensional systems, especially under certain assumptions, where saddle-point solutions maximize the objective and satisfy the constraints(Achiam et al., 2017).

One of the most prominent but often overlooked facts is that real-world CMDPs rarely involve a single constraint. Instead, agents are often required to meet a variety of safety, resource, and/or performance constraints during both training and deployment. For instance, a robotic system needs to avoid unsafe behaviors while satisfying torque and energy constraints. (Liu et al., 2022; Junges et al., 2016); The autonomous driving agents must take into account the safety margins, passenger comfort, and traffic regulations (Zhang et al., 2023; 2021); and power systems must balance supply and demand while maintaining safe voltage and capacity constraints (Wu et al., 2023; Chen et al., 2022). These constraints are rarely independent and often conflict, making multi-constraint settings the norm rather than the exception.

A critical but insufficiently studied aspect of Lagrangian safe RL is the value-critic architecture for multi-constraint problems. Although the CMDP formalism and constraint-aware algorithms such as CPO define per-constraint(dedicated) quantities for policy updates (Achiam et al., 2017), there remains *no theoretical justification* in the literature for why to use this approach in practice. On the other hand, most widely used implementations, including PPO-/TRPO-Lagrangian baselines (Ray et al., 2019b; Stooke et al., 2020; Yang et al., 2020; Bhatnagar et al., 2009; Kim et al., 2023), implicitly collapse all constraints into a single mixed penalty term and estimate it using one cost critic. While simple and computationally efficient, this design sidesteps the unique challenges posed by multi-constraint CMDPs. As shown in Figure 1, practitioners tackling real-world problems are left without clear criteria for choosing between approaches. However, the theoretical validation of these approaches remains unproven, and the optimization bias may induce remains underexplored. This issue is not merely theoretical. Empirical evidence shows that PPO-Lagrangian, despite its widespread use, suffers from instability and inconsistent per-

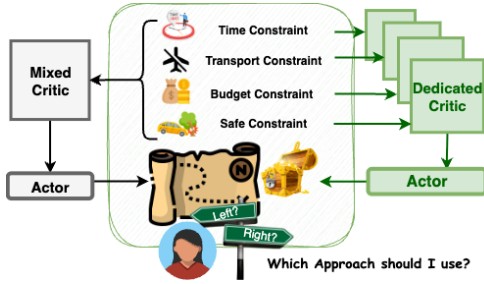

*Figure 1.* Difference between the two approaches.

formance in multi-constraint settings (Stooke et al., 2020; Tessler et al., 2019). These observations further motivate a deeper theoretical analysis of critic design and support our proposal for a dedicated-critic framework as a necessary advancement for stable and scalable safe RL under multiple constraints.

Motivated by the widespread adoption of mixed critic architectures in safe reinforcement learning (RL) and the increasing necessity for multi-constraint decision-making in realistic scenarios, we bridge the significant theoretical gap in constrained RL. To this end, we provide the first formal analysis comparing the mixed critic and dedicated critic approaches in the setting of constrained RL using the Lagrangian method. Our analysis proves that the mixed critic approach, when used in multi-constraint scenarios, results in a structural bias in the actor update. In particular, as the Lagrange multipliers change during the training process, the target of the critic changes in a manner that contradicts the stationarity condition required in temporal difference learning, leading to a steady-state error in the estimation of the policy gradient. We further show that the dedicated critic approach, where separate critics are used for the reward and the individual constraint signals, completely eliminates the dual-induced critic drift. To support the claims of the analysis, we have implemented the mixed critic and the dedicated critic approaches in the constrained bandit problem and the constrained energy control problem, where multiple constraints are involved. The simulation results show that the mixed critic approach frequently violates the constraints, while the dedicated critic approach always converges to stable learning and satisfying the constraints. The main contributions of the work are as follows: 1), we provide the first formal analysis comparing the mixed critic and the dedicated critic approaches in the setting of constrained RL using the Lagrangian method, where we show that the mixed critic approach is plagued by dual-induced bias; 2), we further show that the dedicated critic approach, where separate critics are used for the reward and the individual constraint signals, completely eliminates the dual-induced critic drift, as the target is stationary, thus removing the steady-state error in the estimation of the policy gradient; 3), we validate the claims using constrained Markov Decision Processes (MDPs) and the constrained energy control problem, where

multiple constraints are involved.

## 2. Related Work

Safe RL aims to train agents that not only maximize long-term performance but also respect safety or risk-related constraints during learning and deployment. This is often formalized through the framework of *Constrained Markov Decision Processes* (CMDPs), where the objective is to maximize expected return while ensuring that expected costs, representing safety violations or resource usage, remain below specified thresholds. (Altman, 2021; Garcıa & Fernández, 2015). This formulation admits a primal–dual view in which constraints are handled by Lagrange multipliers, giving rise to the widely used *Lagrangian (lag-based)* methods: they update policy parameters to ascend a Lagrangian objective and update dual variables toward feasibility. Prominent examples include TRPO-Lagrangian and PPO-Lagrangian (Achiam et al., 2017; Ray et al., 2019b), SAC-Lagrangian variants (Ray et al., 2019b), and Reward-Constrained Policy Optimization (RCPO) (Tessler et al., 2019). Compared to alternative approaches(Stooke et al., 2020; Liu et al., 2020; Xu et al., 2021; Chow et al., 2018), Lagrangian methods offer several practical advantages that have led to their widespread adoption (Achiam et al., 2017; Schulman et al., 2015; Ray et al., 2019b; Yang et al., 2021; Kim et al., 2023). They are *plug-and-play* compatible with both on-policy and off-policy learners, and introduce only a small number of hyperparameter. These properties make them highly amenable to integration within standard RL pipelines. Consequently, Lagrangian variants like PPO-Lag and TRPO-Lag have become *de facto baselines* in major Safe RL benchmarks and toolkits. Their accessibility, combined with consistently strong empirical performance, has made them the dominant choice in both robotics and simulated safety-critical control environments.

A critical yet underexplored aspect of Lagrangian based safe reinforcement learning is the architecture of value critics when dealing with multiple constraints. The question of how to estimate constraints returns in deep RL has received far less attention than objective or dual design, but it underlies much of the instability noted in safe RL. In early safe RL and constrained MDP work, classic algorithms (e.g., Constrained Policy Iteration) implicitly worked with per-constraint value functions, but without deeply discussing representation in function approximation settings (Altman, 2021; Achiam et al., 2017). As deep safe RL matured, many practical baselines resorted to collapsing cost signals into an aggregated penalty and training a mixed "cost critic" alongside a reward critic; this pattern is pervasive in benchmark codebases (e.g. Safety Starter Agents, PPO-/TRPO-Lagrangian) (Ray et al., 2019b; Stooke et al., 2020). Some recent methods extend to multiple constraints,

but often leave the critic architecture unspecified or adopt ad-hoc shared representations rather than formally treating per-constraint estimation (Kim et al., 2023). Work on stabilizing Lagrangian dual updates, such as PID-Lagrangian, dual clipping, or adaptive multiplier heuristics, addresses the dual dynamics but typically retains the standard two-critic collapse architecture (Stooke et al., 2020; Xu et al., 2021; Liu et al., 2020). In off-policy safe RL methods like SAC-Lagrangian variants or worst-case safety critics (e.g., WCSAC), the separation between reward and cost critics is common, but again usually implemented at the aggregate cost level even when multiple constraints are present (Tessler et al., 2019). Across the literature, the critic architecture, whether to collapse or separate constraints, is treated as an afterthought, often chosen for ease or efficiency rather than guided by theoretical insight. This pervasive gap means that many empirical instability observations, violation spikes, slow convergence remain underexplained-pointing to a need for more rigorous analysis of critic structure in multi-constraint safe RL. In this work, we fill this gap by theoretically and empirically analyzing these design choices. Our analysis shows that mixed constraint critics can introduce structural bias in multi-constraint settings, whereas dedicated critics mitigate this bias by isolating constraint signals.

## 3. Problem Formulation

In this section, we begin by formalising the multi-constraint constrained reinforcement learning problem. A Constrained Markov Decision Process (CMDP)(Altman, 2021) is specified by the tuple $(\mathcal{S}, \mathcal{A}, P, \gamma, r, \{c_i\}_{i=1}^m)$, where: $\mathcal{S}$ is the (possibly infinite) state space; $\mathcal{A}$ is the action space; $P(\cdot|s, a)$ is the transition matrix; $\gamma \in (0, 1)$ is the discount factor; $r : \mathcal{S} \times \mathcal{A} \to \mathbb{R}$ is the reward signal we aim to maximize; $c_i : \mathcal{S} \times \mathcal{A} \to \mathbb{R}_+$ are the $i^{th}$ cost signals corresponding to a safety or resource constraint. For a stochastic policy $\pi_\theta(a|s)$ parameterized by $\theta$, the expected discounted return of a signal $x \in \{r, c_1, \ldots, c_m\}$ is:

$$J_x(\pi_\theta) = \mathbb{E}_{\pi_\theta}\left[\sum_{t=0}^\infty \gamma^t x(s_t, a_t)\right]. \quad (1)$$

In particular, $J_r(\pi_\theta)$ is the expected reward return and $J_{c_i}(\pi_\theta)$ is the expected discounted cost associated with constraint $i$.

Constraints, in the form of $J_{c_i}(\pi_\theta) \leq d_i$ can be enforced via a Lagrangian formulation (Achiam et al., 2017). By introducing multipliers $\lambda = (\lambda_1, \ldots, \lambda_m) \in \mathbb{R}_+^m$, we have $\mathcal{L}(\theta, \lambda) = J_r(\pi_\theta) - \sum_{i=1}^m \lambda_i (J_{c_i}(\pi_\theta) - d_i)$. We could proceed the optimization process in a *primal–dual* fashion: the actor seeks to maximize $\mathcal{L}(\theta, \lambda)$ over $\theta$, while the dual variables $\lambda$ adaptively adjust to enforce the constraints. Here, we use $\mathcal{E}_{\pi_\theta}^x(\cdot; \omega)$ to denote a learned *signal estimator*

for $x \in \{r, c_1, \ldots, c_m\}$ parameterized by $\omega$. It is worth mention that, the input of this estimator may be $s$ or $(s, a)$ depending on the method (e.g., value-, advantage-, or return-based).

By the policy gradient theorem, we could write the gradient of the Lagrangian w.r.t. $\theta$ as:

$$\nabla_\theta \mathcal{L}(\theta, \lambda) = \mathbb{E}_{s,a\sim\pi_\theta}\Big[\nabla_\theta \log \pi_\theta(a|s) \left(\mathcal{E}_{\pi_\theta}^r(s, a)\right.$$
$$\left. - \sum_{i=1}^m \lambda_i \mathcal{E}_{\pi_\theta}^{c_i}(s, a)\right)\Big], \quad (2)$$

where $\mathcal{E}_{\pi_\theta}^r$ and $\mathcal{E}_{\pi_\theta}^{c_i}$ denote the learned signal estimators for reward and each cost under $\pi_\theta$. Note that $\lambda$ merely scales the $\mathcal{E}_{\pi_\theta}^{c_i}$ contributions and the estimators themselves depend only on $\pi_\theta$.

Now we define two critic architectures: the dedicated critic and the mixed critic as following:

**Mixed-critic (Classic) methods.** Constraints are *aggregated* before (or within) estimation, so there is no per-constraint head. Two common variants both qualify as "mixed critic":

*(a) Single-Estimator:* $\mathcal{E}_{\pi_\theta}^{\text{mix}}(\cdot; \omega) \approx \mathcal{E}_{\pi_\theta}^{r - \sum_{i=1}^m \lambda_i c_i}(\cdot)$.

*(b) Two-Estimator:*

$$\mathcal{E}_{\pi_\theta}^r(\cdot; \omega^r), \mathcal{E}_{\pi_\theta}^{\text{cost-agg}}(\cdot; \omega^c) \approx \mathcal{E}_{\pi_\theta}^{\sum_{i=1}^m \lambda_i c_i}(\cdot).$$

In both (a) (Altman, 2021) and (b) (Ray et al., 2019b; Stooke et al., 2020), all constraints are *mixed into a single scalar cost signal*, hence "mixed critic."

**Dedicated-critic methods.** Maintain one estimator *per signal*, i.e., a *Separate Estimator* for reward and for *each* constraint (Achiam et al., 2017): $\{\mathcal{E}_{\pi_\theta}^x(\cdot; \omega^x) : x \in \{r, c_1, \ldots, c_m\}\}$.

## 4. Theoretical Analysis: Why we need Dedicated Critics for Multi-Constraint Problems

In this section, we systematically show that training a mixed critic on multiple signals generally produces actor updates that fail to track the true Lagrangian gradient $\nabla_\theta \mathcal{L}(\theta, \lambda)$ during learning, unless one imposes stronger timescale separation (critic faster than *both* actor and dual) and near-exact critics. In contrast, a *dedicated-critic* design (one critic per signal) does not suffer from this issue.

**Assumption 4.1** (Stepsizes and timescale separation). Critic, actor, and dual stepsizes $\eta_t, \alpha_t, \beta_t > 0$ satisfy $\sum_{t=0}^\infty \eta_t = \infty$, $\sum_{t=0}^\infty \eta_t^2 < \infty$. $\frac{\alpha_t}{\eta_t} \to 0$, $\frac{\beta_t}{\eta_t} \to 0$.

**Assumption 4.2** (Bounded policy score and compact dual domain). At iteration $t$, if $(s_t, a_t)$ are sampled on-policy

under $\pi_{\theta_t}$, there exists $G < \infty$ such that the log-policy score is uniformly bounded almost surely: $\|\nabla_\theta \log \pi_{\theta_t}(a_t \mid s_t)\| \leq G$. And the dual variable sequence $\{\lambda_t\}_{t \geq 0} \subset \mathbb{R}_{\geq 0}^m$ remains in a fixed compact set $\Lambda \subset \mathbb{R}_{\geq 0}^m$.

**Assumption 4.3** (Critic noise regularity). Let the critic update use the population linear form with additive martingale-difference noise: $\omega_{t+1} = \omega_t + \eta_t(b_t - A_t\omega_t) + \eta_t \zeta_{t+1}$, where $A_t := A(\theta_t)$ and $b_t := b(\theta_t)$ are $\mathcal{F}_t$-measurable. The noise $\{\zeta_{t+1}\}_{t \geq 0}$ satisfies $\mathbb{E}[\zeta_{t+1} \mid \mathcal{F}_t] = 0$ and $\mathbb{E}[\|\zeta_{t+1}\|^2 \mid \mathcal{F}_t] \leq \sigma^2 < \infty$ a.s. for all $t$.

Here, our Assumption 4.1 follows common stochastic approximation practice and shrinking stepsizes and converges faster than the actor and dual. Assumption 4.2 is common for typical policies, such as softmax, Gaussian so for models with clipped parameters, and for methods based on lags, projecting/clipping $\lambda_t$ onto a box $\Lambda$. Assumption 4.3 is satisfied in a standard on-policy scenario because, as is common in deep RL analysis, having a bounded feature vector and using mini-batch estimates of gradients yields a martingale difference noise with a uniformly bounded conditional variance. All of these conditions are mild and sufficient for analytical work. the dual-induced drift term that drives the need for per-signal critics.

For a given policy $\pi_\theta$ and a given signal $x \in \{r, c_1, \ldots, c_m\}$, the critic for each signal within a linear class (Sutton, 1988; Tsitsiklis & Van Roy, 1996a) sfollows the projected Bellman equation (PBE) (Munos, 2003; Tsitsiklis & Van Roy, 1996b), which gives us the normal equations $A(\theta)\,\omega^{x,*}(\theta) = b^x(\theta)$ with $A(\theta) = \Phi^\top D_\theta(I - \gamma P_{\pi_\theta})\Phi$ and $b^x(\theta) = \Phi^\top D_\theta r^x$ (state–action and advantage/GAE variants give the same linear template for the state-action and advantage functions, respectively, with the correct $A, b$). We work assume that there exist $A(\theta) \succeq \mu I$ for some $\mu > 0$, and $A(\cdot), b^x(\cdot)$ are locally Lipschitz in $\theta$ (See Appendix C and G for details).

### 4.1. Mixed-Critic in Multi-Constraint CMDPs

When a mixed critic is used in the scalarized signal, $r_{\lambda_t} = r - \sum_{i=1}^m \lambda_{t,i} c_i$, the PBE is given by $A(\theta_t)\,\omega_t^{\mathrm{mix},*} = b_t^{\mathrm{mix}} := b^r(\theta_t) - \sum_{i=1}^m \lambda_{t,i} b^{c_i}(\theta_t)$. The stochastic update follows a Robbins-Monro update towards this fixed point through mini-batch estimates of $A(\theta_t)$ and $b_t^{\mathrm{mix}}$. The update in population form, along with an error term with mean zero, is written as follows:

$$\omega_{t+1}^{\mathrm{mix}} = \omega_t^{\mathrm{mix}} + \eta_t(b_t^{\mathrm{mix}} - A(\theta_t)\,\omega_t^{\mathrm{mix}}) + \eta_t \zeta_{t+1}, \quad (3)$$

where $\eta_t > 0$ is the critic stepsize and $\zeta_{t+1}$ is a martingale-difference noise, which is used to capture finite-sample and sampling variability. For a given parameter set $(\theta, \lambda)$, the Nash equilibrium for the mixed signal problem is given by $A(\theta)\,\omega^{\mathrm{mix}}(\theta, \lambda) = b^r(\theta) - \sum_{i=1}^m \lambda_i b^{c_i}(\theta)$. By linearity of the operator, we can decompose the solution as

$\omega^{\mathrm{mix}}(\theta, \lambda) = \omega^r(\theta) - \sum_{i=1}^m \lambda_i\,\omega^{c_i}(\theta)$, where $\omega^r(\theta)$ and $\omega^{c_i}(\theta)$ are the solutions for the PBE corresponding to the reward signal and each cost signal, respectively. The target at time t is $\omega_t^{\mathrm{mix},\star} = \omega^{\mathrm{mix}}(\theta_t, \lambda_t), e_t = \omega_t^{\mathrm{mix}} - \omega_t^{\mathrm{mix},\star}$. That is, $e_t$ is the critic error with respect to the exact PBE solution for the current pair$(\theta_t, \lambda_t)$. Subtracting $\omega_{t+1}^{\mathrm{mix},\star}$ from both sides of the recursion equation 3 yields the exact error recursion

$$\begin{aligned} e_{t+1} = &(I - \eta_t A(\theta_t))\,e_t + \\ &\underbrace{(\omega_t^{\mathrm{mix},\star} - \omega_{t+1}^{\mathrm{mix},\star})}_{\text{target drift}} + \eta_t \zeta_{t+1} + \Delta_t^\theta, \end{aligned} \quad (4)$$

where $\Delta_t^\theta$ denote the small perturbations of the matrix $A(\theta)$; $I \in \mathbb{R}^{d \times d}$ denote the $d \times d$ identity matrix, and $b(\theta)$ induced by $\theta_{t+1} \neq \theta_t$. Using the stationary equivalence $\omega_t^{\mathrm{mix},\star} = \omega^r(\theta_t) - \sum_{i=1}^m \lambda_{t,i}\,\omega^{c_i}(\theta_t)$ and the analogous expression at time $t+1$, we have

$$\omega_t^{\mathrm{mix},\star} - \omega_{t+1}^{\mathrm{mix},\star} = \omega^r(\theta_t) - \sum_{i=1}^m \lambda_{t,i}\,\omega^{c_i}(\theta_t) -$$

$$\left[\omega^r(\theta_{t+1}) - \sum_{i=1}^m \lambda_{t+1,i}\,\omega^{c_i}(\theta_{t+1})\right] = \omega^r(\theta_t) - \omega^r(\theta_{t+1}) -$$

$$\sum_{i=1}^m \left(\lambda_{t,i}\,\omega^{c_i}(\theta_t) - \lambda_{t+1,i}\,\omega^{c_i}(\theta_{t+1})\right). \quad (5)$$

Add and subtract $\lambda_{t,i}\,\omega^{c_i}(\theta_{t+1})$ inside the sum:

$$\begin{aligned} &\lambda_{t,i}\,\omega^{c_i}(\theta_t) - \lambda_{t+1,i}\,\omega^{c_i}(\theta_{t+1}) = \\ &\lambda_{t,i}(\omega^{c_i}(\theta_t) - \omega^{c_i}(\theta_{t+1})) + (\lambda_{t,i} - \lambda_{t+1,i})\omega^{c_i}(\theta_{t+1}). \end{aligned} \quad (6)$$

Substituting equation 6 into equation 5 yields

$$\omega_t^{\mathrm{mix},\star} - \omega_{t+1}^{\mathrm{mix},\star} =$$

$$\left(\omega^r(\theta_t) - \omega^r(\theta_{t+1})\right) - \sum_{i=1}^m \lambda_{t,i}\left(\omega^{c_i}(\theta_t) - \omega^{c_i}(\theta_{t+1})\right)$$

$$- \sum_{i=1}^m \left(\lambda_{t+1,i} - \lambda_{t,i}\right)\omega^{c_i}(\theta_{t+1}). \quad (7)$$

By Lipschitz continuity, there exist $L_r, L_{c_i} < \infty$ such that $\|\omega^r(\theta_{t+1}) - \omega^r(\theta_t)\| \leq L_r \|\theta_{t+1} - \theta_t\|, \|\omega^{c_i}(\theta_{t+1}) - \omega^{c_i}(\theta_t)\| \leq L_{c_i} \|\theta_{t+1} - \theta_t\|$. Using the standard update rule for an actor $\theta_{t+1} = \theta_t + \alpha_t g_t^{\mathrm{act}}$, where $g_t^{\mathrm{act}}$ is a stochastic policy-gradient estimate and $\|g_t^{\mathrm{act}}\| \leq C_\theta$, we have $\|\theta_{t+1} - \theta_t\| = \alpha_t \|g_t^{\mathrm{act}}\| = O(\alpha_t)$. Therefore,

$$\|\omega^r(\theta_t) - \omega^r(\theta_{t+1})\| = O(\alpha_t),$$

$$\|\lambda_{t,i}(\omega^{c_i}(\theta_t) - \omega^{c_i}(\theta_{t+1}))\| \leq \|\lambda_t\|_\infty$$

$$L_{c_i}\|\theta_{t+1} - \theta_t\| = O(\alpha_t),$$

using $\lambda_t \in \Lambda$ compact. Thus the first two terms in equation 7 are $O(\alpha_t)$.

Consider the dual-induced part of equation 7: $\sum_{i=1}^{m} (\lambda_{t+1,i} - \lambda_{t,i}) \omega^{c_i}(\theta_{t+1})$. By Assumption 4.2, $\lambda_t \in \Lambda$ with $\Lambda$ compact, and local Lipschitzness, the map $\theta \mapsto \omega^{c_i}(\theta)$ is continuous; hence $M := \sup_{i,\theta} \|\omega^{c_i}(\theta)\| < \infty$ Therefore

$$\left\| \sum_{i=1}^{m} (\lambda_{t+1,i} - \lambda_{t,i}) \omega^{c_i}(\theta_{t+1}) \right\|$$
$$\leq \|\lambda_{t+1} - \lambda_t\|_1 \max_{i} \|\omega^{c_i}(\theta_{t+1})\|$$
$$\leq M \|\lambda_{t+1} - \lambda_t\|.$$

In a standard projected dual update, we have $\lambda_{t+1} = \Pi_\Lambda(\lambda_t + \beta_t g_t)$, with $\Pi_\Lambda$ nonexpansive and $\|g_t\| \leq C_\lambda$ ($g_t \in \mathbb{R}^m$ denotes a subgradient of the dual objective with respect to $\lambda$). Therefore, we get $\|\lambda_{t+1} - \lambda_t\| \leq \|\lambda_t + \beta_t g_t - \lambda_t\| = \beta_t \|g_t\| \leq \beta_t C_\lambda = O(\beta_t)$.

Combining the two displays yields $\left\| \sum_{i=1}^{m} (\lambda_{t+1,i} - \lambda_{t,i}) \omega^{c_i}(\theta_{t+1}) \right\| \leq MC_\lambda \beta_t = O(\beta_t)$. The target drift can also be written as $\omega_t^\star - \omega_{t+1}^\star = O(\alpha_t) - O(\beta_t)$. The $O(\alpha_t)$ term comes from policy updates, a property common to all actor-critic methods, and an extra term $O(\beta_t)$ is added for additional complexity. In this setup, the target not only changes as a result of policy parameter updates $\theta$ (the standard $O(\alpha_t)$ policy-driven drift), but also as a result of updates to the dual variables $\lambda$, resulting in an additional $O(\beta_t)$ that causes bias to the actor gradient due to this additional dual-driven term.

*Remark* 4.4 (Drift for the reward critic + cost critic design). In the two-critic setup, one critic estimates the reward target $\omega^r(\theta)$ and a second critic estimates the aggregated cost target $\omega^{\text{cost}}(\theta, \lambda) = \sum_{i=1}^{m} \lambda_i \omega^{c_i}(\theta)$, (used in PPO-LAG, TRPO-LAG) follow the same update process on $\omega^{\text{cost}}$, we have

$$\omega_t^{\text{cost},\star} - \omega_{t+1}^{\text{cost},\star} = -\sum_{i=1}^{m} \lambda_{t,i} \left( \omega^{c_i}(\theta_t) - \omega^{c_i}(\theta_{t+1}) \right) - $$
$$\sum_{i=1}^{m} (\lambda_{t+1,i} - \lambda_{t,i}) \omega^{c_i}(\theta_{t+1}). \quad (8)$$

In this case, the reward part $\omega^r(\theta_t) - \omega^r(\theta_{t+1})$ does not appear because it is handled by the *reward critic's* own recursion; equation 8 isolates the drift of the *aggregated-cost* head, which contains a policy-induced component (via $\theta_{t+1} - \theta_t$) and a dual-induced component (via $\lambda_{t+1} - \lambda_t$).

**Lemma 4.5** (Mixed-critic error bound). *If $A(\theta) \succeq \mu I$ uniformly and the stepsizes satisfy $\alpha_t/\eta_t \to 0$, $\beta_t/\eta_t \to 0$, then there exist constants $C_\lambda, C_\theta$, such that:*

$$\limsup_{t\to\infty} \mathbb{E}\|e_t\| \leq$$
$$\frac{C_\lambda}{\mu} \limsup_{t\to\infty} \frac{\beta_t}{\eta_t} + \frac{C_\theta}{\mu} \limsup_{t\to\infty} \frac{\alpha_t}{\eta_t} + O(1). \quad (9)$$

*Proof sketch.* From equation 4 and since $A(\theta_t) \succeq \mu I$, $\|(I - \eta_t A(\theta_t))e_t\| \leq (1 - \mu\eta_t)\|e_t\|$, using equation 7, we have: $\|\omega_t^\star - \omega_{t+1}^\star\| \leq C_\lambda \|\lambda_{t+1} - \lambda_t\| + C_\theta \|\theta_{t+1} - \theta_t\| \leq C_\lambda \beta_t + C_\theta \alpha_t$. Taking expectations and using bounded martingale difference sequence noise and apply the standard stochastic approximation comparison: if $x_{t+1} \leq (1 - a_t)x_t + b_t$ with $a_t = \mu\eta_t$, then $\limsup x_t \leq \limsup b_t/a_t$. Hence, $\limsup_{t\to\infty} \mathbb{E}\|e_t\| \leq \frac{1}{\mu} \limsup_{t\to\infty} \left( \frac{C_\lambda \beta_t}{\eta_t} + \frac{C_\theta \alpha_t}{\eta_t} + O(\eta_t) \right)$, which yields equation 9 since $\eta_t \to 0$. Please refer to Appendix D for detailed proof. □

**Lemma 4.6** (Actor-gradient bias bound). *Let $\widehat{g}_t$ and $g_t^\star$ be the actor's estimated and ideal gradients,*

$$\widehat{g}_t = \mathbb{E}_t[\nabla_\theta \log \pi_{\theta_t}(a_t|s_t) \phi(s_t, a_t)^\top \omega_t], \quad (10a)$$
$$g_t^\star = \mathbb{E}_t[\nabla_\theta \log \pi_{\theta_t}(a_t|s_t) \phi(s_t, a_t)^\top \omega_t^\star], \quad (10b)$$

*with critic error $e_t = \omega_t - \omega_t^\star$. Assume the score and features are bounded as $\|\nabla_\theta \log \pi_{\theta_t}(a_t|s_t)\| \leq G, \|\phi(s_t, a_t)\| \leq L_\phi$. Then the actor-gradient bias $B_t := \widehat{g}_t - g_t^\star$ satisfies*

$$\|B_t\| \leq GL_\phi \|e_t\|. \quad (11)$$

*Proof.* See Appendix E for detailed proof. □

**Theorem 4.7** (Bias from a Mixed Critic). *Suppose Assumptions 4.1–4.3 hold and $A(\theta) \succeq \mu I$ uniformly in $\theta$. Then the actor-gradient bias $B_t$ incurred by using a single mixed critic satisfies*

$$\limsup_{t\to\infty} \mathbb{E}\|B_t\| \leq$$
$$GL_\phi \left( \frac{C_\lambda}{\mu} \limsup_{t\to\infty} \frac{\beta_t}{\eta_t} + \frac{C_\theta}{\mu} \limsup_{t\to\infty} \frac{\alpha_t}{\eta_t} \right). \quad (12)$$

*Proof.* See Appendix F for detailed proof. □

Mixed-critic design introduces an **additional bias term** of order $\beta_t/\eta_t$, arising from the dependence of the mixed constraints on the dual variables $\lambda$. Consequently, the actor's update does not follow the true Lagrangian gradient unless the critic runs much faster than the dual ($\beta_t/\eta_t \to 0$ sufficiently quickly), or is essentially exact.

## 4.2. Dedicated-Critic in Multi-Constraint CMDPs

For dedicated-critic design, we maintain a separate critic with parameters $\omega_t^x$ for each reward and constraints $x \in \{r, c_1, \ldots, c_m\}$, updated by

$$\omega_{t+1}^x = \omega_t^x + \eta_t \left( -A(\theta_t) \omega_t^x + b^x(\theta_t) + \zeta_{t+1}^x \right), \quad (13)$$
$$x \in \{r, c_1, \ldots, c_m\}.$$

Define the signal-specific fixed point $\omega^{x,\star}(\theta)$ by $A(\theta)\,\omega^{x,\star}(\theta) = b^x(\theta)$ and let the tracking error be $e_t^x := \omega_t^x - \omega_t^{x,\star}$. Subtract $\omega_{t+1}^{x,\star}(\theta_{t+1})$ to equation 13 to obtain

$$e_{t+1}^x = \omega_{t+1}^x - \omega_{t+1}^{x,\star}(\theta_{t+1})$$
$$= \left(\omega_t^x - \eta_t A(\theta_t)\omega_t^x + \eta_t b^x(\theta_t) + \eta_t \zeta_{t+1}^x\right) - \omega_{t+1}^{x,\star}(\theta_{t+1})$$
$$= \left(I - \eta_t A(\theta_t)\right)e_t^x + \eta_t \zeta_{t+1}^x + \Delta_t^{\theta,x}. \tag{14}$$

A dedicated critic answers a question: under the *current policy*, what is the expected cumulative value of one signal– either reward or a single cost? As this question does not mention the penalty weights, changing $\lambda$ does not change what the critic is trying to predict; only changing the policy does. More specifically, each dedicated critic estimates a *signal-specific* fixed point $\omega^{x,\star}(\theta)$ defined by $A(\theta)\,\omega^{x,\star}(\theta) = b^x(\theta)$, where both $A(\theta)$ and $b^x(\theta)$ depend on $\theta$ and the single signal $x \in \{r, c_1, \ldots, c_m\}$, but not on the Lagrange multipliers; Therefore, in the one-step error recursion the only drift term comes from policy movement $\theta_t \to \theta_{t+1}$, and no $(\lambda_{t+1} - \lambda_t)$ term appears. By contrast, a mixed-critic's target is defined using $\lambda$ (it blends reward and costs with those weights), so every time $\lambda$ is updated, the target itself shifts, creating the extra $(\lambda_{t+1} - \lambda_t)$ drift.

**Lemma 4.8** (Dedicated-critic tracking error). *Suppose assumptions 4.1 and 4.3 hold, then there exists $\widetilde{C}_\theta < \infty$ such that for every $x \in \{r, c_1, \ldots, c_m\}$,*

$$\limsup_{t\to\infty} \mathbb{E}\|e_t^x\| \leq \frac{\widetilde{C}_\theta}{\mu}\,\limsup_{t\to\infty}\frac{\alpha_t}{\eta_t}. \tag{15}$$

*Proof.* Please refer to Appendix D for detailed proof. $\square$

**Theorem 4.9** (Dedicated-critic Bias). *Suppose Assumptions 4.1–4.3 hold, let $\widehat{g}_t$ and $g_t^\star$ be the actor's estimated and ideal gradients, the dedicated-critic actor bias be $B_t^{multi} := \widehat{g}_t^{multi} - g_t^\star$. Then*

$$\limsup_{t\to\infty} \mathbb{E}\|B_t^{multi}\| \leq GL_\phi \frac{\widetilde{C}_\theta}{\mu}\,\limsup_{t\to\infty}\frac{\alpha_t}{\eta_t}, \tag{16}$$

*Proof.* See Appendix F for detailed proof. $\square$

# 5. Experiments

## 5.1. Simple CMDP Bandit Problem

We design a minimal yet diagnostic constrained bandit CMDP that cleanly isolates the effect of *single* vs. *dedicated-critic* architectures under Lagrangian updates. All implementation choices below are fixed and reported for full reproducibility. We use a one-state bandit with binary actions $\mathcal{A} = \{a_1, a_2\}$, discount $\gamma = 0$, reward $r(a_1) = 0, r(a_2) =$

1, and two costs $c_1(a_1) = 0$, $c_1(a_2) = 1, c_2(a_1) = 1$, $c_2(a_2) = 0$, with constraints $J_{c_1} \leq d_1$ and $J_{c_2} \leq d_2$, where $d_1 = d_2 = 0.5$. The policy $\pi_\theta$ is a Bernoulli with a single logit $\theta \in \mathbb{R}$: $\pi_\theta(a_1) = \sigma(\theta)$, $\pi_\theta(a_2) = 1 - \sigma(\theta)$, $\sigma(\theta) = \frac{1}{1+e^{-\theta}}$. Under this policy, expected costs are $J_{c_1} = 1 - \sigma(\theta)$, $J_{c_2} = \sigma(\theta)$, and expected reward is $J_r = \sigma(\theta)\,r(a_1) + (1 - \sigma(\theta))\,r(a_2) = 1 - \sigma(\theta)$.

We compare two actor–critic variants that share the same actor and dual updates. On each step, we sample $a \sim \pi_\theta$ and apply an update with a (learned) advantage surrogate from the critic(s): $\theta_{t+1} = \theta_t + \alpha\,\widehat{g}_t$, $\widehat{g}_t := \nabla_\theta \log \pi_{\theta_t}(a_t)\,\widehat{Q}_t(a_t)$, where $\nabla_\theta \log \pi_\theta(a_1) = 1 - \sigma(\theta)$ and $\nabla_\theta \log \pi_\theta(a_2) = -\sigma(\theta)$. We maintain $\lambda = (\lambda_1, \lambda_2) \in \mathbb{R}_+^2$ with projected stochastic ascent: $\lambda_{i,t+1} = \Pi_{[0,\lambda_{\max}]}\left(\lambda_{i,t} + \beta\,(\widehat{c}_i - d_i)\right)$, $i \in \{1,2\}$, where $\widehat{c}_i$ is the instantaneous cost sample (0 or 1 in this bandit) and $\lambda_{\max} = 10$. Projection keeps $\lambda_t$ in a compact set. In this bandit, the true Lagrangian gradient has a closed form. Let $\pi_1 = \sigma(\theta)$ and $f(a) := r(a) - \lambda_1 c_1(a) - \lambda_2 c_2(a)$, $f(a_1) = r(a_1) - \lambda_2$, $f(a_2) = r(a_2) - \lambda_1$. Then $g_t := \nabla_\theta \mathcal{L}(\theta_t, \lambda_t) = \pi_1(1 - \pi_1)\,\big(f(a_1) - f(a_2)\big) = \sigma(\theta_t)\big(1 - \sigma(\theta_t)\big)\big((r(a_1) - \lambda_{2,t}) - (r(a_2) - \lambda_{1,t})\big)$.

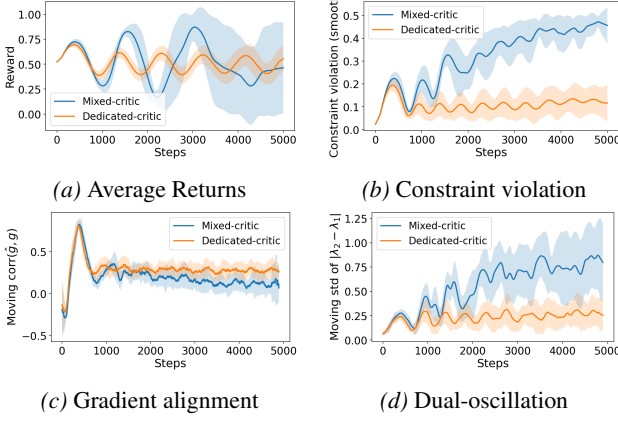

*(a) Average Returns*  *(b) Constraint violation*

*(c) Gradient alignment*  *(d) Dual-oscillation*

*Figure 2.* Performance for CMDP Bandit.

**Results and Discussion:** Both methods attain similar average returns, but the mixed-critic curve has much higher variance (large confidence band, occasional dips). The dedicated-critic maintains comparable reward with markedly lower variability. The mixed-critic exhibits large and growing volatility, whereas the dedicated-critic remains low and stable. This matches the theory: a mixed critic's target moves with $\Delta\lambda_t$, inducing oscillatory dual dynamics; dedicated critics avoid this $\lambda$–coupling.

To measure safety, we compute the violation which quantifies by how much the learned policy exceeds constraint thresholds at each step. Mixed-critic drifts to higher violations, while dedicated-critic settles much lower, implying better safety during training rather than only at convergence. We also measure how well the actor's update

direction matches the true Lagrangian gradient by computing a moving Pearson correlation between the estimated gradient $\widehat{g}_t$ and the true gradient $g_t$. Based on the results, dedicated-critic sustains a higher correlation than mixed-critic, indicating the actor follows the true Lagrangian gradient more reliably when critics are per-signal. We quantify the stability of the dual variables by tracking the moving standard deviation of the gap $|\lambda_{2,t} - \lambda_{1,t}|$. This measures whether Lagrange multipliers converge smoothly or oscillate over time. The mixed-critic exhibits large and growing volatility, whereas the dedicated-critic remains low and stable. This matches the theory: a mixed critic's target moves with $\Delta\lambda_t$, inducing oscillatory dual dynamics; dedicated critics avoid this $\lambda$–coupling.

### 5.2. Multi-constraint Power System Application

Rather than relying on standard safe-RL benchmarks (whose constraints are few and stylized), we evaluate in a *complex energy scenario* designed to stress realism and constraint diversity. Compared with standard, the environment couples stochastic demand, renewable generation uncertainty, ramping limits, transmission congestion, reserve requirements, and device-level safety, yielding *multiple, interacting* constraints with heterogeneous timescales. This setting captures (i) tight operational envelopes,(ii) correlated risks across assets, and (iii) nontrivial trade-offs between cost and safety. In this case, we adopt a standard deep PPO configuration with neural-network critics (two-layer MLPs with 256 units per layer), so the empirical results directly evaluate our approach in the deep RL regime rather than in the idealised linear setting used for the theory.

**System Overview and Constraints.** We consider a radial distribution network with high rooftop PV penetration, where community battery energy storage systems (CBESSs) are coordinated to ensure safe and efficient operation (observation dimension: 105, action dimension: 24). Each CBESS is subject to power, efficiency, and state-of-charge (SoC) constraints, can transact with the upstream grid under trading limits, and incurs both trading and degradation costs. When storage is saturated, PV curtailment is applied with fairness constraints to avoid disproportionate restrictions across buses. The system is modeled using the LinDist-Flow approximation. The central control task is to schedule CBESS actions and PV curtailment to minimize trading cost while maintaining constraint satisfaction across multiple operational and fairness dimensions. To enforce safety and equity, we define five cost terms (constraints) monitored over the scheduling horizon: (1) *Voltage Violation Ratio* penalizes the number of buses breaching voltage limits; (2) *Voltage Deviation Degree* penalizes the severity of such violations; (3) *Line Loading Cost* penalizes thermal overloads on network branches; (4) *Battery Degradation Cost* discourages excessive CBESS cycling; and (5) *PV Curtail-*

*ment Unfairness* penalizes uneven curtailment across buses. These constraints interact over heterogeneous timescales, capturing the multifaceted trade-offs in real-world power systems. Full modeling details are provided in Appendix K.

**Results and Discussion:** In this work, we prioritize *constraint satisfaction* as the central performance objective. Under consistent PPO backbones and training configurations, we compare two architectures: (i) the widely used PPO-Lagrangian baseline, which utilizes a single reward critic and a mixed critic for the aggregated cost signal, and (ii) the proposed Dedicated critic setup, which retains a shared reward critic but replaces the single cost critic with multiple per-constraint critics. Experimental details are provided in Table 3 in Appendix K. As shown in Fig. 3, the dedicated architecture exhibits significantly more stable training behavior, with smoother learning curves and reduced variance in value estimates, while the baseline often suffers from unstable updates and erratic dual dynamics, particularly when constraint signals conflict. On unseen demand and renewable profiles, the Dedicated model consistently achieves the lowest violation rates and magnitudes across all five constraint dimensions, with noticeably fewer and shorter spikes in unsafe behavior. To quantify trade-offs between return and safety, we construct empirical Pareto fronts using $\varepsilon$-constraint sweeps. As shown in Fig. 4, dedicated policies cluster tightly near the estimated frontiers, consistently outperforming the baseline across a broad range of safety budgets, achieving either lower constraint violations for the same reward, or higher reward at equivalent violation levels. These results collectively shows that performance gains of the dedicated setup are consistent with the theoretical mechanism identified in our analysis: by estimating each constraint with its own critic, the actor update depends on per-constraint advantages that are independent of the evolving multipliers, thereby avoiding the $\lambda$-driven target drift that can destabilise training. Per-constraint critics preserve the relative scale and variance of individual constraint signals, enabling head-wise normalisation and reducing "winner-takes-all" effects where the most active constraint dominates updates. This appears crucial for tracing clean Pareto sets: Dedicated policies concentrate near the frontier across budgets, whereas the aggregated-cost baseline often lies inside the frontier, indicative of optimisation bias introduced by collapsing constraints.

### 5.3. Additional Benchmarks and Timescale Ablation

We extend our experiments to a more complex electric vehicle charging station (EVCS) problem to further evaluate how well the dedicated-critic approach scales and whether its benefits persist in realistic multi-constraint settings (Observation dimension: 219; action dimension: 40). This environment models coordination problems, where the goal is to minimize charging costs while enforcing multiple charg-

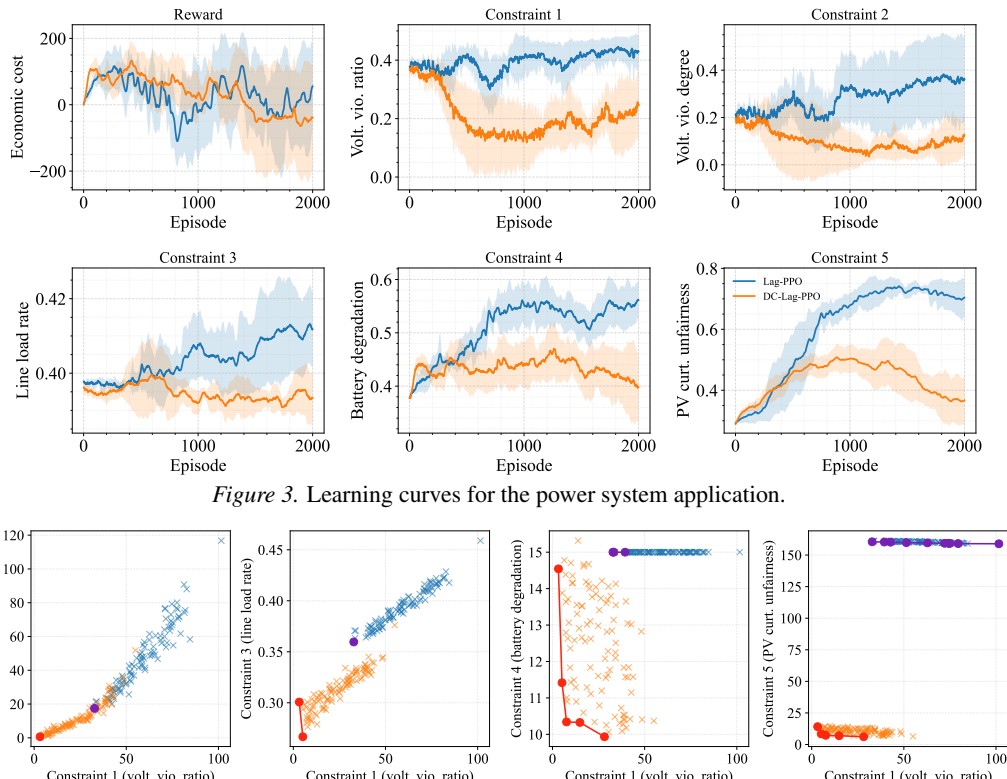

*Figure 3.* Learning curves for the power system application.

*Figure 4.* Pareto fronts from the test results of the power system application(the lower the better).

ing related constraints, including voltage limits, EV battery degradation, and charging demand satisfaction. In contrast to the previous problem, this setting involves coordinating multiple EVCSs, each operating dozens of chargers and responding to highly stochastic and heterogeneous EV behaviours (arrival and departure times, charging demands, battery capacities, etc.). As a result, both the state and action spaces are substantially higher-dimensional, and the additional uncertainty introduced by EV dynamics makes the EVCS coordination task considerably more challenging than community battery scheduling. Appendix L reports results for this environment, which are consistent with our findings: dedicated critics perform consistently better than mixed critics, achieving comparable or higher returns while substantially reducing average constraint violations and their variance across our benchmarks.

To assess whether our diagnosis extends beyond the power-system CMDP, we additionally test the mixed- vs. dedicated-critic design on six standard Safe-RL benchmarks with a *single* safety constraint. (See Appendix N for detail) Although these environments do not capture the full multi-constraint setting, dedicated critics still consistently improve safety (lower violation) across all six tasks and improve reward relative to the mixed-critic in some tasks. We also perform a *timescale intervention* to validate the causal mechanism behind mixed-critic degradation. In Appendix O and P (Robbins–Monro Lag-PPO), we enforce diminishing step-

sizes that effectively slow the dual updates (so $\beta_t/\eta_t \to 0$) and find that the gap between mixed and dedicated critics largely vanishes, with both variants exhibiting similar reward/constraint behaviour and similar $\lambda$ convergence. This controlled result is consistent with Theorem 4.7 and supports our claim that the mixed-critic failure mode is driven by dual-induced target drift when $\lambda$ updates are not sufficiently slow relative to critic tracking.

## 6. Conclusion

This study investigates the influence of critic architecture on stability and safety within Lagrangian (policy-gradient) approaches to constrained Markov decision processes. We showed, both theoretically and empirically, that mixing all reward and cost signals into a mixed critic couples the evaluation target to the evolving dual variables, introducing a form of dual-induced nonstationarity that can impair learning stability. In contrast, dedicated per-signal critics yield targets that depend solely on the policy, eliminating this source of drift. Our experiments across both bandit and stylized power system environments confirm these theoretical insights. This paper provides concrete guidance for the design of safe reinforcement learning algorithms under multiple constraints, highlighting the importance of critic architecture in ensuring both stability and constraint satisfaction in Lagrangian-based methods.

## Impact Statement

This work investigates algorithmic design choices for safe reinforcement learning in constrained Markov decision processes. All experiments are conducted exclusively in simulated environments; no human subjects, personal data, or identifiable information are involved at any stage. The complex energy-system environment used in our study is a stylized simulator designed to explore safety–performance trade-offs in high-stakes decision-making. It does not interface with or control any real-world infrastructure. Our aim is to advance the understanding of algorithmic safety in reinforcement learning without posing risks to individuals, communities, or operational systems.

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

## A. Limitations

A fundamental tension exists in safe RL between maximizing cumulative reward and satisfying multiple constraints, particularly in realistic, high-stakes domains. While our **Dedicated Critic** architecture significantly enhances constraint adherence and stabilizes training dynamics, it does not resolve the inherent trade-off: enforcing stricter safety often reduces achievable reward or slows convergence. The approach also introduces greater computational and memory requirements, as it maintains a separate critic for each constraint. This design may be less scalable in environments with many constraints or where critic updates are expensive. Although shared-backbone models with multiple heads offer a partial remedy, they require careful balancing and tuning to be effective.

Another limitation is the static one-to-one mapping between constraints and critics. In settings where constraints vary in relevance or activate sparsely, some critics may be under-trained, reducing sample efficiency (see Appendix G). Future directions could involve adaptive critic selection or shared-parameter architectures that dynamically reallocate capacity based on constraint salience.

The theoretical results provide asymptotic error bounds and bias characterisations, rather than full finite-time guarantees for PPO-style deep RL under realistic training regimes. However, this limitation is not specific to our approach: most practical deep RL algorithms rely on similar heuristic stabilisation mechanisms and likewise lack end-to-end finite-time guarantees. In our implementation, we mitigate these issues using standard techniques, including advantage normalisation, conservative learning rates, PPO ratio clipping, gradient clipping, and bounding the dual variables, and empirically observe stable learning across seeds. Nevertheless, a more refined finite-time analysis that explicitly captures these practical design choices remains an open direction, so our current guarantees should be interpreted as qualitative guidance on critic design rather than a complete convergence certificate for deep safe RL.

Finally, although our theoretical results are first presented under linear function approximation for analytical clarity, we also extend them to nonlinear cases such as neural networks. Still, understanding the implications of gradient bias in deep architectures, particularly when critics share representations, remains an open question warranting further theoretical and empirical investigation.

## B. Reproducibility Statement

To support the reproducibility of our results, we provide comprehensive details on training procedures, including step sizes, optimization parameters, and evaluation metrics used throughout the experiments. With the release of our codebase, we will ensure full transparency by including random seeds, environment specifications, dependency versions, and scripts necessary to replicate all experiments and figures. All results presented in the paper can be reproduced using the provided scripts without manual tuning. Where applicable, we will also include pretrained models and logs to facilitate result verification and benchmarking.

## C. PBE

We justify that, for each fixed policy $\pi_\theta$ and each signal $x \in \{r, c_1, \ldots, c_m\}$, the population target of the per-signal critic is the unique solution of a linear system $A(\theta)\, \omega^{x,*}(\theta) = b^x(\theta)$ with $A(\theta) \succeq \mu I$, and that $A(\cdot), b^x(\cdot)$ are locally Lipschitz in $\theta$ under standard conditions.

Let $\mathcal{E}^x_{\pi_\theta}(s)$ denote the discounted state value for signal $x$ under policy $\pi_\theta$ and let $\phi : \mathcal{S} \to \mathbb{R}^d$ be a fixed feature map. We approximate $\mathcal{E}^x_{\pi_\theta}(s) \approx \phi(s)^\top \omega^x$. Write $\Phi \in \mathbb{R}^{n \times d}$ for the matrix stacking feature rows $\phi(s)^\top$, $D = \mathrm{diag}(d_{\pi_\theta})$ for the diagonal matrix of the on-policy stationary distribution over states, and $P_{\pi_\theta}$ for the state transition kernel.

The projected fixed-point equation (PFE) in the $D$-weighted norm is

$$\Phi\omega^x \;=\; \Pi\left(\mathcal{T}^x_{\pi_\theta}(\Phi\omega^x)\right), \qquad \mathcal{T}^x_{\pi_\theta} v \;=\; r^x + \gamma P_{\pi_\theta} v,$$

where $\Pi$ is the $D$-orthogonal projection onto $\mathrm{span}(\Phi)$ and $r^x \in \mathbb{R}^n$ is the immediate signal vector ($r$ for $x = r$, $c_i$ for $x = c_i$). The normal equations are

$$\Phi^\top D\Big(\Phi\omega^x - (r^x + \gamma P_{\pi_\theta}\Phi\omega^x)\Big) = 0 \iff \underbrace{\Phi^\top D(I - \gamma P_{\pi_\theta})\Phi}_{A(\theta)}\, \omega^x \;=\; \underbrace{\Phi^\top D\, r^x}_{b^x(\theta)}.$$

Hence the population target satisfies $A(\theta)\,\omega^{x,*}(\theta) = b^x(\theta)$.

Assume (i) *ergodicity*: the Markov chain under $\pi_\theta$ admits a stationary distribution $d_{\pi_\theta}$ with full support on the on-policy visited set; (ii) *feature non-degeneracy*: the columns of $D^{1/2}\Phi$ are linearly independent. Then $A(\theta) = \Phi^\top D(I - \gamma P_{\pi_\theta})\Phi$ is symmetric positive definite; in particular there exists $\mu > 0$ with $A(\theta) \succeq \mu I$, so the solution $\omega^{x,*}(\theta)$ is unique.

If $\pi_\theta$ is $C^1$ in $\theta$ and ergodicity holds on a neighbourhood, then $P_{\pi_\theta}$ and $d_{\pi_\theta}$ vary locally Lipschitzly in $\theta$. Since $\Phi$ is fixed, $A(\theta) = \Phi^\top D(I - \gamma P_{\pi_\theta})\Phi$ and $b^x(\theta) = \Phi^\top D r^x$ inherit local Lipschitzness.

If the estimator depends on $(s,a)$, take features $\phi : \mathcal{S} \times \mathcal{A} \to \mathbb{R}^d$, stack $\Phi$ over $(s,a)$, let $D = \mathrm{diag}(d_{\pi_\theta}(s,a))$ be the on-policy state–action occupancy matrix, and use the state–action transition kernel $P^{\pi_\theta}$. The same PFE derivation yields

$$A(\theta) \;=\; \Phi^\top D\,(I - \gamma P^{\pi_\theta})\,\Phi, \qquad b^x(\theta) \;=\; \Phi^\top D\, r^x,$$

so $A(\theta)\omega^{x,*}(\theta) = b^x(\theta)$ with $A(\theta) \succeq \mu I$ under the analogues of ergodicity and feature non-degeneracy for $(s,a)$. Local Lipschitzness follows as above.

When training with advantages (e.g., GAE), two standard constructions lead to a linear system:

(1) *Difference-of-values:* Learn $V^x$ (or $Q^x$) with the state/state–action equations above, and form $A^x = Q^x - V^x$; the critic parameters still solve $A(\theta)\omega^{x,*} = b^x(\theta)$.

(2) *Least-squares to generalized returns:* Regress $\phi(z)^\top \omega^x$ onto generalized returns $\hat{G}^x$ (e.g., GAE targets) in the $D$-weighted norm, i.e. $\min_{\omega^x} \mathbb{E}_{z \sim d_{\pi_\theta}}\big[\,(\phi(z)^\top \omega^x - \hat{G}^x(z))^2\,\big]$. The normal equations are

$$\underbrace{\Phi^\top D\,\Phi}_{A(\theta)}\,\omega^{x,*}(\theta) \;=\; \underbrace{\Phi^\top D\,\hat{G}^x}_{b^x(\theta)}.$$

Thus the linear model still holds (with a different $A$), and $A(\theta) \succeq \mu I$ under $D^{1/2}\Phi$ full column rank. If $\hat{G}^x$ depends smoothly on $\theta$ through $\pi_\theta$, $b^x(\cdot)$ is locally Lipschitz.

**Lemma C.1** (Positive definiteness of $A(\theta)$). *Fix a policy $\pi_\theta$ and let $D = \mathrm{diag}(d_{\pi_\theta})$ be the diagonal matrix of the on-policy stationary distribution over states. Let $P_{\pi_\theta}$ be the corresponding state transition kernel (row-stochastic) satisfying $d_{\pi_\theta}^\top P_{\pi_\theta} = d_{\pi_\theta}^\top$. Let $\Phi \in \mathbb{R}^{n \times d}$ stack feature rows and assume $D^{1/2}\Phi$ has full column rank. For $\gamma \in [0,1)$ define*

$$A(\theta) \;=\; \Phi^\top D\,(I - \gamma P_{\pi_\theta})\,\Phi.$$

*Then $A(\theta)$ is symmetric positive definite and*

$$v^\top A(\theta)\,v \;\geq\; (1-\gamma)\,\lambda_{\min}\big(\Phi^\top D\,\Phi\big)\,\|v\|_2^2 \qquad \text{for all } v \in \mathbb{R}^d.$$

*In particular, $A(\theta) \succeq \mu I$ with $\mu = (1-\gamma)\,\lambda_{\min}(\Phi^\top D\Phi) > 0$.*

*Proof.* Let $y = \Phi v$. Using the $D$-weighted inner product $\langle u, w \rangle_D := u^\top D w$ and norm $\|u\|_D^2 := \langle u, u \rangle_D$,

$$v^\top A(\theta)\,v = y^\top D(I - \gamma P_{\pi_\theta})y = \|y\|_D^2 - \gamma\,\langle y, P_{\pi_\theta} y \rangle_D.$$

Because $P_{\pi_\theta}$ is a Markov operator with invariant measure $d_{\pi_\theta}$, it is a non-expansion in $L_2(D)$, i.e., $\|P_{\pi_\theta} y\|_D \leq \|y\|_D$ and therefore $\langle y, P_{\pi_\theta} y \rangle_D \leq \|y\|_D \|P_{\pi_\theta} y\|_D \leq \|y\|_D^2$. Hence

$$v^\top A(\theta)\,v \;\geq\; \|y\|_D^2 - \gamma\,\|y\|_D^2 = (1-\gamma)\,\|y\|_D^2.$$

Finally, $\|y\|_D^2 = v^\top \Phi^\top D\Phi\, v \geq \lambda_{\min}(\Phi^\top D\Phi)\,\|v\|_2^2$ because $D^{1/2}\Phi$ has full column rank. Combining the inequalities yields the claim. $\square$

**Corollary C.2** (State–action variant). *Let $D = \mathrm{diag}(d_{\pi_\theta}(s,a))$ be the on-policy state–action occupancy matrix, $P^{\pi_\theta}$ the state–action transition kernel (row-stochastic) with $d_{\pi_\theta}^\top P^{\pi_\theta} = d_{\pi_\theta}^\top$, and $\Phi$ stack features over $(s,a)$ with $D^{1/2}\Phi$ full column rank. Define*

$$A(\theta) \;=\; \Phi^\top D\,(I - \gamma P^{\pi_\theta})\,\Phi.$$

*Then $A(\theta) \succeq (1-\gamma)\,\lambda_{\min}(\Phi^\top D\Phi)\, I$ and is symmetric positive definite.*

# D. Detailed Proof for Mixed-critic Error Bound

**Lemma D.1** (Mixed-critic error bound). *If $A(\theta) \succeq \mu I$ uniformly, and the stepsizes satisfy Robbins–Monro conditions with $\alpha_t/\eta_t \to 0$ and $\beta_t/\eta_t \to 0$, then there exist constants $C_\lambda, C_\theta < \infty$ such that*

$$\limsup_{t\to\infty} \mathbb{E}\|e_t\| \ \leq \ \frac{C_\lambda}{\mu} \limsup_{t\to\infty} \frac{\beta_t}{\eta_t} \ + \ \frac{C_\theta}{\mu} \limsup_{t\to\infty} \frac{\alpha_t}{\eta_t} \ + \ O(1). \tag{17}$$

*Proof.* Recall the error recursion equation 4:

$$e_{t+1} = (I - \eta_t A(\theta_t))e_t + \underbrace{(\omega_t^\star - \omega_{t+1}^\star)}_{\text{target drift}} + \eta_t \zeta_{t+1} + \Delta_t^\theta.$$

Since $A(\theta_t) \succeq \mu I$, we have for all $v$, $\|(I - \eta_t A(\theta_t))v\| \leq (1 - \mu\eta_t)\|v\|$. Hence

$$\|(I - \eta_t A(\theta_t))e_t\| \ \leq \ (1 - \mu\eta_t) \|e_t\|. \tag{18}$$

Using the drift expansion equation 7,

$$\omega_t^\star - \omega_{t+1}^\star = -\sum_{i=1}^m (\lambda_{i,t+1} - \lambda_{i,t}) \, \omega^{c_i}(\theta_t) + O(\|\theta_{t+1} - \theta_t\|).$$

Assuming the PBE solutions $\omega^{c_i}(\theta)$ and $\omega^r(\theta)$ are Lipschitz in $\theta$ (true under our linear/PBE setup with $A(\theta)$, $b^x(\theta)$ smoothly varying), there exist constants $C_\lambda, C_\theta$ s.t.

$$\|\omega_t^\star - \omega_{t+1}^\star\| \ \leq \ C_\lambda \|\lambda_{t+1} - \lambda_t\| + C_\theta \|\theta_{t+1} - \theta_t\|. \tag{19}$$

By the definitions of the dual and actor steps, $\|\lambda_{t+1} - \lambda_t\| = O(\beta_t)$ and $\|\theta_{t+1} - \theta_t\| = O(\alpha_t)$, hence

$$\|\omega_t^\star - \omega_{t+1}^\star\| \ \leq \ C_\lambda \beta_t + C_\theta \alpha_t. \tag{20}$$

Write $\delta_t := (\omega_t^\star - \omega_{t+1}^\star) + \Delta_t^\theta$. By equation 20 and Lipschitz variation of $A(\theta), b(\theta)$ (collected in $\Delta_t^\theta$), there exists $\widetilde{C}_\theta$ s.t.

$$\mathbb{E}\|\delta_t\| \ \leq \ C_\lambda \beta_t + \widetilde{C}_\theta \alpha_t. \tag{21}$$

Using equation 18 and the triangle inequality,

$$\|e_{t+1}\| \ \leq \ (1 - \mu\eta_t)\|e_t\| + \|\delta_t\| + \eta_t\|\zeta_{t+1}\|.$$

Take conditional expectation and then total expectation. With $\mathbb{E}[\zeta_{t+1}|\mathcal{F}_t] = 0$ and $\mathbb{E}\|\zeta_{t+1}\|^2 \leq \sigma^2$, standard SA arguments (via a mean-square detour or BDG inequality) yield

$$\mathbb{E}\big[\eta_t\|\zeta_{t+1}\|\big] \ \leq \ C_{\text{noise}} \, \eta_t^2, \tag{22}$$

for some constant $C_{\text{noise}}$ (intuitively, the "linear in $\eta_t$" noise can be handled through a square-norm contraction; in the first-moment recursion it appears as $O(\eta_t^2)$). Hence, taking total expectation and applying equation 21 gives

$$\mathbb{E}\|e_{t+1}\| \ \leq \ (1 - \mu\eta_t) \, \mathbb{E}\|e_t\| \ + \ C_\lambda \beta_t \ + \ \widetilde{C}_\theta \alpha_t \ + \ C_{\text{noise}} \, \eta_t^2. \tag{23}$$

We now use a standard comparison lemma: if a nonnegative sequence $(x_t)$ satisfies

$$x_{t+1} \leq (1 - a_t)x_t + b_t, \quad a_t \in (0,1), \ \sum_t a_t = \infty, \ a_t \to 0,$$

then

$$\limsup_{t\to\infty} x_t \ \leq \ \limsup_{t\to\infty} \frac{b_t}{a_t}.$$

Applying this to equation 23 with $x_t = \mathbb{E}\|e_t\|$, $a_t = \mu\eta_t$ and

$$b_t = C_\lambda \beta_t + \widetilde{C}_\theta \alpha_t + C_{\text{noise}} \eta_t^2,$$

gives

$$\limsup_{t \to \infty} \mathbb{E}\|e_t\| \leq \frac{1}{\mu} \limsup_{t \to \infty} \left( \frac{C_\lambda \beta_t}{\eta_t} + \frac{\widetilde{C}_\theta \alpha_t}{\eta_t} + C_{\text{noise}} \eta_t \right).$$

Since $\eta_t \to 0$ and $\sum_t \eta_t^2 < \infty$, the last term contributes $O(1)$. Renaming $\widetilde{C}_\theta$ as $C_\theta$ yields equation 9. $\quad\square$

**Lemma D.2** (dedicated-critic tracking error). *Suppose $A(\theta) \succeq \mu I$ uniformly, Assumptions 4.1 and 4.3 hold, then there exists $\widetilde{C}_\theta < \infty$ such that for every $x \in \{r, c_1, \ldots, c_m\}$,*

$$\limsup_{t \to \infty} \mathbb{E}\|e_t^x\| \leq \frac{\widetilde{C}_\theta}{\mu} \limsup_{t \to \infty} \frac{\alpha_t}{\eta_t}. \tag{24}$$

*Proof.* Use $A(\theta_t) \succeq \mu I$:

$$\|e_{t+1}^x\| \leq \|(I - \eta_t A(\theta_t))e_t^x\| + \eta_t\|\zeta_{t+1}^x\| + \|\Delta_t^{\theta,x}\| \leq (1 - \mu\eta_t)\|e_t^x\| + \eta_t\|\zeta_{t+1}^x\| + C_\theta\|\theta_{t+1} - \theta_t\|.$$

Take conditional expectation given $\mathcal{F}_t$ and then expectation; by Assumption 4.3, $\mathbb{E}[\eta_t\|\zeta_{t+1}^x\|] \leq c\,\eta_t$ and yields $O(\eta_t^2)$ at the level of first-moment recursion. By Assumption 4.1, $\|\theta_{t+1} - \theta_t\| = O(\alpha_t)$. Thus,

$$\mathbb{E}\|e_{t+1}^x\| \leq (1 - \mu\eta_t)\,\mathbb{E}\|e_t^x\| + C_\theta\,\alpha_t + O(\eta_t^2).$$

Apply the standard SA comparison lemma for sequences of the form $x_{t+1} \leq (1 - a_t)x_t + b_t$ with $a_t = \mu\eta_t$ and $b_t = C_\theta\alpha_t + O(\eta_t^2)$, using $\sum_t \eta_t = \infty$, $\sum_t \eta_t^2 < \infty$, and $\alpha_t/\eta_t \to 0$. This yields

$$\limsup_{t \to \infty} \mathbb{E}\|e_t^x\| \leq \frac{C_\theta}{\mu} \limsup_{t \to \infty} \frac{\alpha_t}{\eta_t},$$

and absorbing constants into $\widetilde{C}_\theta$ gives equation 15. $\quad\square$

# E. Detailed Proof for Actor-gradient bias bound

**Lemma E.1** (Actor-gradient bias bound). *Let $\widehat{g}_t$ and $g_t^\star$ be the actor's estimated and ideal gradients,*

$$\widehat{g}_t = \mathbb{E}_t\big[\nabla_\theta \log \pi_{\theta_t}(a_t|s_t)\, \phi(s_t, a_t)^\top \omega_t\big], \qquad g_t^\star = \mathbb{E}_t\big[\nabla_\theta \log \pi_{\theta_t}(a_t|s_t)\, \phi(s_t, a_t)^\top \omega_t^\star\big], \tag{25}$$

*with critic error $e_t = \omega_t - \omega_t^\star$. Assume the score and features are bounded as*

$$\|\nabla_\theta \log \pi_{\theta_t}(a_t|s_t)\| \le G, \qquad \|\phi(s_t, a_t)\| \le L_\phi \quad a.s.$$

*Then the actor-gradient bias*

$$B_t \;:=\; \widehat{g}_t - g_t^\star$$

*satisfies*

$$\|B_t\| \;\le\; GL_\phi\, \|e_t\|. \tag{26}$$

*Proof.* Recall the definitions

$$\widehat{g}_t = \mathbb{E}\big[\nabla_\theta \log \pi_{\theta_t}(a_t|s_t)\, \phi(s_t, a_t)^\top \omega_t \,\big|\, \mathcal{F}_t\big], \quad g_t^\star = \mathbb{E}\big[\nabla_\theta \log \pi_{\theta_t}(a_t|s_t)\, \phi(s_t, a_t)^\top \omega_t^\star \,\big|\, \mathcal{F}_t\big],$$

and the critic error $e_t = \omega_t - \omega_t^\star$. Here $\mathcal{F}_t$ is the sigma-field generated by everything up to time $t$; in particular, $\theta_t, \lambda_t, \omega_t, \omega_t^\star, e_t$ are $\mathcal{F}_t$-measurable, while $(s_t, a_t)$ are drawn from $\pi_{\theta_t}$ at time $t$ and are not $\mathcal{F}_t$-measurable.

Using linearity of conditional expectation and $e_t = \omega_t - \omega_t^\star$,

$$\begin{aligned}
B_t &:= \widehat{g}_t - g_t^\star \\
&= \mathbb{E}\big[\nabla_\theta \log \pi_{\theta_t}(a_t|s_t)\, \phi(s_t, a_t)^\top (\omega_t - \omega_t^\star) \,\big|\, \mathcal{F}_t\big] \\
&= \mathbb{E}\big[\nabla_\theta \log \pi_{\theta_t}(a_t|s_t)\, \phi(s_t, a_t)^\top e_t \,\big|\, \mathcal{F}_t\big].
\end{aligned}$$

Since $e_t$ is $\mathcal{F}_t$-measurable, we can factor it outside the conditional expectation: for any random matrix/vector $X$ and $\mathcal{F}_t$-measurable (deterministic under $\mathbb{E}[\cdot|\mathcal{F}_t]$) vector $Y$,

$$\mathbb{E}[X\,Y \mid \mathcal{F}_t] \;=\; \mathbb{E}[X \mid \mathcal{F}_t]\,Y.$$

Applying this with $X := \nabla_\theta \log \pi_{\theta_t}(a_t|s_t)\, \phi(s_t, a_t)^\top$ and $Y := e_t$,

$$B_t = \mathbb{E}\big[\nabla_\theta \log \pi_{\theta_t}(a_t|s_t)\, \phi(s_t, a_t)^\top \,\big|\, \mathcal{F}_t\big]\, e_t.$$

Equivalently, without explicitly pulling out the matrix, we can directly bound the norm inside the conditional expectation as follows.

From the triangle inequality for norms, we have

$$\|B_t\| = \Big\|\mathbb{E}\big[\nabla_\theta \log \pi_{\theta_t}(a_t|s_t)\, \phi(s_t, a_t)^\top e_t \,\big|\, \mathcal{F}_t\big]\Big\| \;\le\; \mathbb{E}\big[\big\|\nabla_\theta \log \pi_{\theta_t}(a_t|s_t)\, \phi(s_t, a_t)^\top e_t\big\| \,\big|\, \mathcal{F}_t\big],$$

where we used Jensen's inequality for the convex function $x \mapsto \|x\|$ and conditional expectation.

Now use submultiplicativity of operator/vector norms:

$$\big\|\nabla_\theta \log \pi_{\theta_t}(a_t|s_t)\, \phi(s_t, a_t)^\top e_t\big\| \;\le\; \|\nabla_\theta \log \pi_{\theta_t}(a_t|s_t)\|\, \|\phi(s_t, a_t)\|\, \|e_t\|.$$

Here we regard the product $\psi\, \phi^\top e_t$ (with $\psi := \nabla_\theta \log \pi_{\theta_t}(a_t|s_t)$) as $(\psi\, \phi^\top)e_t$; the operator norm of the rank-1 matrix $\psi\, \phi^\top$ is $\|\psi\|\, \|\phi\|$.

Therefore,

$$\|B_t\| \;\le\; \mathbb{E}\big[\|\nabla_\theta \log \pi_{\theta_t}(a_t|s_t)\|\, \|\phi(s_t, a_t)\|\, \|e_t\| \,\big|\, \mathcal{F}_t\big].$$

Assume the standard boundedness conditions hold almost surely:

$$\|\nabla_\theta \log \pi_{\theta_t}(a_t|s_t)\| \le G, \qquad \|\phi(s_t, a_t)\| \le L_\phi.$$

Since $\|e_t\|$ is $\mathcal{F}_t$-measurable, we can treat it as a constant inside the conditional expectation. Hence,

$$\|B_t\| \leq \mathbb{E}\big[G\, L_\phi\, \|e_t\| \,\big|\, \mathcal{F}_t\big] = G\, L_\phi\, \|e_t\|.$$

This establishes the claimed Lipschitz bound

$$\|B_t\| \leq GL_\phi\|e_t\|,$$

which is precisely equation 11. $\qquad\square$

## F. Detailed Proof for Actor-gradient bias bound

**Theorem F.1** (Bias from a Mixed Critic). *Assume the conditions of Lemma 4.5 hold, and the score/features are uniformly bounded $\|\nabla_\theta \log \pi_{\theta_t}(a_t|s_t)\| \leq G$, $\|\phi(s_t, a_t)\| \leq L_\phi$ a.s. Then the actor-gradient bias $B_t = \widehat{g}_t - g_t^\star$ satisfies*

$$\limsup_{t\to\infty} \mathbb{E}\|B_t\| \leq GL_\phi \left( \frac{C_\lambda}{\mu} \limsup_{t\to\infty} \frac{\beta_t}{\eta_t} + \frac{C_\theta}{\mu} \limsup_{t\to\infty} \frac{\alpha_t}{\eta_t} \right). \tag{27}$$

*Proof.* By Lemma 4.6,

$$\|B_t\| \leq GL_\phi \|e_t\|.$$

Taking expectations preserves the inequality (monotonicity of $\mathbb{E}$):

$$\mathbb{E}\|B_t\| \leq GL_\phi \mathbb{E}\|e_t\|. \tag{28}$$

Lemma 4.5 states that, for some finite $C_\lambda, C_\theta$,

$$\limsup_{t\to\infty} \mathbb{E}\|e_t\| \leq \frac{C_\lambda}{\mu} \limsup_{t\to\infty} \frac{\beta_t}{\eta_t} + \frac{C_\theta}{\mu} \limsup_{t\to\infty} \frac{\alpha_t}{\eta_t} + O(1). \tag{29}$$

Here the $O(1)$ term collects vanishing contributions such as $O(\eta_t)$ from the noise control (cf. the proof of Lemma 4.5).

Taking $\limsup_{t\to\infty}$ on both sides of equation 28 and using equation 29 yields

$$\limsup_{t\to\infty} \mathbb{E}\|B_t\| \leq GL_\phi \limsup_{t\to\infty} \mathbb{E}\|e_t\| \leq GL_\phi \left( \frac{C_\lambda}{\mu} \limsup_{t\to\infty} \frac{\beta_t}{\eta_t} + \frac{C_\theta}{\mu} \limsup_{t\to\infty} \frac{\alpha_t}{\eta_t} + O(1) \right).$$

Since $G, L_\phi, C_\lambda, C_\theta, \mu$ are constants (independent of $t$), and $\limsup(O(1)) = 0$, we can drop the vanishing term to obtain exactly equation 12.

In Lemma 4.5, the $\frac{\beta_t}{\eta_t}$ contribution arises from the *dual-driven target drift* in the mixed-critic error dynamics (see the decomposition of $\omega_t^\star - \omega_{t+1}^\star$). Thus the bound equation 12 explicitly exposes the additional bias component inherited from the mixed critic's dependence on $\lambda$. $\qquad\square$

**Theorem F.2** (dedicated-critic bias). *Suppose Assumptions 4.1–4.3 hold, $A(\theta) \succeq \mu I$ uniformly, and $\|\nabla_\theta \log \pi_{\theta_t}(a_t|s_t)\| \leq G$, $\|\phi(s_t, a_t)\| \leq L_\phi$ a.s. Let $\widehat{g}_t$ and $g_t^\star$ be the actor's estimated and ideal gradients, the dedicated-critic actor bias be $B_t^{multi} := \widehat{g}_t^{multi} - g_t^\star$. Then*

$$\limsup_{t\to\infty} \mathbb{E}\|B_t^{multi}\| \leq GL_\phi \frac{\widetilde{C}_\theta}{\mu} \limsup_{t\to\infty} \frac{\alpha_t}{\eta_t}, \tag{30}$$

*Proof.* Define the ideal (mixed) gradient at time $t$ and its estimator as

$$g_t^\star = \mathbb{E}_t\bigg[\nabla_\theta \log \pi_{\theta_t}(a_t|s_t) \big(\phi(s_t, a_t)^\top \omega^r(\theta_t) - \sum_{i=1}^m \lambda_{i,t}\, \phi(s_t, a_t)^\top \omega^{c_i}(\theta_t)\big)\bigg],$$

$$\widehat{g}_t^{multi} = \mathbb{E}_t\bigg[\nabla_\theta \log \pi_{\theta_t}(a_t|s_t) \big(\phi(s_t, a_t)^\top \omega_t^r - \sum_{i=1}^m \lambda_{i,t}\, \phi(s_t, a_t)^\top \omega_t^{c_i}\big)\bigg]. \tag{31}$$

From equation 31 and linearity,

$$B_t^{\text{multi}} = \mathbb{E}_t\Big[\nabla_\theta \log \pi_{\theta_t}(a_t|s_t)\, \phi(s_t, a_t)^\top \Big(e_t^r - \sum_{i=1}^m \lambda_{i,t}\, e_t^{c_i}\Big)\Big],$$

where $e_t^x = \omega_t^x - \omega^x(\theta_t)$. Using Jensen, submultiplicativity, and boundedness of score and features,

$$\|B_t^{\text{multi}}\| \;\leq\; \mathbb{E}_t\Big[\|\nabla_\theta \log \pi_{\theta_t}(a_t|s_t)\|\, \|\phi(s_t, a_t)\|\Big(\|e_t^r\| + \sum_{i=1}^m \|\lambda_{i,t}\|\, \|e_t^{c_i}\|\Big)\Big] \;\leq\; GL_\phi\Big(\|e_t^r\| + \Lambda \max_i \|e_t^{c_i}\|\Big),$$

where $\Lambda = \sup_t \|\lambda_t\| < \infty$ due to projection onto a compact set. Taking expectations and $\limsup$,

$$\limsup_{t\to\infty} \mathbb{E}\|B_t^{\text{multi}}\| \;\leq\; GL_\phi\Big(1+\Lambda\Big) \limsup_{t\to\infty} \max_{x\in\{r,c_i\}} \mathbb{E}\|e_t^x\|.$$

Apply Lemma 4.8 to bound each $\mathbb{E}\|e_t^x\|$ by $\frac{\widetilde{C}_\theta}{\mu} \limsup \frac{\alpha_t}{\eta_t}$, and absorb $(1+\Lambda)$ into $\widetilde{C}_\theta$ (renaming the constant) to get equation 16. No $\beta_t$ term appears and the target drift involves only $\theta$ (rate $\alpha_t$), not $\lambda$. $\qquad\square$

# G. Dual-induced drift and linearity

Let the mixed critic be trained by minimizing any smooth population loss

$$\mathcal{L}_{\mathrm{mix}}(\omega; \theta, \lambda) \quad \text{(e.g., TD loss, Monte-Carlo/GAE regression, etc.).}$$

Because the scalarized signal is $r_\lambda := r - \sum_{i=1}^{m} \lambda_i c_i$, this loss depends *explicitly* on $\lambda$. Denote the population minimizer by $\omega^\star(\theta, \lambda) \in \arg\min_\omega \mathcal{L}_{\mathrm{mix}}(\omega; \theta, \lambda)$. At any (strict) local minimum, the first-order condition holds:

$$\nabla_\omega \mathcal{L}_{\mathrm{mix}}(\omega^\star(\theta, \lambda); \theta, \lambda) = 0.$$

Assume the Hessian $H(\theta, \lambda) := \nabla_{\omega\omega}^2 \mathcal{L}_{\mathrm{mix}}(\omega^\star(\theta, \lambda); \theta, \lambda)$ is nonsingular (standard local strong convexity around the solution). Then by the implicit function theorem, $\omega^\star$ is differentiable in $(\theta, \lambda)$ and

$$\frac{\partial \omega^\star}{\partial \lambda} = -H(\theta, \lambda)^{-1} \underbrace{\nabla_{\omega\lambda}^2 \mathcal{L}_{\mathrm{mix}}(\omega^\star(\theta, \lambda); \theta, \lambda)}_{\neq 0 \text{ generically}}.$$

Hence for small updates $(\Delta\theta, \Delta\lambda)$,

$$\omega^\star(\theta + \Delta\theta, \lambda + \Delta\lambda) - \omega^\star(\theta, \lambda) = \underbrace{\frac{\partial \omega^\star}{\partial \theta} \Delta\theta}_{\text{policy-induced drift}} + \underbrace{\frac{\partial \omega^\star}{\partial \lambda} \Delta\lambda}_{\text{dual-induced drift}} + o(\|\Delta\theta\| + \|\Delta\lambda\|).$$

The key point is that $\nabla_{\omega\lambda}^2 \mathcal{L}_{\mathrm{mix}} \neq 0$ whenever the training targets or TD errors inside $\mathcal{L}_{\mathrm{mix}}$ depend on $r_\lambda$ (which they do for any mixed critic). Therefore, $\partial\omega^\star/\partial\lambda \neq 0$ generically, and the *dual-induced drift* term proportional to $\Delta\lambda$ appears *regardless of linearity*. The linear case analysed in the main text is just the special instance where $\mathcal{L}_{\mathrm{mix}}$ yields normal equations $A(\theta)\omega = b^r(\theta) - \sum_i \lambda_i b^{c_i}(\theta)$, so that $\partial\omega^\star/\partial\lambda = -A(\theta)^{-1}[b^{c_1}(\theta), \ldots, b^{c_m}(\theta)]$ explicitly.

**Why dedicated critics avoid it.** For per-signal critics, each loss $\mathcal{L}_x(\omega^x; \theta)$ *does not* involve $\lambda$:

$$\nabla_\omega \mathcal{L}_x(\omega^{x,\star}(\theta); \theta) = 0 \qquad \Rightarrow \qquad \frac{\partial \omega^{x,\star}}{\partial \lambda} = 0.$$

Thus their targets drift only through $\theta$ (policy-induced), with *no* dual-induced component. When the actor later combines the already-computed per-signal estimates as $\omega^{\mathrm{mix}} = \omega^r - \sum_i \lambda_i \omega^{c_i}$, the $\lambda$'s appear *outside* the critics and do not change the critics' own population optima.

## H. Computational Resources

We implement all experiments using PyTorch-1.12 on an Ubuntu 18.04 server with two Intel Xeon Gold 6142M CPUs with 16 cores, 24G memory, and one NVIDIA 3090 GPU.

To further clarify constraint satisfaction during testing, Fig. 5 reports the measured computation time per training epoch under different numbers of Lagrangian critics (i.e., constraints). The results exhibit a clear linear trend, quantified by the fitted regression:

$$y = 0.00169x + 0.00776, \tag{32}$$

indicating stable and predictable scaling as the number of constraints increases. Importantly, the variance bars are small across all cases, showing that the training remains stable even when more constraints are introduced.

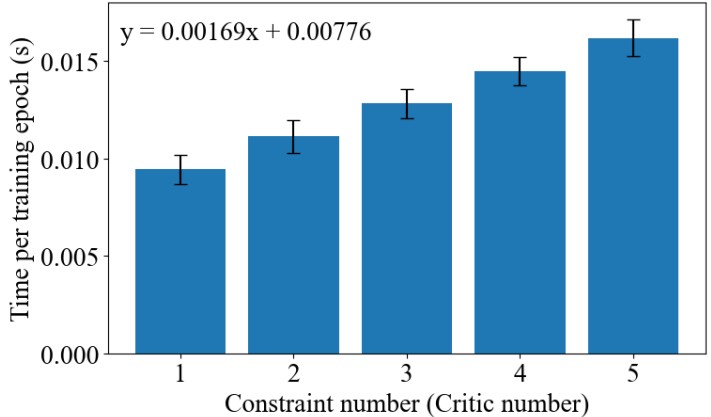

*Figure 5.* Time consumption of the proposed method with different number of critics (constraints).

## I. Experiment Detail - CMDP Bandit

**Single mixed critic:**    A *single* scalar critic per action is trained on the mixed signal

$$r_\lambda(a) \; := \; r(a) - \lambda_1 c_1(a) - \lambda_2 c_2(a).$$

With $\gamma = 0$, a TD(0) bandit update reduces to exponential averaging:

$$Q_{t+1}^{\text{mixed}}(a) \; = \; Q_t^{\text{mixed}}(a) + \eta_{\text{mixed}}\Big(r_\lambda(a_t) \; - \; Q_t^{\text{mixed}}(a_t)\Big)\mathbf{1}\{a_t = a\}. \tag{33}$$

The actor readout in equation 5.1 uses $\widehat{Q}_t(a) = Q_t^{\text{mixed}}(a)$.

**Dedicated-critic:**    We train *separate* per-action critics for reward and each cost:

$$Q_{t+1}^r(a) = Q_t^r(a) + \eta_{\text{multi}}\big(r(a_t) - Q_t^r(a_t)\big)\mathbf{1}\{a_t = a\}, \tag{34}$$

$$Q_{t+1}^{c_1}(a) = Q_t^{c_1}(a) + \eta_{\text{multi}}\big(c_1(a_t) - Q_t^{c_1}(a_t)\big)\mathbf{1}\{a_t = a\}, \tag{35}$$

$$Q_{t+1}^{c_2}(a) = Q_t^{c_2}(a) + \eta_{\text{multi}}\big(c_2(a_t) - Q_t^{c_2}(a_t)\big)\mathbf{1}\{a_t = a\}. \tag{36}$$

The actor combines them *at readout time* with the *current* multipliers:

$$\widehat{Q}_t(a) \; = \; Q_t^r(a) \; - \; \lambda_{1,t}\, Q_t^{c_1}(a) \; - \; \lambda_{2,t}\, Q_t^{c_2}(a). \tag{37}$$

We run $T = 5000$ steps per seed and average over $S = 15$ random seeds for the main curves. For the *timescale ablation* (Sec. I.2), we sweep critic and dual learning rates and average over 8 seeds per grid point. For a mixed scalar summary of conditional alignment (reported once), we optionally use $S = 20$ seeds to reduce variance. $\alpha = 0.02, \beta = 0.02, \eta_{\text{mixed}} = 0.03, \eta_{\text{multi}} = 0.03, \theta_0 = 0, \lambda_{1,0} = \lambda_{2,0} = 0.1, Q(\cdot) = 0$ for all heads at $t = 0$.

### I.1. Evaluation Metrics

**Expected reward.** We report the *on-policy* expected reward $J_r = \sigma(\theta) \, r(a_1) + (1 - \sigma(\theta)) \, r(a_2)$ as a function of steps.

**Constraint violation.** Instantaneous expected violation is

$$\text{Viol}_t \; := \; \max\big(0, \, J_{c_1}(\pi_{\theta_t}) - d_1\big) \; + \; \max\big(0, \, J_{c_2}(\pi_{\theta_t}) - d_2\big) \; = \; \max(0, \, 1 - \sigma(\theta_t) - 0.5) + \max(0, \, \sigma(\theta_t) - 0.5). \quad (38)$$

**Unconditional gradient alignment.** We compute a *moving* Pearson correlation between the estimated actor gradient $\widehat{g}_t$ (from equation 5.1 with the appropriate critic readout) and the true gradient $g_t$, using a centered window of width $w = 201$ with boundary normalization:

$$\text{corr}_t(\widehat{g}, g) \; = \; \frac{\text{Cov}_t(\widehat{g}, g)}{\sqrt{\text{Var}_t(\widehat{g}) \, \text{Var}_t(g)}}, \quad (39)$$

with $\text{Cov}_t(\cdot, \cdot), \; \text{Var}_t(\cdot)$ computed over the window and normalized by its effective length.

**Conditional gradient alignment.** Same as equation 39, but *restricted* to timesteps in the window where the ground-truth magnitude exceeds a threshold $\varepsilon = 10^{-3}$:

$$\text{corr}_t^{\text{cond}}(\widehat{g}, g) \; = \; \text{corr}\big(\{\widehat{g}_\tau : |g_\tau| > \varepsilon\}, \, \{g_\tau : |g_\tau| > \varepsilon\}\big) \quad (40)$$

This metric emphasizes periods with a meaningful learning signal, computed with boundary-normalized counts; windows with $< 5$ effective samples are masked.

**Dual oscillation magnitude.** We quantify multiplier oscillations via the *moving standard deviation* of the signed gap $|\lambda_2 - \lambda_1|$, again using a boundary-normalized window of width $w = 201$:

$$\text{Osc}_t \; = \; \sqrt{\max\big(0, \, \mathbb{E}_t[\Delta^2] - \big(\mathbb{E}_t[\Delta]\big)^2\big)}, \qquad \Delta_\tau := |\lambda_{2,\tau} - \lambda_{1,\tau}|. \quad (41)$$

**Smoothing (for curves) and uncertainty bands.** For reward and violation we plot *boundary-normalized* running means:

$$\widetilde{x}_t \; = \; \frac{\sum_{\tau=t-\lfloor w/2 \rfloor}^{t+\lfloor w/2 \rfloor} x_\tau}{\#\{\tau \text{ inside range}\}}, \quad (42)$$

then average $\widetilde{x}_t$ across seeds and show $\pm 1$ standard deviation bands across seeds.

### I.2. Timescale Ablation (Critic vs. Dual)

To mirror the theory's timescale conclusions, we sweep critic and dual learning rates on a grid:

$$\eta \in \{0.01, \, 0.03, \, 0.10\}, \qquad \beta \in \{0.005, \, 0.02, \, 0.08\},$$

holding the actor step $\alpha = 0.02$ fixed. For each $(\eta, \beta)$, we run the *mixed-critic* variant for $T = 5000$ steps with 8 seeds and report:

1. **Violation AUC:** $\sum_{t=1}^{T} \text{Viol}_t / T$,

2. **Late conditional alignment:** mean of equation 40 over the last 500 steps,

3. **Late dual oscillation:** mean of equation 41 over the last 500 steps.

Results are visualized as heatmaps over $(\eta, \beta)$.

### I.3. Compute, Randomization, and Reproducibility

All runs are CPU-only and complete within seconds. Random seeds $s \in \{1000, \ldots, 1000 + S - 1\}$ control action sampling only (initial parameters are deterministic). Each figure reports the mean across seeds with $\pm 1$ standard deviation. We save raw arrays (per-seed trajectories for reward, violation, gradients, and multipliers) to a serialized file for exact reproduction of all plots.

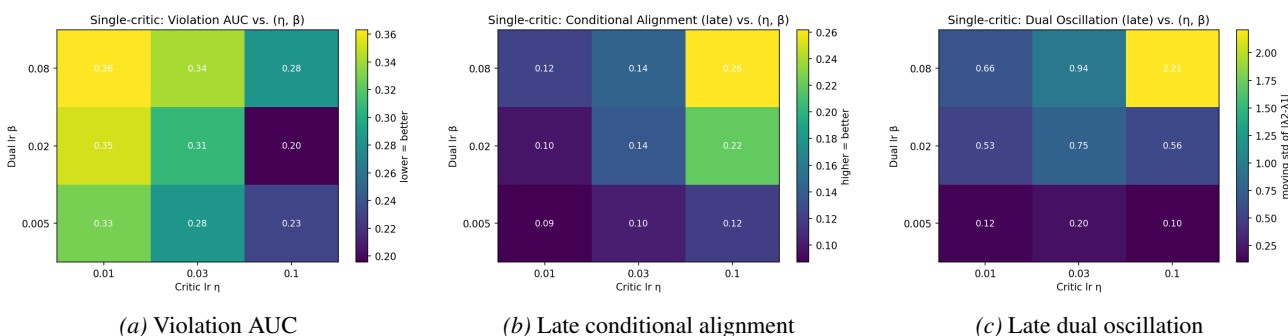

*(a)* Violation AUC        *(b)* Late conditional alignment        *(c)* Late dual oscillation

*Figure 6.* Performance for CMDP Bandit.

# J. Algorithm: Dedicated-Critic PPO-Lag

---

**Algorithm 1** Dedicated-Critic Lagrangian PPO (multi-constraint, single-constraint is special case)

---

1: Initialize policy parameters $\theta$; Initialize reward critic parameters $\phi_r$; Initialize cost critic parameters $\phi_{c_i}$ for $i = 1, \ldots, m$; Initialize dual variables $\lambda_i \leftarrow \lambda_{\text{init}} \geq 0$ for $i = 1, \ldots, m$
2: **for** iteration $k = 0, 1, 2, \ldots$ **do**
3:     Reset buffers $\{s_t, a_t, r_t, c_t^{(i)}, \text{done}_t, \log \pi_t^{\text{old}}, V_t^r, V_t^{c_i}\}_{t=0}^{T-1}$; $s_0 \leftarrow \mathcal{E}.\text{reset}()$
4:     **for** $t = 0, \ldots, T - 1$ **do**
5:         Sample $a_t \sim \pi_\theta(\cdot \mid s_t)$; $\log \pi_t^{\text{old}} \leftarrow \log \pi_\theta(a_t \mid s_t)$; $V_t^r \leftarrow V_r(s_t; \phi_r)$
6:         **for** $i = 1, \ldots, m$ **do**
7:             $V_t^{c_i} \leftarrow V_{c_i}(s_t; \phi_{c_i})$
8:         **end for**
9:         $s_{t+1}, r_t, \{c_t^{(i)}\}_{i=1}^m, \text{done}_t \leftarrow \mathcal{E}.\text{step}(a_t)$
10:        Store $(s_t, a_t, r_t, \{c_t^{(i)}\}, \text{done}_t, \log \pi_t^{\text{old}}, V_t^r, \{V_t^{c_i}\})$ in buffer
11:     **end for**
12:     **if** $s_T$ is terminal **then**
13:         $V_T^r \leftarrow 0, \quad V_T^{c_i} \leftarrow 0 \ \ \forall i$
14:     **else**
15:         $V_T^r \leftarrow V_r(s_T; \phi_r)$
16:         $V_T^{c_i} \leftarrow V_{c_i}(s_T; \phi_{c_i}) \ \ \forall i$
17:     **end if**
18:     Initialize $A_T^r \leftarrow 0$ and $A_T^{c_i} \leftarrow 0$ for all $i$
19:     **for** $t = T - 1, \ldots, 0$ **do**
20:         $\delta_t^r \leftarrow r_t + \gamma(1 - \text{done}_{t+1})V_{t+1}^r - V_t^r$
21:         $A_t^r \leftarrow \delta_t^r + \gamma \lambda_{\text{GAE}}(1 - \text{done}_{t+1})A_{t+1}^r$
22:         **for** $i = 1, \ldots, m$ **do**
23:             $\delta_t^{c_i} \leftarrow c_t^{(i)} + \gamma(1 - \text{done}_{t+1})V_{t+1}^{c_i} - V_t^{c_i}$
24:             $A_t^{c_i} \leftarrow \delta_t^{c_i} + \gamma \lambda_{\text{GAE}}(1 - \text{done}_{t+1})A_{t+1}^{c_i}$
25:         **end for**
26:         $R_t^r \leftarrow A_t^r + V_t^r$; $R_t^{c_i} \leftarrow A_t^{c_i} + V_t^{c_i} \ \ \forall i$
27:     **end for**
28:     **for** $t = 0, \ldots, T - 1$ **do**
29:         $A_t^{\text{Lag}} \leftarrow A_t^r - \sum_{i=1}^m \lambda_i A_t^{c_i}$
30:     **end for**
31:     **for** PPO epoch $e = 1, \ldots, K$ **do**
32:         **for** minibatch $\mathcal{M}$ **do**
33:             **for** $(s_t, a_t, \log \pi_t^{\text{old}}, A_t^{\text{Lag}}) \in \mathcal{M}$ **do**
34:                 $\log \pi_t \leftarrow \log \pi_\theta(a_t \mid s_t)$
35:                 $\rho_t \leftarrow \exp(\log \pi_t - \log \pi_t^{\text{old}})$
36:                 $\hat{L}_t \leftarrow \min\left(\rho_t A_t^{\text{Lag}}, \ \text{clip}(\rho_t, 1 - \epsilon, 1 + \epsilon)A_t^{\text{Lag}}\right)$
37:             **end for**
38:             $L_\pi \leftarrow -\frac{1}{|\mathcal{M}|} \sum_{t \in \mathcal{M}} \hat{L}_t$
39:             Update $\theta \leftarrow \theta - \alpha_\pi \nabla_\theta L_\pi$
40:             $L_V^r \leftarrow \frac{1}{|\mathcal{M}|} \sum_{t \in \mathcal{M}} \left(V_r(s_t; \phi_r) - R_t^r\right)^2$
41:             **for** $i = 1, \ldots, m$ **do**
42:                 $L_V^{c_i} \leftarrow \frac{1}{|\mathcal{M}|} \sum_{t \in \mathcal{M}} \left(V_{c_i}(s_t; \phi_{c_i}) - R_t^{c_i}\right)^2$
43:             **end for**
44:             $L_V \leftarrow L_V^r + \sum_{i=1}^m L_V^{c_i}$
45:             Update $\phi_r, \{\phi_{c_i}\} \leftarrow \phi_r, \{\phi_{c_i}\} - \alpha_V \nabla L_V$
46:         **end for**
47:     **end for**
48:     $\theta_{\text{old}} \leftarrow \theta$
49:     **for** $i = 1, \ldots, m$ **do**
50:         Estimate average cost $\widehat{J}_{c_i} \leftarrow \frac{1}{T} \sum_{t=0}^{T-1} c_t^{(i)}$
51:         $\lambda_i \leftarrow \max\left(0, \ \lambda_i + \alpha_\lambda(\widehat{J}_{c_i} - d_i)\right)$
52:     **end for**
53: **end for**

---

# K. Experiment Details - Case 1

## K.1. System Description

We model a radial distribution network with high rooftop PV penetration, where a set of community battery energy storage systems (CBESSs) are coordinated to ensure operational safety and efficiency. Each CBESS is constrained by efficiency, power, and state-of-charge (SOC) limits, and can exchange energy with the upstream grid within trading bounds, incurring both trading and degradation costs. When storage is saturated, PV curtailment at the bus level is introduced with fairness considerations to avoid disproportionate restrictions. The distribution network is described using the LinDistFlow approximation, including power balance, voltage regulation, and branch thermal limits. Voltage violations and line loading are penalized in the objective. The overall scheduling problem minimizes the aggregated penalties and costs associated with CBESS operations, grid trading, and PV curtailment fairness.

We consider a radial distribution network $(\mathcal{N}, \mathcal{L})$ operated by a DNSP over intra-day periods $t \in \mathcal{T} = \{1, \ldots, T\}$. The system model consists of CBESS operation, PV curtailment, and PDN-level constraints.

### K.1.1. CBESS

Let $\mathcal{M}$ denote the set of CBESSs. Each CBESS $m \in \mathcal{M}$ is connected to bus $\xi(m)$, with charging/discharging efficiencies $(\eta_m^{\text{ch}}, \eta_m^{\text{dis}})$, charging and discharging limits $(\overline{P}_m^{\text{ch}}, \overline{P}_m^{\text{dis}})$, and SOC range $[\underline{SOC}_m, \overline{SOC}_m]$. The charging/discharging power are $p_{m,t}^{\text{ch}}$ and $p_{m,t}^{\text{dis}}$, the reactive support is $q_{m,t}^{\text{CB}}$, and stored energy is $E_{m,t}$ with capacity $E_m^{\text{Cap}}$. Their dynamics are:

$$E_{m,t+1} = E_{m,t} + \eta_m^{\text{ch}} p_{m,t}^{\text{ch}} \Delta t - \frac{1}{\eta_m^{\text{dis}}} p_{m,t}^{\text{dis}} \Delta t, \tag{43a}$$

$$SOC_{m,t} = \frac{E_{m,t}}{E_m^{\text{Cap}}}, \quad \underline{SOC}_m \leq SOC_{m,t} \leq \overline{SOC}_m, \tag{43b}$$

$$0 \leq p_{m,t}^{\text{ch}} \leq \overline{P}_m^{\text{ch}}, \quad 0 \leq p_{m,t}^{\text{dis}} \leq \overline{P}_m^{\text{dis}}, \tag{43c}$$

$$p_{m,t}^{\text{ch}} \cdot p_{m,t}^{\text{dis}} = 0, \tag{43d}$$

$$(p_{m,t}^{\text{dis}} - p_{m,t}^{\text{ch}})^2 + (q_{m,t}^{\text{CB}})^2 \leq (S_m^{\text{CB}})^2, \tag{43e}$$

$$E_{m,0} = E_m^{\text{init}}. \tag{43f}$$

CBESSs also trade with the main grid through a ratio $\rho_{m,t}^{\text{trade}} \in [0, 1]$. With buy/sell prices $(\phi_t^{\text{buy}}, \phi_t^{\text{sell}})$, the trading cost is:

$$f_t^{\text{ET}} = \sum_{m \in \mathcal{M}} f_{m,t}^{\text{trade}}, \tag{44a}$$

$$f_{m,t}^{\text{trade}} = \phi_t^{\text{buy}} p_{m,t}^{\text{ch}} \rho_{m,t}^{\text{trade}} - \phi_t^{\text{sell}} p_{m,t}^{\text{dis}} \rho_{m,t}^{\text{trade}}, \tag{44b}$$

$$0 \leq p_{m,t}^{\text{ch}} \rho_{m,t}^{\text{trade}} \leq \overline{P}_m^{\text{trade,ch}}, \quad 0 \leq p_{m,t}^{\text{dis}} \rho_{m,t}^{\text{trade}} \leq \overline{P}_m^{\text{trade,dis}}. \tag{44c}$$

Battery degradation is approximated linearly:

$$f_t^{\text{BD}} = \sum_{m \in \mathcal{M}} c_m^{\text{deg}} (p_{m,t}^{\text{ch}} + p_{m,t}^{\text{dis}}), \tag{45a}$$

where $c_m^{\text{deg}} > 0$ is the degradation cost coefficient.

### K.1.2. PV CURTAILMENT

When all CBESSs are full, PV generation is curtailed via ratio $\gamma_{i,t} \in [0, 1]$:

$$\tilde{p}_{i,t}^{\text{PV}} = (1 - \gamma_{i,t}) p_{i,t}^{\text{PV}}, \tag{46a}$$

$$0 \leq \gamma_{i,t} \leq 1. \tag{46b}$$

Fairness is enforced by comparing each bus's curtailed ratio $\pi_i^{\text{curt}}$ with its proportional target $\pi_i^{\text{tar}}$:

$$f^{\text{PVF}} = \sum_{i \in \mathcal{N}} (\pi_i^{\text{curt}} - \pi_i^{\text{tar}})^2. \tag{47}$$

### K.1.3. PDN

The PDN is described by lossless LinDistFlow. For each branch $(i,j) \in \mathcal{L}$:

$$p_{ij,t} = \sum_{k:(j,k)\in\mathcal{L}} p_{jk,t} + p_{j,t}^{\text{load}} - \tilde{p}_{j,t}^{\text{PV}} - \sum_{m:\xi(m)=j} (p_{m,t}^{\text{dis}} - p_{m,t}^{\text{ch}}), \tag{48a}$$

$$q_{ij,t} = \sum_{k:(j,k)\in\mathcal{L}} q_{jk,t} + q_{j,t}^{\text{load}} - \sum_{m:\xi(m)=j} q_{m,t}^{\text{CB}}. \tag{48b}$$

Voltage drop is given by:

$$V_{j,t} = V_{i,t} - 2(r_{ij}p_{ij,t} + x_{ij}q_{ij,t}), \tag{49}$$

with bounds $\underline{V} \leq V_{i,t} \leq \overline{V}$. Penalties for voltage violations are:

$$f_t^{\text{VD}} = \sum_{i \in \mathcal{N}} \left( [V_{i,t} - \overline{V}]^+ + [\underline{V} - V_{i,t}]^+ \right), \tag{50a}$$

$$f_t^{\text{VN}} = \sum_{i \in \mathcal{N}} \mathbb{I}(V_{i,t} > \overline{V} \vee V_{i,t} < \underline{V}). \tag{50b}$$

Line loading penalty is:

$$f_t^{\text{LL}} = \sum_{(i,j)\in\mathcal{L}} r_{ij} \frac{p_{ij,t}^2 + q_{ij,t}^2}{V_0^2}, \tag{51a}$$

$$p_{ij,t}^2 + q_{ij,t}^2 \leq \overline{S}_{ij}^2. \tag{51b}$$

### K.1.4. OBJECTIVE

The goal is to coordinate CBESS operation under PV-rich PDNs to ensure network safety and efficiency. At each time step, CBESSs decide charging/discharging and grid trading ratios. The optimization problem is:

$$\min_{p^{\text{ch}},p^{\text{dis}},q_{m,t}^{\text{CB}},\rho^{\text{trade}}} \sum_{t \in \mathcal{T}} \left( f_t^{\text{VD}} + f_t^{\text{VN}} + f_t^{\text{LL}} + f_t^{\text{BD}} + f_t^{\text{ET}} \right) + f^{\text{PVF}}, \tag{52a}$$

$$\text{s.t.} \quad \text{equation 43, equation 44c, equation 46, equation 48, and equation 49.} \tag{52b}$$

### K.2. CMDP Modeling with Dedicated-Critic Lagrangian RL

We cast the CBESS coordination as a constrained Markov decision process (CMDP) $\left( \mathcal{S}, \mathcal{A}, P, r, \{c_i\}_{i=1}^m, \gamma, \{d_i\}_{i=1}^m \right)$, where $\mathcal{S}$ and $\mathcal{A}$ denote the state and action spaces, $P(\cdot|s,a)$ the transition kernel, $\gamma \in (0,1)$ the discount factor, $r(s,a)$ the reward, and $c_i(s,a)$ the cost signal for constraint $i$ with threshold $d_i$. Given a stochastic policy $\pi_\theta(a|s)$, define the discounted returns

$$J_r(\pi_\theta) = \mathbb{E}_\pi \left[ \sum_{t=0}^{\infty} \gamma^t r(s_t, a_t) \right], \quad J_{c_i}(\pi_\theta) = \mathbb{E}_\pi \left[ \sum_{t=0}^{\infty} \gamma^t c_i(s_t, a_t) \right]. \tag{53}$$

The CMDP objective is

$$\max_\theta J_r(\pi_\theta) \quad \text{s.t.} \quad J_{c_i}(\pi_\theta) \leq d_i, \quad i = 1, \ldots, m. \tag{54}$$

**Reward & costs from the PDN model.** Let the instantaneous penalties/costs at time $t$ be those defined in the system model: $f_t^{\text{VD}}, f_t^{\text{VN}}, f_t^{\text{LL}}, f_t^{\text{BD}}, f_t^{\text{ET}}$ and the daily PV-curtailment fairness term $f^{\text{PVF}}$. A practical partition is:

$$r(s_t, a_t) = -\left(\alpha_{\text{BD}} f_t^{\text{BD}} + \alpha_{\text{ET}} f_t^{\text{ET}}\right), \tag{55}$$

$$c_1(s_t, a_t) = f_t^{\text{VD}}, \quad c_2(s_t, a_t) = f_t^{\text{VN}}, \quad c_3(s_t, a_t) = f_t^{\text{LL}}, \tag{56}$$

and an episodic fairness constraint

$$C_4(\tau) \triangleq f^{\text{PVF}} \quad \text{with} \quad \mathbb{E}_\pi[C_4(\tau)] \leq d_4, \tag{57}$$

where $\tau$ denotes a full episode (day). If desired, $f^{\text{PVF}}$ can be spread as a per-step density $c_4(s_t, a_t)$ so that $\sum_t \gamma^t c_4(s_t, a_t)$ recovers the same daily target. The weights $\alpha_{\text{BD}}, \alpha_{\text{ET}} > 0$ reflect economic preferences. Alternative partitions (e.g., moving $f^{\text{ET}}$ into constraints) are also supported without changing the derivations below.

**Lagrangian relaxation with per-constraint critics.** Introduce multipliers $\lambda = (\lambda_1, \ldots, \lambda_m) \succeq 0$ and define

$$\mathcal{L}(\theta, \lambda) = J_r(\pi_\theta) - \sum_{i=1}^m \lambda_i \left(J_{c_i}(\pi_\theta) - d_i\right). \tag{58}$$

We perform the standard primal–dual updates:

$$\theta \text{ update: } \nabla_\theta \mathcal{L}(\theta, \lambda) = \nabla_\theta J_r(\pi_\theta) - \sum_{i=1}^m \lambda_i \nabla_\theta J_{c_i}(\pi_\theta), \tag{59}$$

$$\lambda \text{ update: } \lambda_i \leftarrow \Pi_{[0, \lambda_{\max}]}\left(\lambda_i + \beta\left[J_{c_i}(\pi_\theta) - d_i\right]\right), \tag{60}$$

where $\Pi$ denotes projection to stabilize $\lambda$.

**Signal-wise value functions and advantages.** For each signal $x \in \{r, c_1, \ldots, c_m\}$ define

$$Q_\pi^x(s, a) = \mathbb{E}_\pi\left[\sum_{t=0}^\infty \gamma^t x(s_t, a_t) \,\Big|\, s_0{=}s, a_0{=}a\right], \tag{61}$$

$$V_\pi^x(s) = \mathbb{E}_{a \sim \pi}[Q_\pi^x(s, a)], \quad A_\pi^x(s, a) = Q_\pi^x(s, a) - V_\pi^x(s). \tag{62}$$

Using the policy score function, the actor gradient becomes

$$\nabla_\theta \mathcal{L}(\theta, \lambda) = \mathbb{E}_\pi\Big[\nabla_\theta \log \pi_\theta(a|s) \underbrace{\left(A_\pi^r(s, a) - \sum_{i=1}^m \lambda_i A_\pi^{c_i}(s, a)\right)}_{\widetilde{A}_\pi(s,a)}\Big]. \tag{63}$$

**Per-constraint critics.** We learn one critic per signal $x \in \{r, c_1, \ldots, c_m\}$ with parameters $\omega_x$:

$$Q_{\omega_x}(s, a) \approx Q_\pi^x(s, a), \qquad \delta_t^x = x_t + \gamma\, Q_{\omega_x}(s_{t+1}, a_{t+1}) - Q_{\omega_x}(s_t, a_t), \tag{64}$$

and minimize $\mathbb{E}[(\delta_t^x)^2]$ (or use GAE to reduce variance). Advantages are estimated by $A_t^x$ (e.g., GAE($\lambda$)) and plugged into equation 63.

**PPO-style actor (with dedicated-critic advantage).** Let $r_t(\theta) = \frac{\pi_\theta(a_t|s_t)}{\pi_{\theta_{\text{old}}}(a_t|s_t)}$ and $\widetilde{A}_t = A_t^r - \sum_i \lambda_i A_t^{c_i}$. The clipped surrogate is

$$\mathcal{J}_{\text{PPO}}(\theta) = \mathbb{E}\Big[\min\left(r_t(\theta)\,\widetilde{A}_t,\ \text{clip}(r_t(\theta), 1{-}\epsilon, 1{+}\epsilon)\,\widetilde{A}_t\right)\Big] + \eta\,\mathbb{E}[\mathcal{H}(\pi_\theta(\cdot|s_t))], \tag{65}$$

where $\mathcal{H}$ is policy entropy and $\eta \geq 0$.

*Table 1.* Key hyperparameters, reward, and CMDP constraints for the energy management case study.

| Category | Hyperparameter / Term | Value | Notes / Definition |
|---|---|---|---|
| **Reward & constraints** | | | |
| Reward $r_t$ | $-f_t^{\mathrm{ET}}$ | – | $f_t^{\mathrm{ET}} = \sum_m \left( \phi_t^{\mathrm{buy}} p_{m,t}^{\mathrm{ch}} \rho_{m,t}^{\mathrm{trade}} - \phi_t^{\mathrm{sell}} p_{m,t}^{\mathrm{dis}} \rho_{m,t}^{\mathrm{trade}} \right)$ |
| Constraint 1 $c_t^{\mathrm{VN}}$ | $\frac{1}{|\mathcal{N}|} \sum_i \mathbb{I}\left(V_{i,t} \notin [\underline{V}, \overline{V}]\right)$ | $[0,1]$ | Count of voltage violations (normalized by bus count) |
| Constraint 2 $c_t^{\mathrm{VD}}$ | $\sum_i \left( [V_{i,t} - \overline{V}]^+ + [\underline{V} - V_{i,t}]^+ \right)$ | – | Degree of voltage violation (no extra scaling) |
| Constraint 3 $c_t^{\mathrm{LL}}$ | $\sum_{(i,j)} [\ell_{ij,t} - \tau^{\mathrm{line}}]^+$ | – | Line thermal overload beyond threshold (p.u. or %) |
| Constraint 4 $c_t^{\mathrm{BD}}$ | $\dfrac{\sum_m (p_{m,t}^{\mathrm{ch}} + p_{m,t}^{\mathrm{dis}}) \Delta t}{|\mathcal{M}| \overline{P}^{\mathrm{CB}} \Delta t}$ | $[0,1]$ | Battery degradation (throughput, normalized) |
| Constraint 5 $c_t^{\mathrm{PVF}}$ | $\dfrac{\mathrm{var}(\{\gamma_{i,t}\}_{i \neq 0})}{0.25}$ | $[0,1]$ | PV curtailment unfairness (variance normalized by max 0.25) |
| Lag-PPO constraint | $c_t^{\mathrm{VN}} + c_t^{\mathrm{VD}} + c_t^{\mathrm{LL}} + c_t^{\mathrm{BD}} + c_t^{\mathrm{PVF}}$ | - | Summation of all constraints |
| **General training parameters** | | | |
| Learning rate $\alpha/\eta$ | – | 3e-4 | Shared by actor/critic |
| Clip coefficient | – | 0.2 | Ratio clipping $[1-\epsilon, 1+\epsilon]$ |
| Target KL | – | 0.015 | Early stop when approx-KL exceeds threshold |
| Value loss coeff. | – | 0.5 | Weight on value loss |
| Entropy coeff. | – | 0.0 | Entropy regularization |
| Grad norm clip | – | 0.5 | Global gradient clipping |
| Hidden sizes | – | (256, 256) | MLP for actor/critic |
| Init log-std | – | -0.5 | Gaussian policy init |
| Discount $\gamma$, GAE | – | 0.99, 0.95 | For returns and advantages |
| Dual learning rate $\lambda$ | – | 5e-3 | Step size for dual updates in Lagrangian RL paradigm |
| $\lambda$ init / max | – | 0.0 / $10^4$ | Projected to $[0, \lambda_{\max}]$ |
| **Training schedule & environment** | | | |
| PPO episodes | – | 2000 | Total training episodes |
| Steps / episode | – | 288 | $\Delta t = 5$ min $\Rightarrow$ one day per episode |
| Env time step | – | 5 min | Day length = 288 steps |

**Episodic fairness constraint.** If keeping $f^{\mathrm{PVF}}$ as episodic, use the per-episode estimator $\widehat{J}_{c_4} = \frac{1}{N} \sum_{k=1}^N C_4(\tau^{(k)})$ in equation 83. A practical alternative is to define a per-step density $c_4(s_t, a_t)$ whose discounted sum equals the daily fairness value, enabling a standard critic update as in equation 88.

**Concrete instantiation for this problem.** With equation 75–equation 57, we have $m \in \{3, 4\}$ constraints:

$$\text{Critics: } Q_{\omega_r} \text{ for reward,} \quad Q_{\omega_{c_1}}, Q_{\omega_{c_2}}, Q_{\omega_{c_3}} \text{ (and } Q_{\omega_{c_4}} \text{ if episodic fairness is densified);} \tag{66}$$

$$\text{Advantage: } \widetilde{A}_t = A_t^r - \lambda_1 A_t^{c_1} - \lambda_2 A_t^{c_2} - \lambda_3 A_t^{c_3} \; (-\lambda_4 A_t^{c_4} \text{ if used);} \tag{67}$$

$$\text{Dual: } \lambda_i \leftarrow \Pi_{[0,\lambda_{\max}]}\left( \lambda_i + \beta \left[ \widehat{J}_{c_i} - d_i \right] \right), \quad i = 1, \ldots, m. \tag{68}$$

**Notes on stability and practice.** (i) Use separate target networks or Polyak averaging for each critic to stabilize TD. (ii) Normalize every $A_t^x$ before forming $\widetilde{A}_t$ to balance scales across constraints. (iii) Choose $d_i$ from engineering limits (e.g., allowable daily voltage violation budget, line loading budget); start with conservative $d_i$ then relax. (iv) Bound $\lambda$ via projection or log-parameterization to avoid runaway dual ascent; optionally add a small L2 penalty on $\lambda$. (v) For mixed episodic/step constraints, update episodic multipliers once per episode and stepwise ones per minibatch.

### K.3. Experimental Parameters

**Symbols.** The key parameters of the power system management case study are provided in Table 3. $\phi_t^{\mathrm{buy}}, \phi_t^{\mathrm{sell}}$: upstream buy/sell prices; $\rho_{m,t}^{\mathrm{trade}} \in [0,1]$: trading ratio for CBESS $m$; $[x]^+ = \max\{x, 0\}$; $\mathbb{I}(\cdot)$: indicator; $\underline{V}, \overline{V}$: voltage bounds (e.g., $[1-\nu, 1+\nu]$ p.u., $\nu > 0$); $V_{i,t}$: bus-$i$ voltage; $\ell_{ij,t}$: loading of line $(i,j)$ (p.u. or %); $\tau^{\mathrm{line}}$: overload threshold (default 0); $\gamma_{i,t} \in [0,1]$: PV curtailment ratio at bus $i$; $\Delta t$: step duration (5 min); $T$: daily horizon (288 steps); $|\mathcal{N}|$: number of buses; $|\mathcal{M}|$: number of CBESS; $\overline{P}^{\mathrm{CB}}$: nameplate active-power rating for normalization.

### K.4. Two-Tiered Statistics

The two-tiered statistical results in Table 2 highlight a clear trade-off between economic performance and system safety. Specifically, the DC-Lag-PPO variant achieves substantial improvements across all five constraint metrics. The violation ratio (c1) and violation degree (c2) of bus voltages are reduced by approximately 33% and 52%, respectively, while the line loading rate (c3) decreases by 11.6%. Similarly, the battery degradation cost (c4) drops by 46%, and the PV curtailment unfairness (c5) improves by 52.7%. These reductions indicate that DC-Lag-PPO enforces network security and operational fairness more effectively than the baseline Lag-PPO.

In contrast, the reward, which reflects the economic cost, declines significantly (-87.2%). Since higher reward is preferred, this suggests that DC-Lag-PPO sacrifices economic efficiency to achieve stronger compliance with safety and fairness constraints. The mechanism is likely due to more conservative charging, discharging, and trading behaviors encouraged by the tightened constraint handling.

In terms of stability, the across-run standard deviations of c1, c2, and c3 decrease considerably, demonstrating more consistent performance in voltage and line-loading metrics. However, the standard deviation of PV curtailment fairness (c5) increases, implying reduced consistency across different runs in this aspect. This suggests that while DC-Lag-PPO reliably improves most safety indicators, its fairness outcomes may vary depending on specific training trajectories.

Overall, DC-Lag-PPO demonstrates its effectiveness as a safer policy with stronger constraint satisfaction, albeit at the cost of economic performance. Future work may seek to balance this trade-off by tuning constraint thresholds, adjusting the dual update step size, or normalizing advantage signals across constraints to prevent overly conservative policies.

*Table 2.* Two-tiered test statistics, where across-run mean $\pm$ across-run std; The higher reward is better, while the lower constraints are better. $\Delta$ = (DC-Lag-PPO $-$ Lag-PPO). Positive improvement % is computed as Lag-PPO $-$ DC-Lag-PPO/Lag-PPO $\times$ 100%.

| Metric | Lag-PPO (n=9) | DC-Lag-PPO (n=9) | $\Delta$ | Improvement % |
|---|---|---|---|---|
| Economic cost (reward) | $25.95 \pm 63.25$ | $3.32 \pm 34.68$ | **-22.64** | **-87.23%** |
| Volt. vio. ratio (c1) | $62.96 \pm 29.41$ | $42.17 \pm 11.61$ | **-20.80** | **+33.03%** |
| Volt. vio. degree (c2) | $54.77 \pm 36.31$ | $26.47 \pm 10.69$ | **-28.30** | **+51.67%** |
| Line load rate (c3) | $0.405 \pm 0.062$ | $0.358 \pm 0.024$ | **-0.047** | **+11.59%** |
| Battery degradation (c4) | $32.23 \pm 13.47$ | $17.35 \pm 11.21$ | **-14.88** | **+46.17%** |
| PV curt. unfairness (c5) | $210.95 \pm 23.09$ | $99.82 \pm 62.79$ | **-111.13** | **+52.68%** |

### K.5. Training Curves

As shown in Fig. 7 - 12, the training curves across multiple runs consistently highlight the strengths of DC-Lag-PPO in terms of constraint satisfaction. While the reward trajectories show that DC-Lag-PPO tends to converge to lower economic returns compared to the baseline Lag-PPO, the improvement in constraint metrics is substantial.

First, the voltage violation metrics (both ratio and degree) are markedly reduced under DC-Lag-PPO. The curves demonstrate faster convergence to lower levels of violations and maintain stability across episodes, especially in Fig. 11 and 12. This indicates that the dual-critic structure effectively penalizes unsafe voltage states, leading to more secure system operation.

Second, the line loading rates remain consistently lower for DC-Lag-PPO. Although the difference is modest compared to voltage metrics, the reduced variance in the curves reflects more stable utilization of line capacity, especially in Fig. 8 and 10.

Third, battery degradation under DC-Lag-PPO is substantially lower. The curves show that the algorithm learns to avoid excessive charging and discharging cycles, which not only improves system longevity but also reduces long-term operational costs.

Finally, PV curtailment unfairness also benefits significantly from DC-Lag-PPO. Although variance is occasionally higher across runs, the overall trajectory converges to much lower unfairness compared to Lag-PPO. This suggests that DC-Lag-PPO is able to balance curtailment more evenly across the network, enhancing fairness.

Overall, the training results confirm that DC-Lag-PPO enforces operational safety and fairness more effectively than the baseline. The cost of this improvement is a reduction in reward, implying that the method prioritizes constraint satisfaction over immediate economic gains. From a practical perspective, this trade-off can be acceptable or even desirable

in safety-critical power systems, where violations may carry severe penalties or risks.

Future extensions could explore adaptive balancing mechanisms, such as dynamic adjustment of dual learning rates or reward re-weighting, to recover part of the economic performance while maintaining the strong safety guarantees observed here.

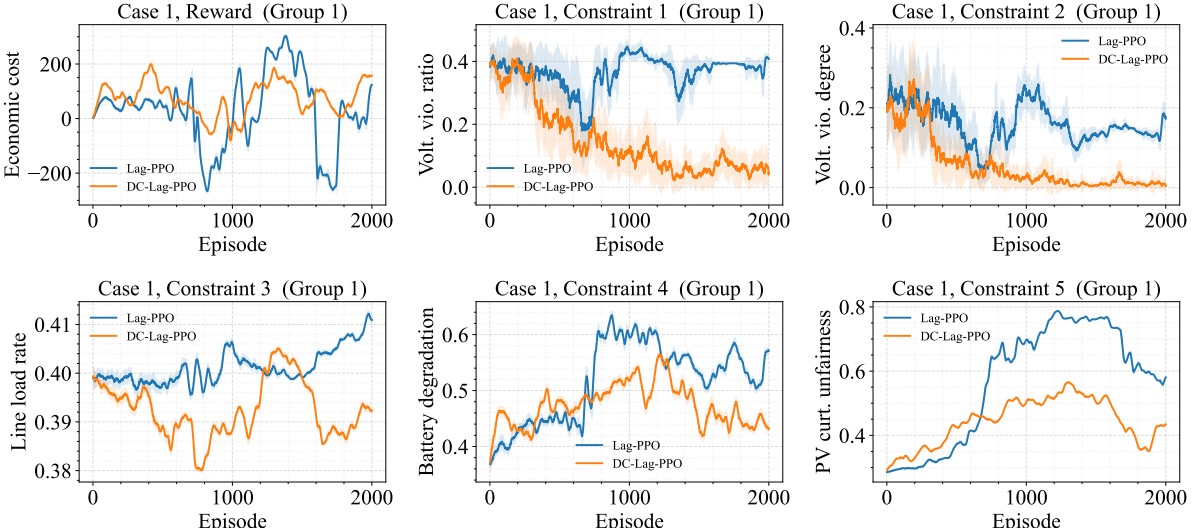

*Figure 7.* Training curves on Lagrangian cost threshold set: [9,9,0.1,30,30].

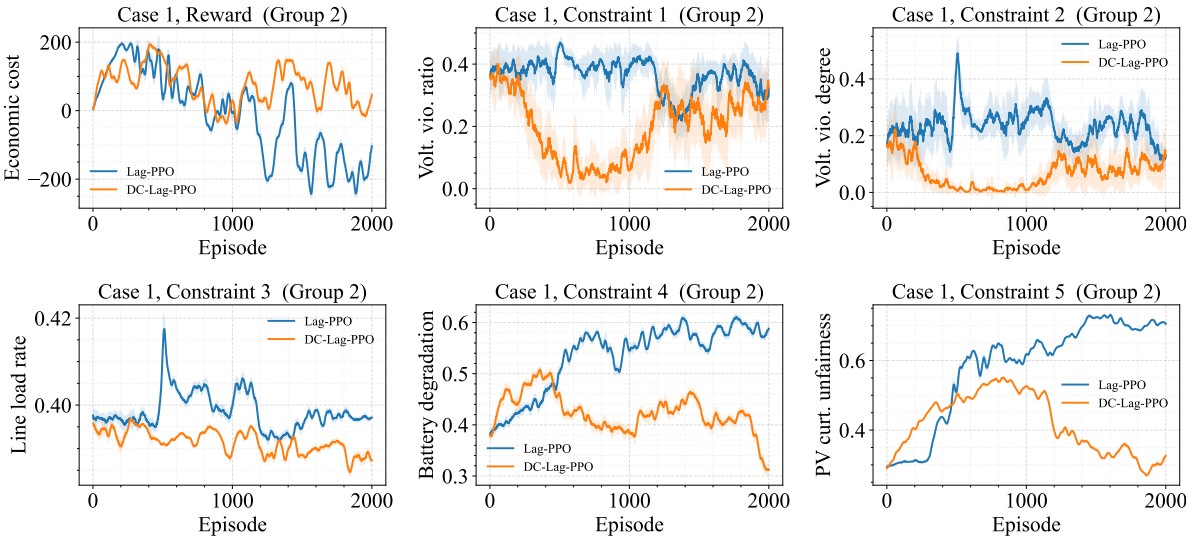

*Figure 8.* Training curves on Lagrangian cost threshold set: [12,12,0.1,30,30].

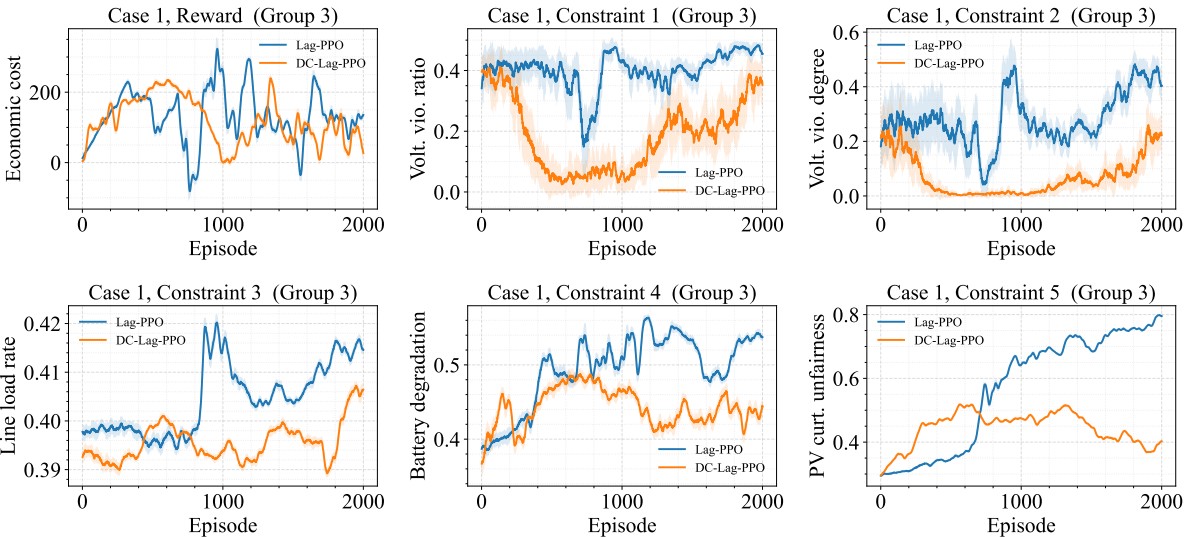

*Figure 9.* Training curves on Lagrangian cost threshold set: [15,15,0.1,30,30].

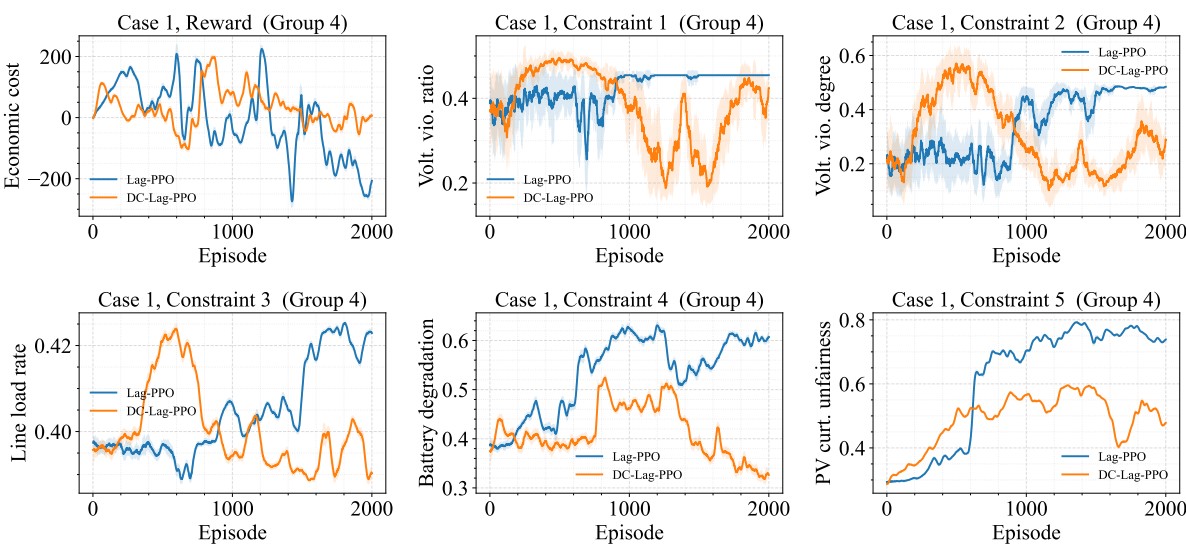

*Figure 10.* Training curves on Lagrangian cost threshold set: [18,18,0.1,20,30].

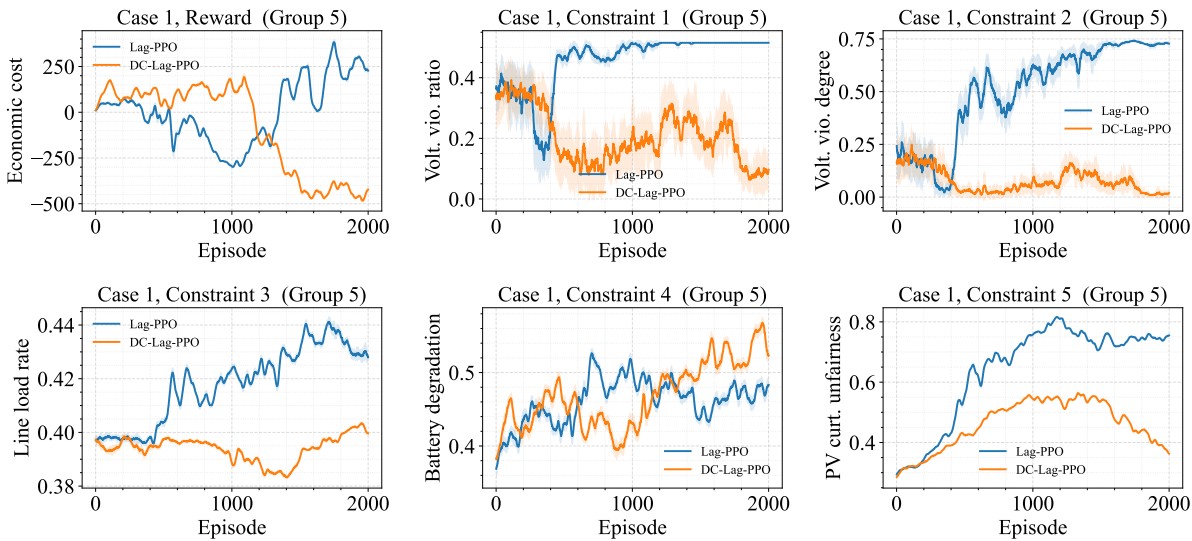

*Figure 11.* Training curves on Lagrangian cost threshold set: [9,9,0.1,30,20].

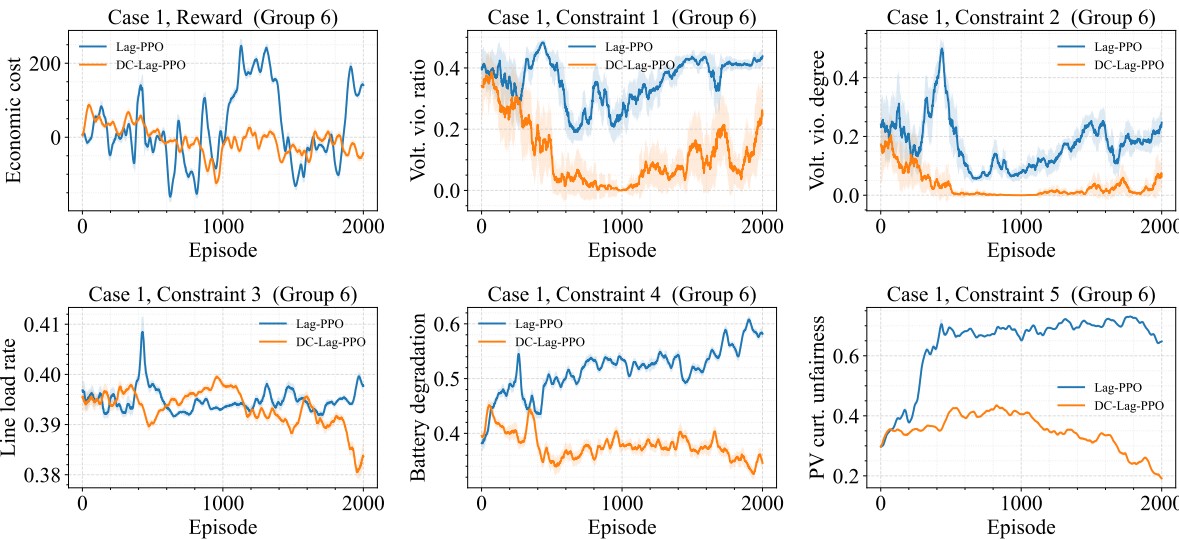

*Figure 12.* Training curves on Lagrangian cost threshold set: [7,7,0.1,15,15].

Additionally, $J_c$ curves are demonstrated in Fig. 13-18. Across all parameter settings, the plots consistently show a clear difference between Lag-PPO and our DC-Lag-PPO. The $J_c$ curve of Lag-PPO represents the summed constraint cost, and it frequently drifts far above the allowed threshold, exhibiting large fluctuations during training (e.g., group 1 and group 5). This indicates that a single shared Lagrange multiplier cannot effectively regulate multiple heterogeneous constraints.

In contrast, DC-Lag-PPO decomposes the constraint cost into five independent components, each with its own critic. The corresponding $J_c$ curves tightly track their respective thresholds across all settings, for large thresholds (e.g., 18, 20, 30) and even for very small ones (d = 0.1). This demonstrates precise constraint satisfaction and significantly improved stability. Therefore, the decomposed multi-critic structure is fundamentally more effective for enforcing multi-constraint safety compared to the single-critic Lag-PPO.

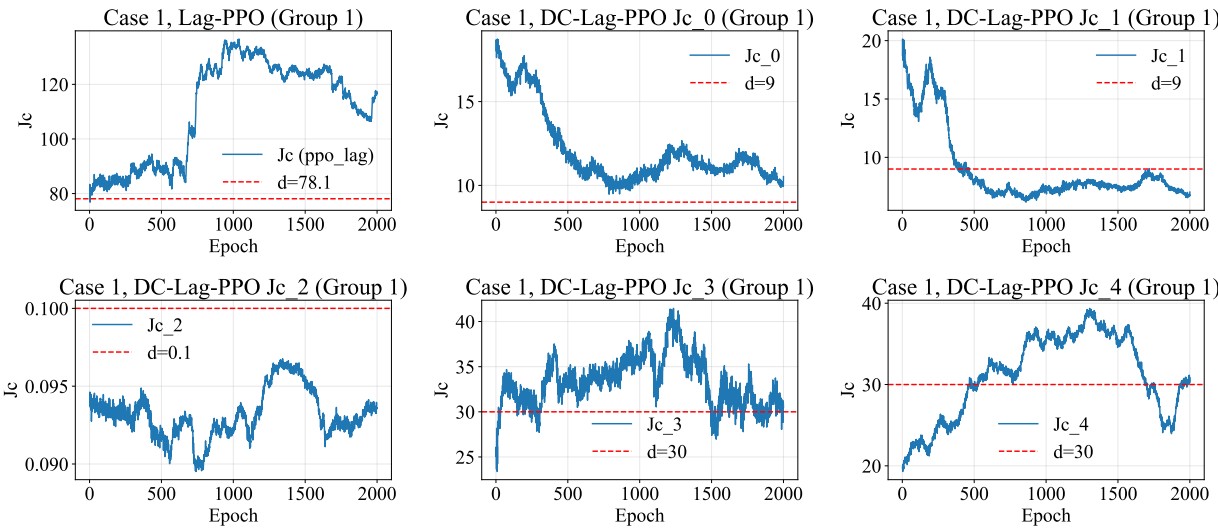

*Figure 13.* Training curves of $J_c$ on Lagrangian cost threshold set: [9,9,0.1,30,30].

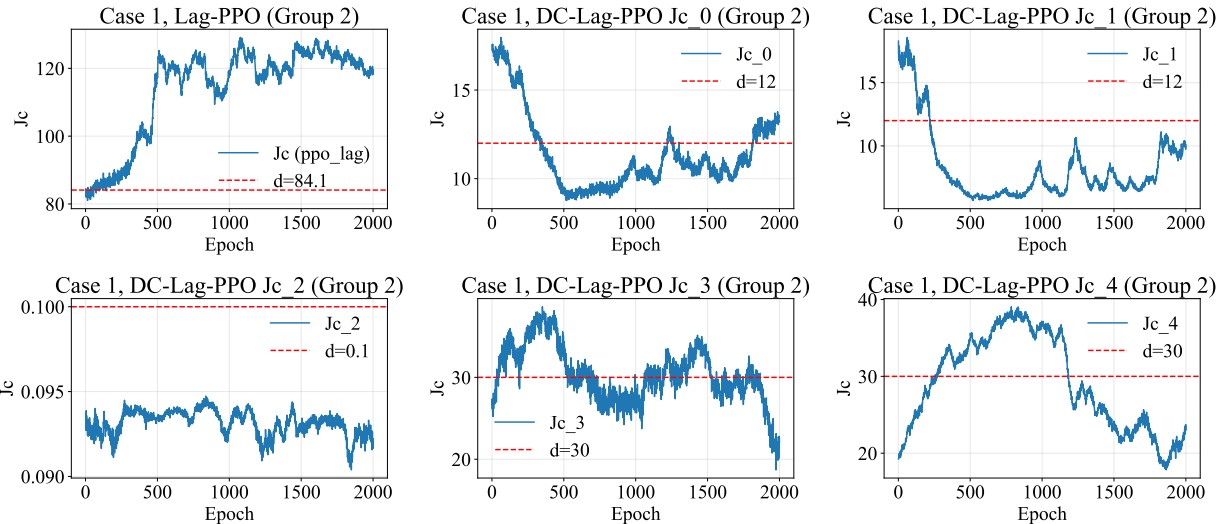

*Figure 14.* Training curves of $J_c$ on Lagrangian cost threshold set: [12,12,0.1,30,30].

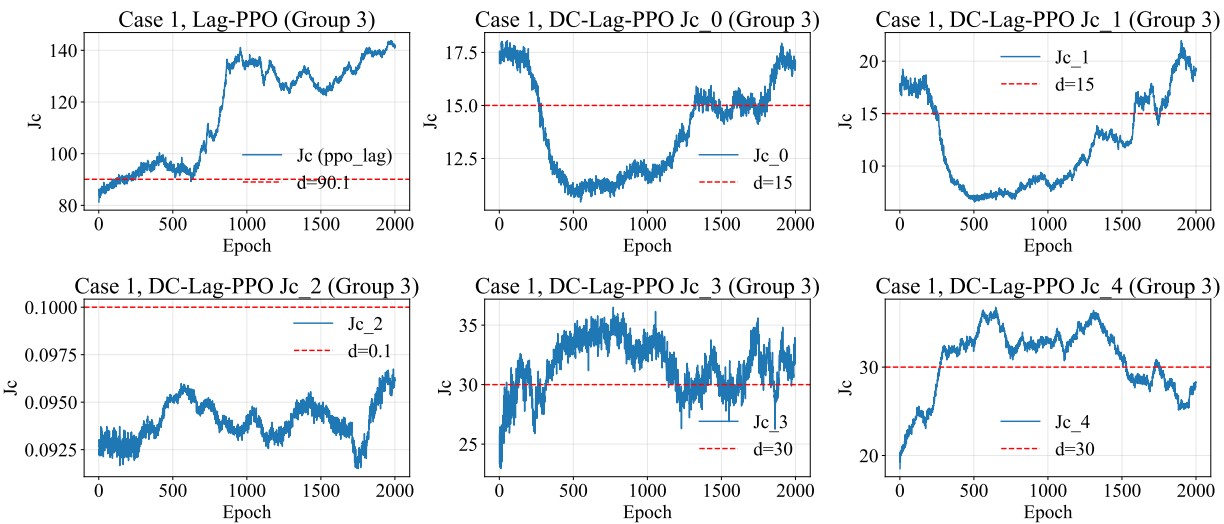

*Figure 15.* Training curves of $J_c$ on Lagrangian cost threshold set: [15,15,0.1,30,30].

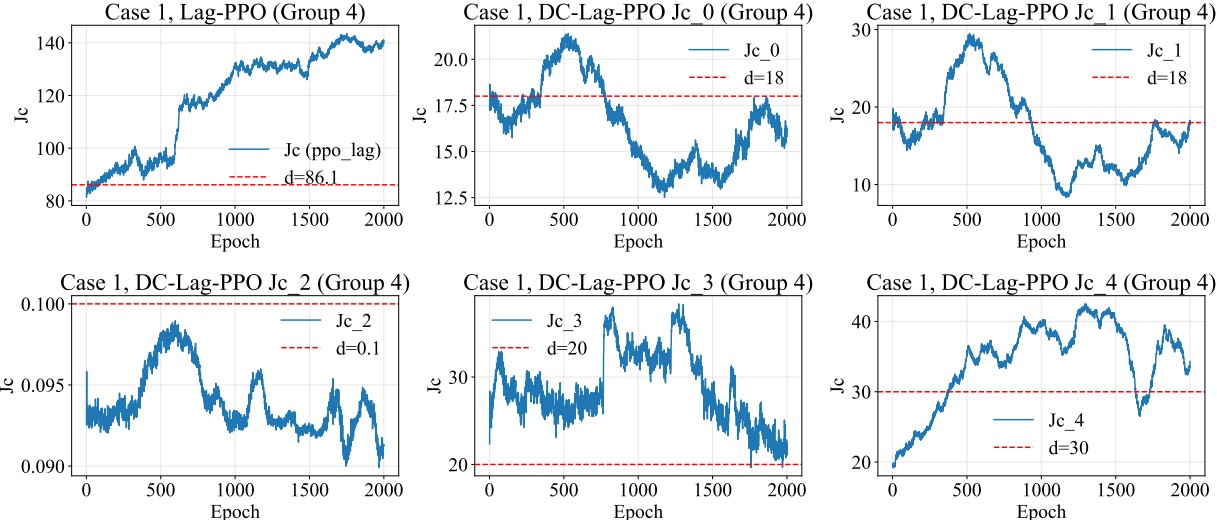

*Figure 16.* Training curves of $J_c$ on Lagrangian cost threshold set: [18,18,0.1,20,30].

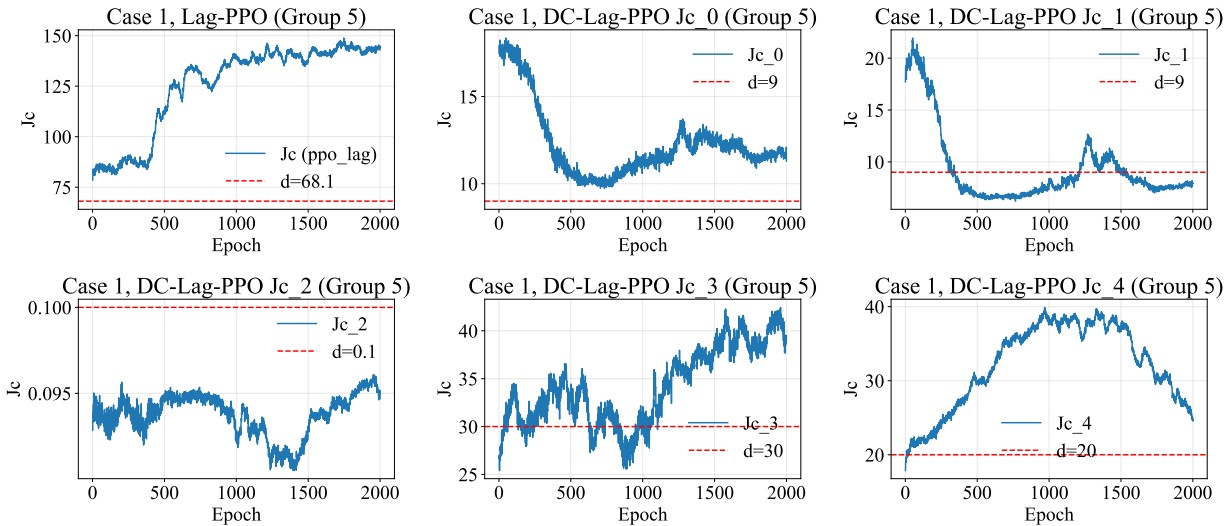

*Figure 17.* Training curves of $J_c$ on Lagrangian cost threshold set: [9,9,0.1,30,20].

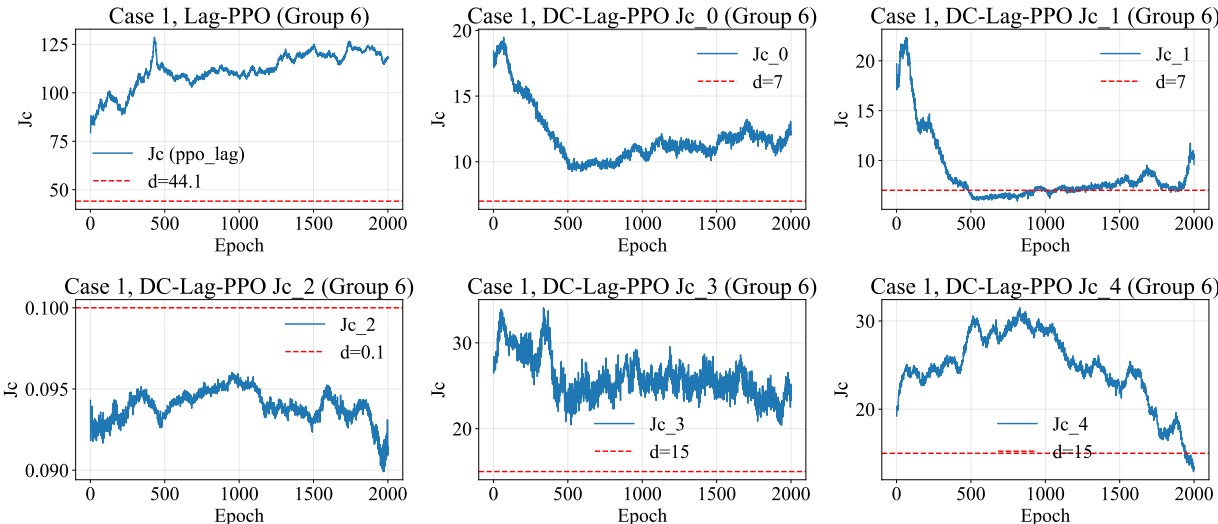

*Figure 18.* Training curves of $J_c$ on Lagrangian cost threshold set: [7,7,0.1,15,15].

## K.6. Lagrangian Multiplier Learning Curves

As shown in Fig. 19, the trajectories of the Lagrange multipliers provide insight into how Lag-PPO and DC-Lag-PPO enforce constraints during training. In the baseline Lag-PPO, the single multiplier tends to grow rapidly and exhibit instability, reflecting difficulty in balancing multiple heterogeneous constraints with a single aggregated signal. In contrast, DC-Lag-PPO assigns a dedicated multiplier to each constraint, and the resulting curves show more moderate growth and better separation among the multipliers. This indicates that the algorithm is able to distinguish between constraints of varying tightness and adjust enforcement accordingly.

Although some multipliers in DC-Lag-PPO still reach relatively high values, the spread across constraints suggests that the framework avoids over-penalizing all dimensions uniformly. Instead, it allocates stricter penalties only where violations are more prevalent. This aligns with the earlier observation that DC-Lag-PPO substantially reduces voltage violations, line overloads, and battery degradation, even though economic rewards are diminished.

Overall, the Lagrange multiplier dynamics confirm that DC-Lag-PPO enforces constraints in a more structured and interpretable way than Lag-PPO. By disentangling constraint signals, it achieves stronger and more balanced compliance

with operational limits, providing a safer and more reliable control policy for power system management.

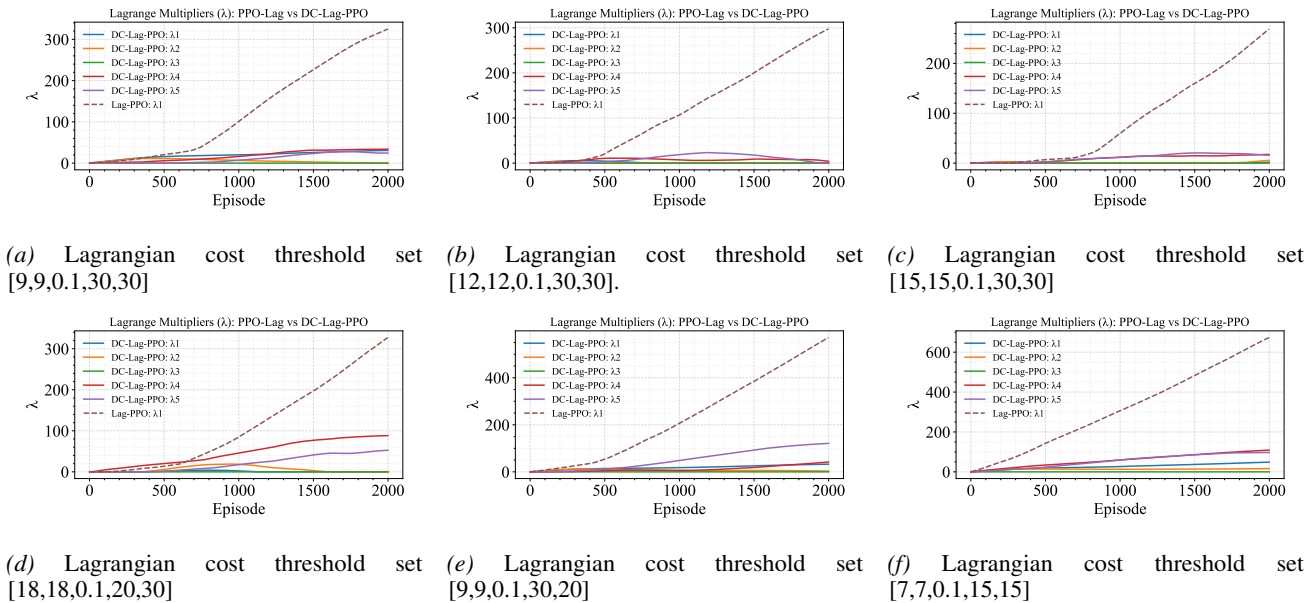

*(a)* Lagrangian cost threshold set [9,9,0.1,30,30]

*(b)* Lagrangian cost threshold set [12,12,0.1,30,30].

*(c)* Lagrangian cost threshold set [15,15,0.1,30,30]

*(d)* Lagrangian cost threshold set [18,18,0.1,20,30]

*(e)* Lagrangian cost threshold set [9,9,0.1,30,20]

*(f)* Lagrangian cost threshold set [7,7,0.1,15,15]

*Figure 19.* Lagrangian multiplier $\lambda$ learning curves.

### K.7. Pareto Fronts

Across all Pareto fronts shown in test cases in Fig. 20-27, which are obtained from different training runs, the DC-Lag-PPO fronts are typically shifted toward lower constraint values for the same (or nearby) reward levels, especially on the voltage metrics (violation ratio c1 and degree c2), indicating stronger constraint satisfaction without requiring large additional sacrifices in reward at the efficient frontier. This shift is visible in the reward-c1/c2 plots and also in the c1-c2, c1-c3, and c1–c4 pairings, where the dedicated-critic front envelopes or nearly envelopes the single-critic front.

A consistent pattern also emerges when examining the $J_c - d$ Pareto views across all parameter settings, where the points are the $J_c$ values of each constraint cost, and $d$, the black dotted line, is the target threshold of each constraint (see Figs. 28-35). For every constraint dimension, the DC-Lag-PPO solutions cluster tightly around, or slightly below, the threshold $d$, forming a compact Pareto front near the lower-left region. In contrast, Lag-PPO's $J_c$ values frequently lie well above the thresholds, and its Pareto front stretches diagonally upward, revealing strong trade-off tensions that arise from using a single shared multiplier. These $J_c - d$ relations provide direct evidence that DC-Lag-PPO not only finds better reward-constraint trade-offs but also fundamentally attains closer adherence to the prescribed limits on every constraint dimension.

**Knee regions and policy selection**   Several plots exhibit knee points on the DC-Lag-PPO front (most clearly on voltage and line-loading axes), where a small relaxation in reward yields a disproportionate drop in violations. These knees are natural operating points for deployment, offering strong safety gains at modest economic cost.

In the $J_c - d$ space, knee behavior manifests as sharp transitions where $J_c$ collapses rapidly once the policy enters the feasible region. These knees appear consistently in DC-Lag-PPO but rarely in Lag-PPO, further indicating that decomposed critics create a more controllable and interpretable constraint landscape.

**Voltage safety trade-offs (C1, C2)**   For voltage ratio and degree, DC-Lag-PPO consistently attains lower violations at comparable reward, producing a "left/downward" movement of the frontier relative to Lag-PPO. The paired-constraint views (C1 vs. C2) show a visibly tighter cloud and a frontier closer to the origin, suggesting better joint compliance.

The corresponding $J_c - d$ results reinforce this pattern: across all runs, DC-Lag-PPO keeps $J_c$ (C1) and $J_c$ (C2) very near the voltage thresholds, whereas Lag-PPO exhibits persistent overshoot. This aligns with the training curves and confirms that decomposed voltage critics effectively isolate and regulate the two voltage-related risks during testing.

**Line loading and degradation (C3, C4)** On line loading (C3) and battery degradation (C4), DC-Lag-PPO fronts again tend to sit below the Lag-PPO fronts for similar reward ranges, implying reduced thermal stress and milder throughput for batteries. The cross-constraint plots (e.g., C2-C3, C3-C4) also show that dedicated-critic solutions better balance these two operational risks simultaneously.

The $J_c - d$ plots show the same effect: DC-Lag-PPO pushes $J_c$(C3) and $J_c$(C4) tightly toward their respective thresholds, often forming extremely compact clusters around $d$, while Lag-PPO's distributions remain dispersed and systematically above the limits. This confirms that multi-critic Lagrangian updates mitigate cross-constraint interference that otherwise destabilizes single-multiplier methods.

**PV curtailment unfairness (C5)** In the reward C5 panels and the mixed-constraint views involving C5, the dedicated-critic frontier usually dominates or matches the single-critic frontier for a broad range, indicating more equitable PV curtailment at similar reward. That said, dispersion varies across runs, hinting that fairness may remain sensitive to training seed or tariff profiles.

The $J_c - d$ comparisons show that DC-Lag-PPO frequently holds $J_c$(C5) near the fairness threshold, whereas Lag-PPO often overshoots or displays large variance. This confirms that separating the fairness critic prevents it from being overshadowed by voltage/thermal constraints during optimization.

**Discussion** DC-Lag-PPO delivers stronger and more balanced constraint satisfaction than Lag-PPO, most prominently on voltage safety and with consistent advantages on line loading, degradation, and fairness. The additional $J_c - d$ evidence strengthens this conclusion: the dedicated-critic design yields systematically lower $J_c$ values tightly aligned with target thresholds, while the single-critic baseline exhibits structural difficulty simultaneously controlling heterogeneous constraints. The Pareto frontier shifts indicate that many safe operating points do not require drastic reward compromises once the policy is tuned to the knee region. Fairness (C5) gains are evident, though variability suggests room for additional stabilization (e.g., densifying episodic fairness or smoothing dual updates) in future runs.

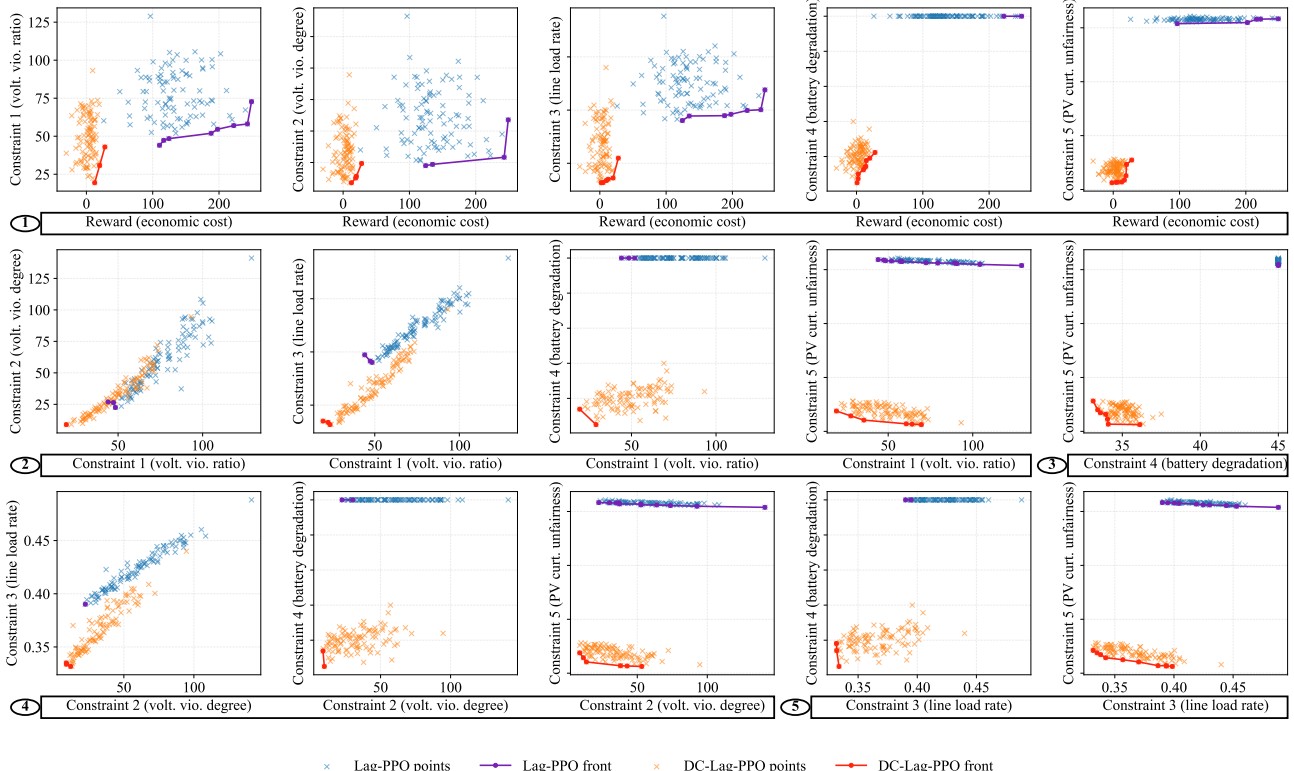

*Figure 20.* Pareto fronts from the test results on Lagrangian cost threshold set [9,9,0.1,30,30].

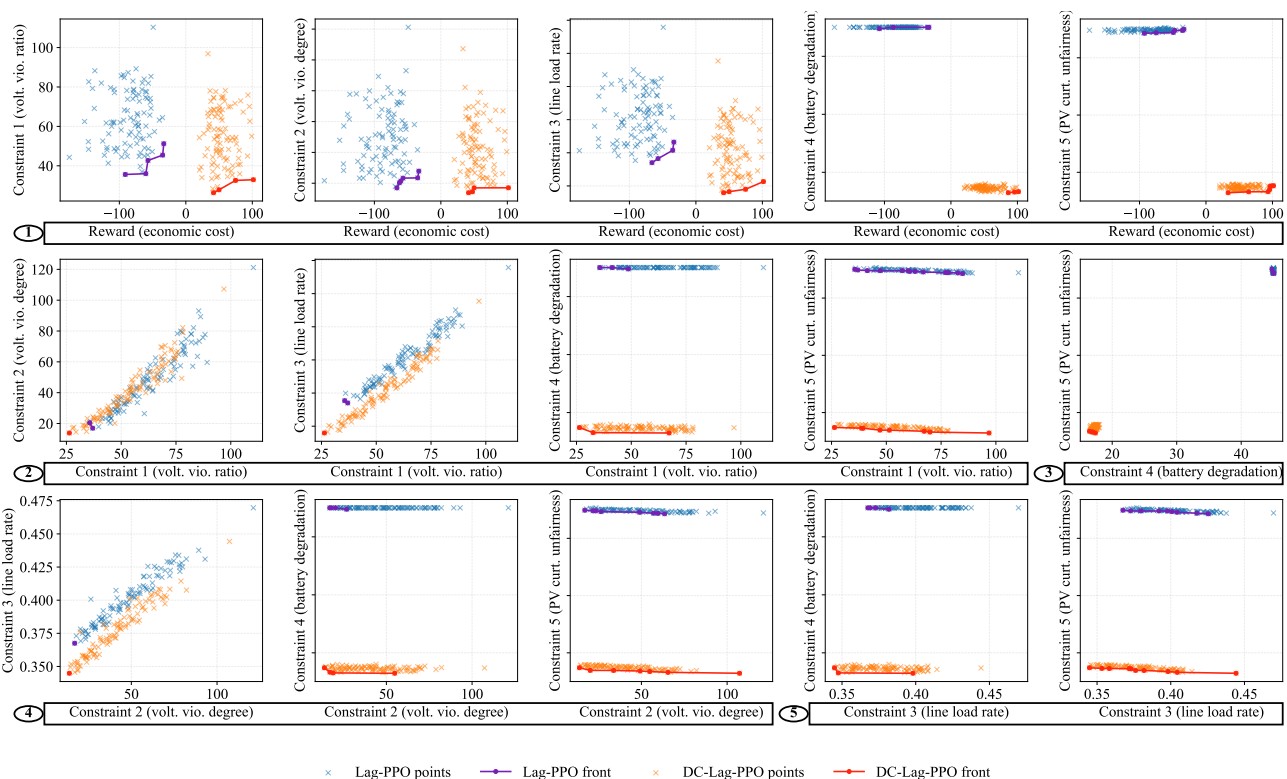

*Figure 21.* Pareto fronts from the test results on Lagrangian cost threshold set [12,12,0.1,30,30].

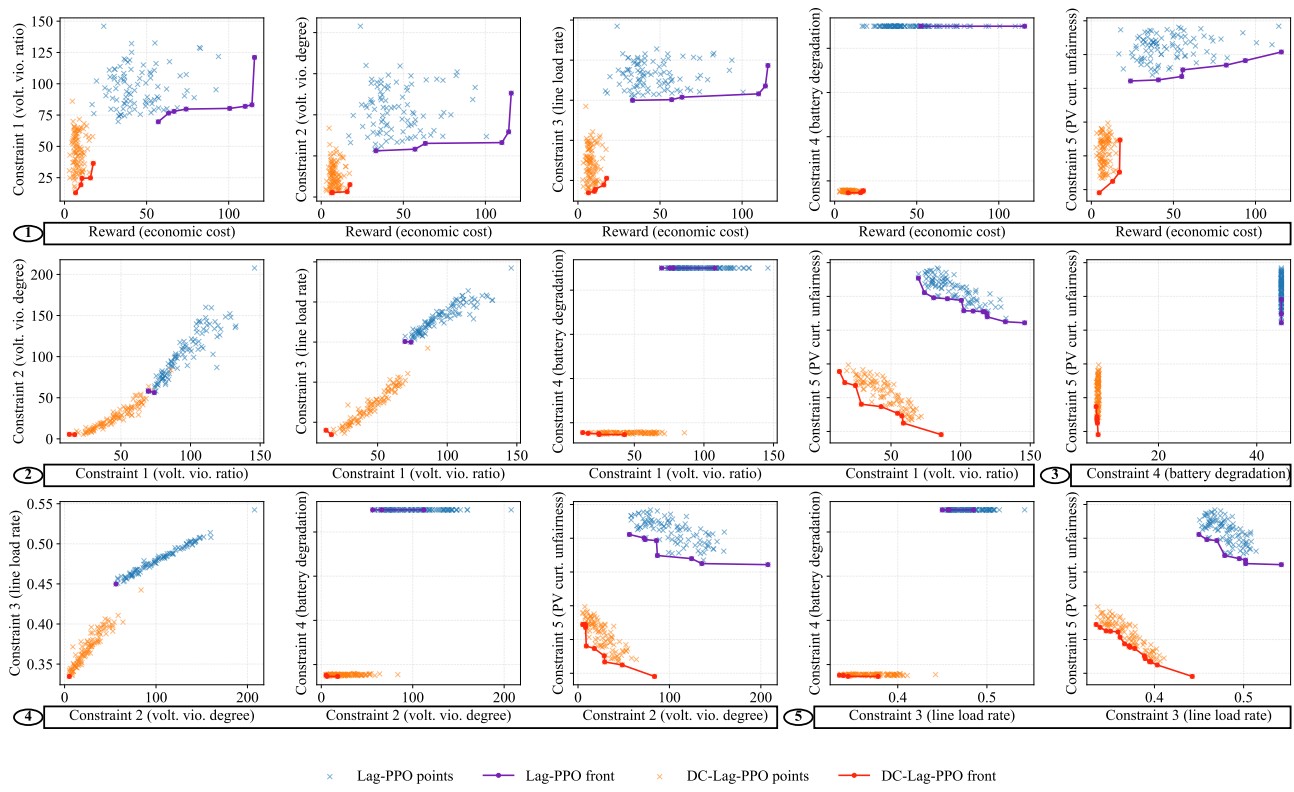

*Figure 22.* Pareto fronts from the test results on Lagrangian cost threshold set [15,15,0.1,20,30].

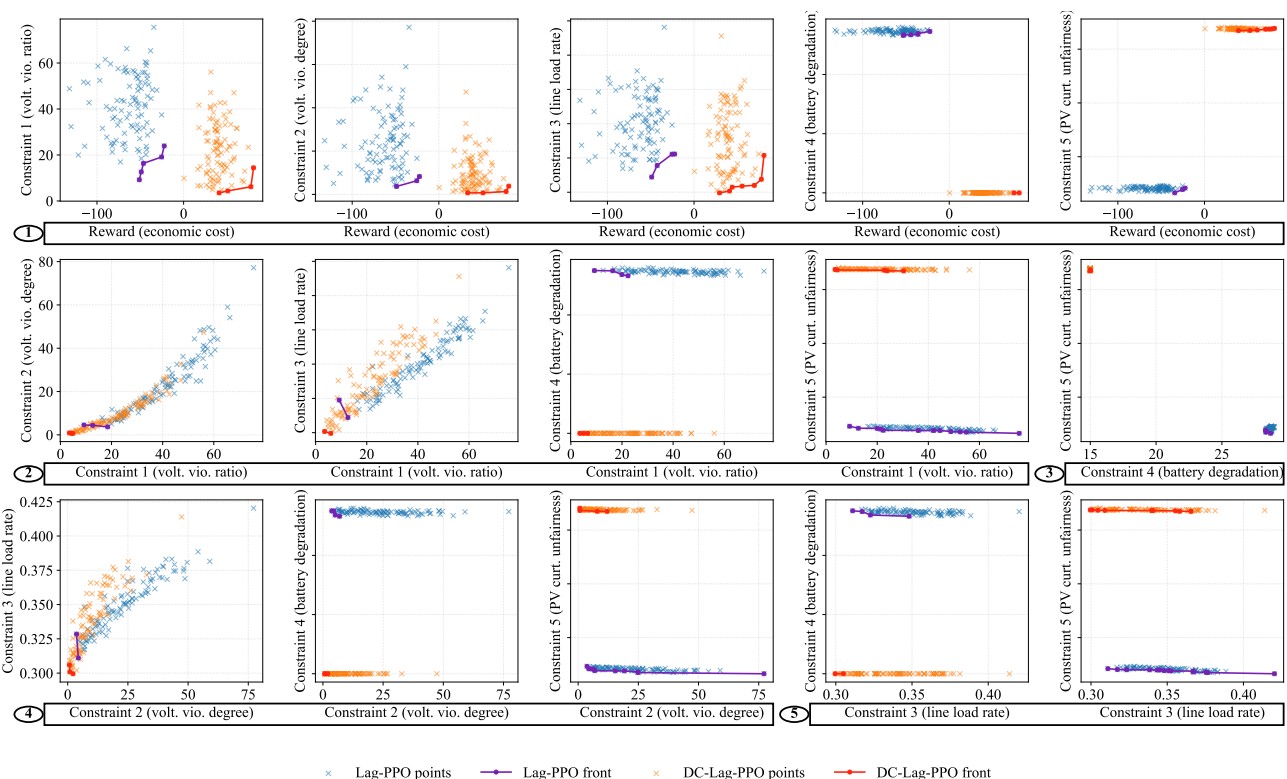

*Figure 23.* Pareto fronts from the test results on Lagrangian cost threshold set [18,18,0.1,20,30].

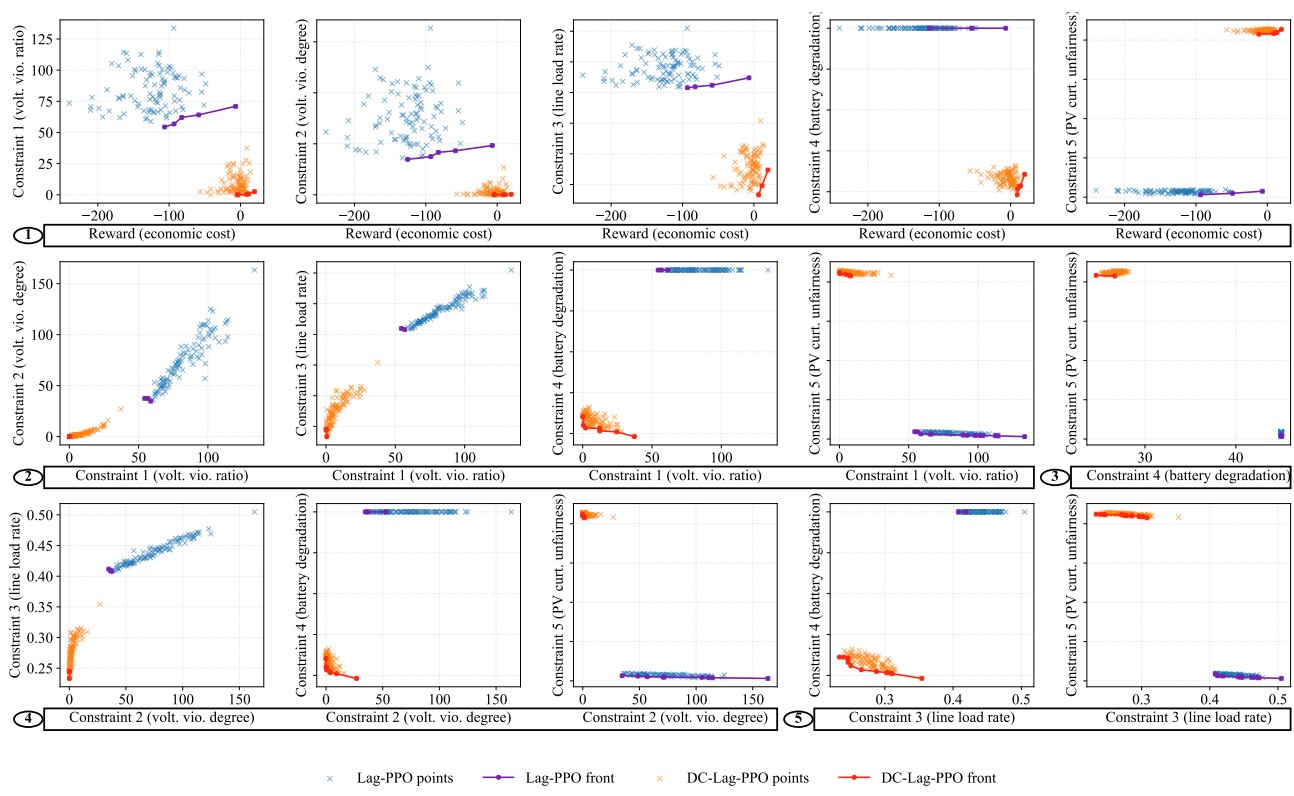

*Figure 24.* Pareto fronts from the test results on Lagrangian cost threshold set [9,9,0.1,30,20].

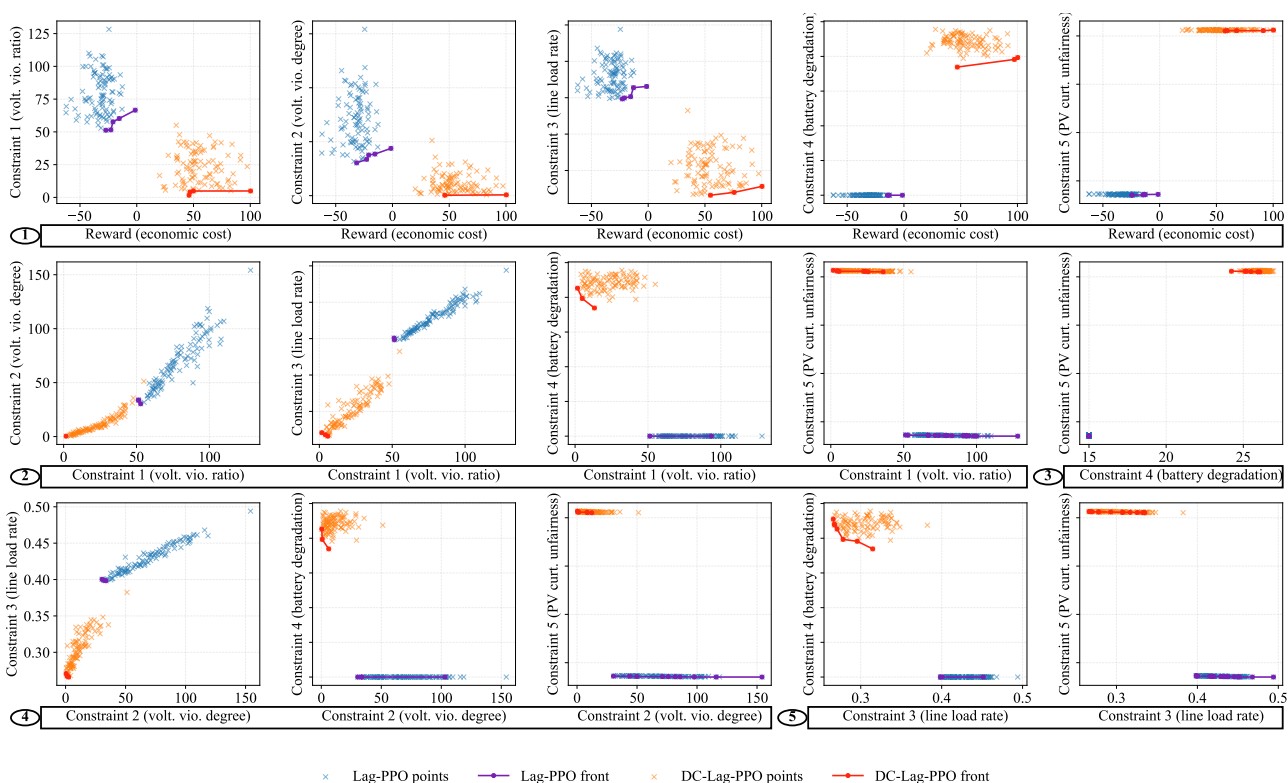

*Figure 25.* Pareto fronts from the test results on Lagrangian cost threshold set [12,12,0.1,30,20].

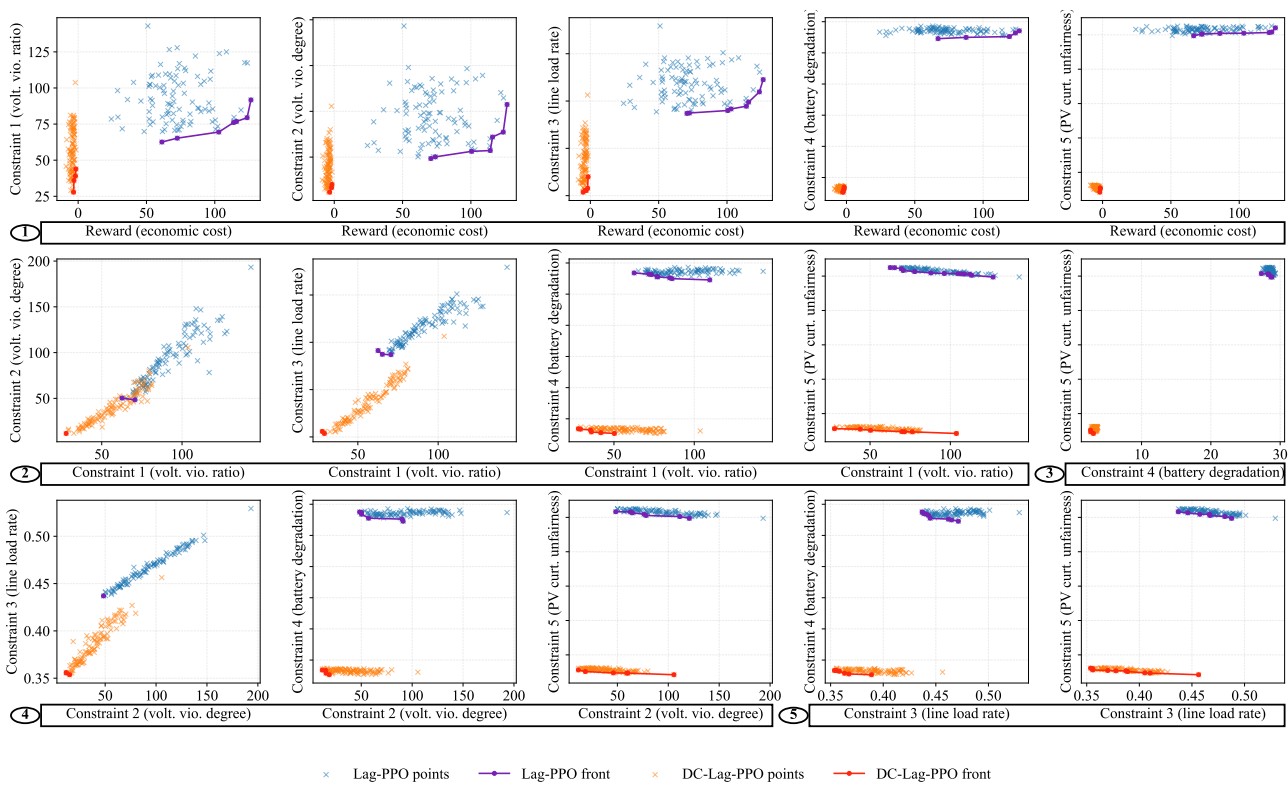

*Figure 26.* Pareto fronts from the test results on Lagrangian cost threshold set [9,9,0.1,20,20].

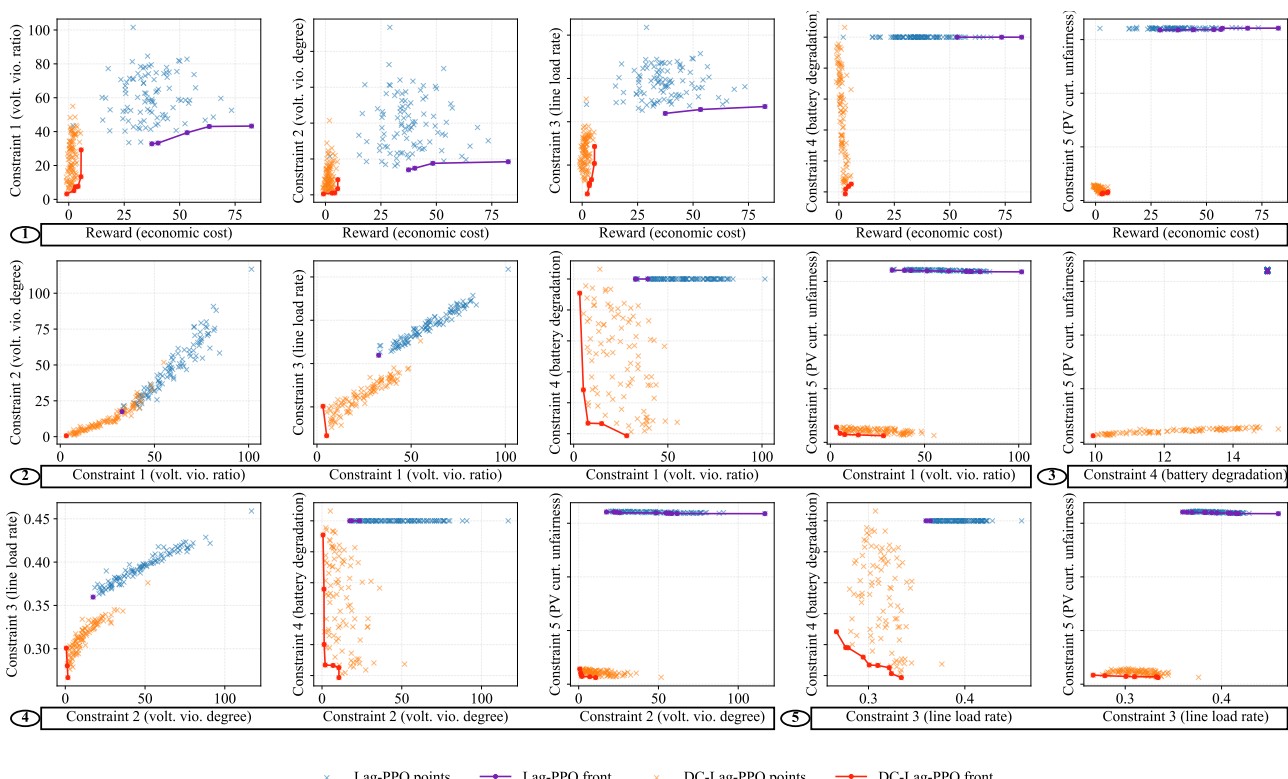

*Figure 27.* Pareto fronts from the test results on Lagrangian cost threshold set [7,7,0.1,15,15].

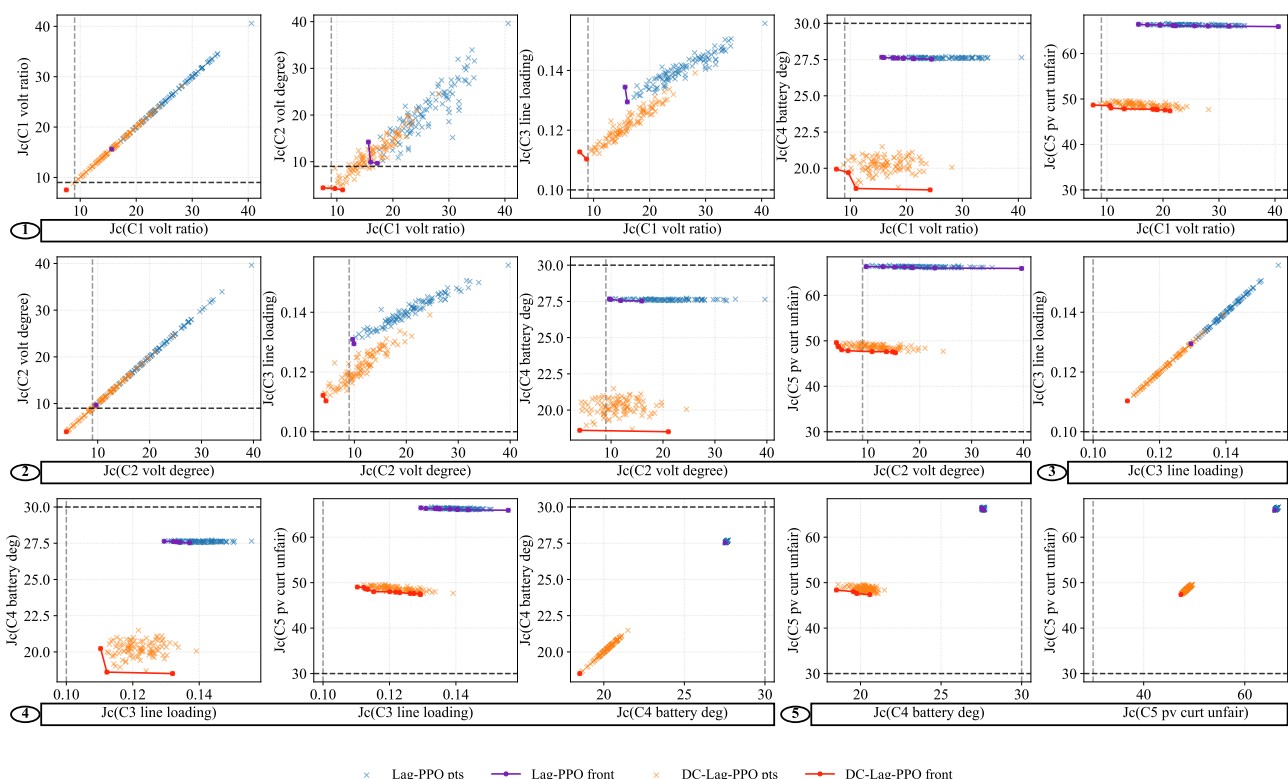

*Figure 28.* Pareto fronts from the test results of $J_c$ on Lagrangian cost threshold set [9,9,0.1,30,30], where the black dotted lines are the thresholds $d$.

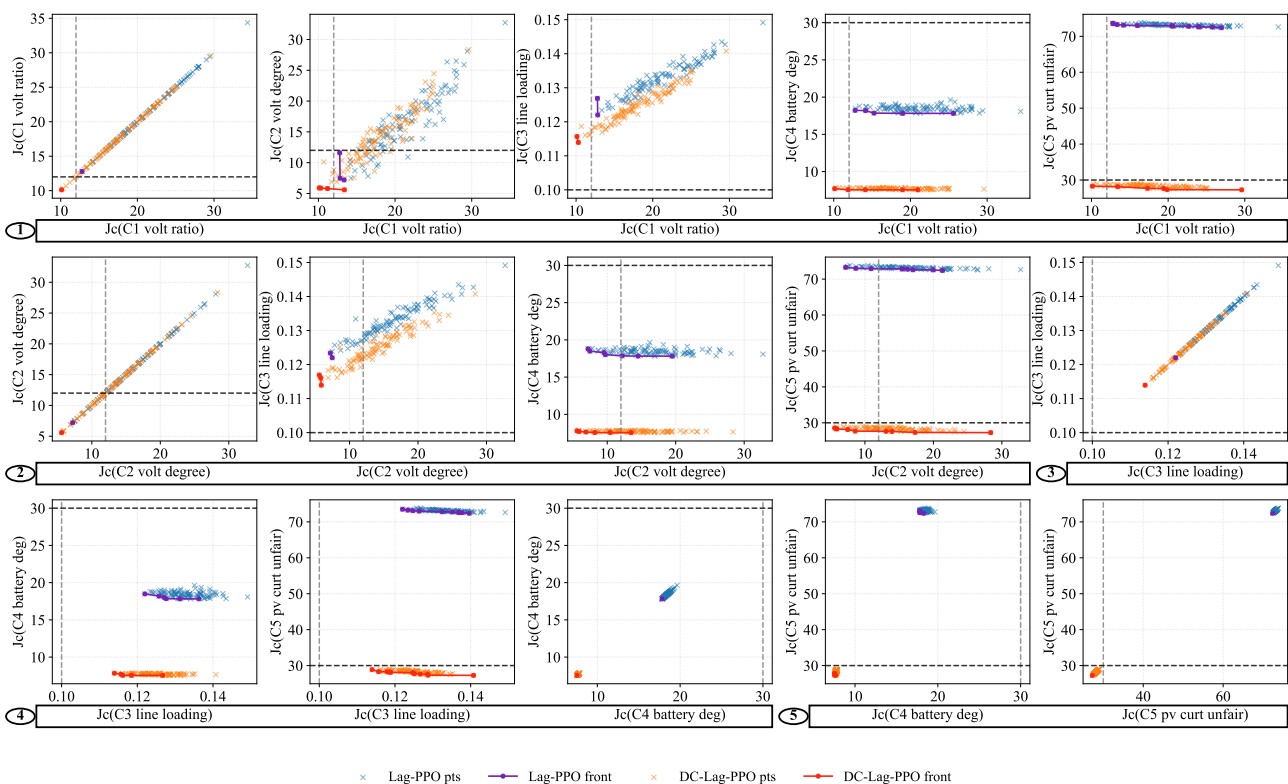

*Figure 29.* Pareto fronts from the test results of $J_c$ on Lagrangian cost threshold set [12,12,0.1,30,30], where the black dotted lines are the thresholds $d$.

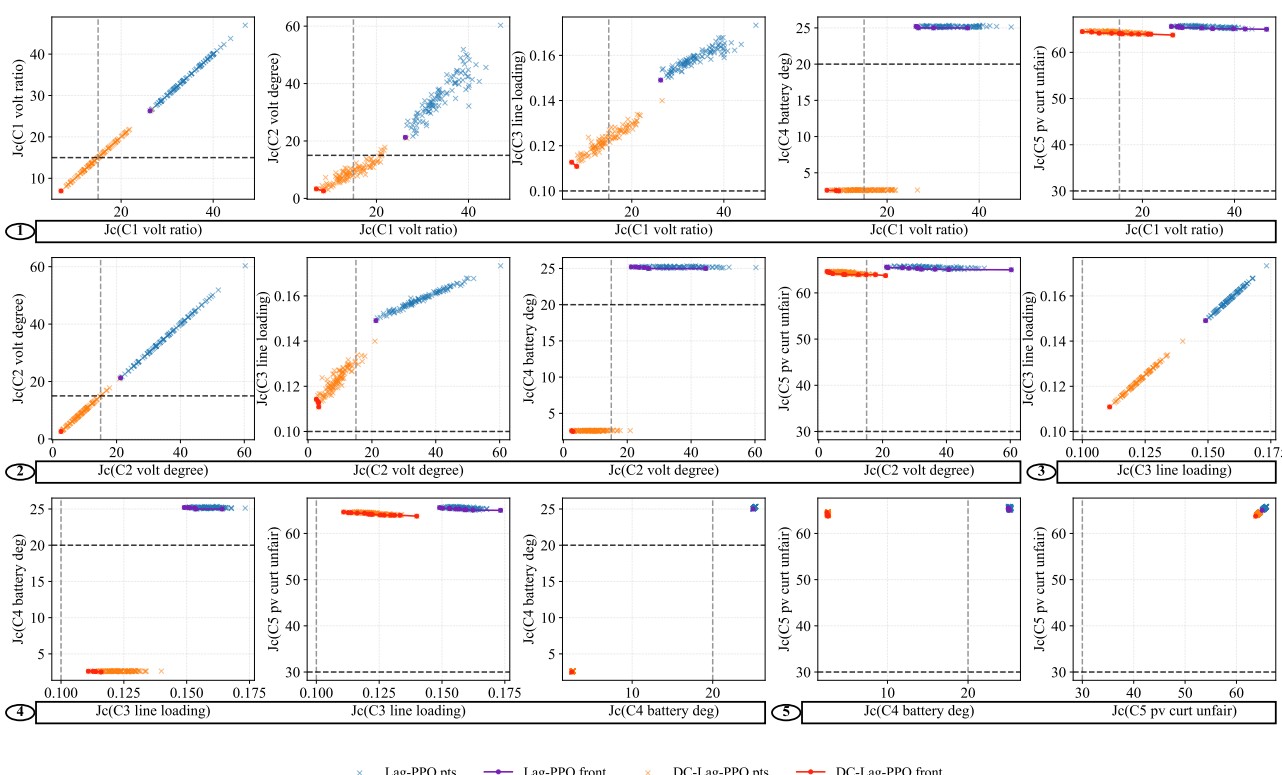

*Figure 30.* Pareto fronts from the test results of $J_c$ on Lagrangian cost threshold set [15,15,0.1,20,30], where the black dotted lines are the thresholds $d$.

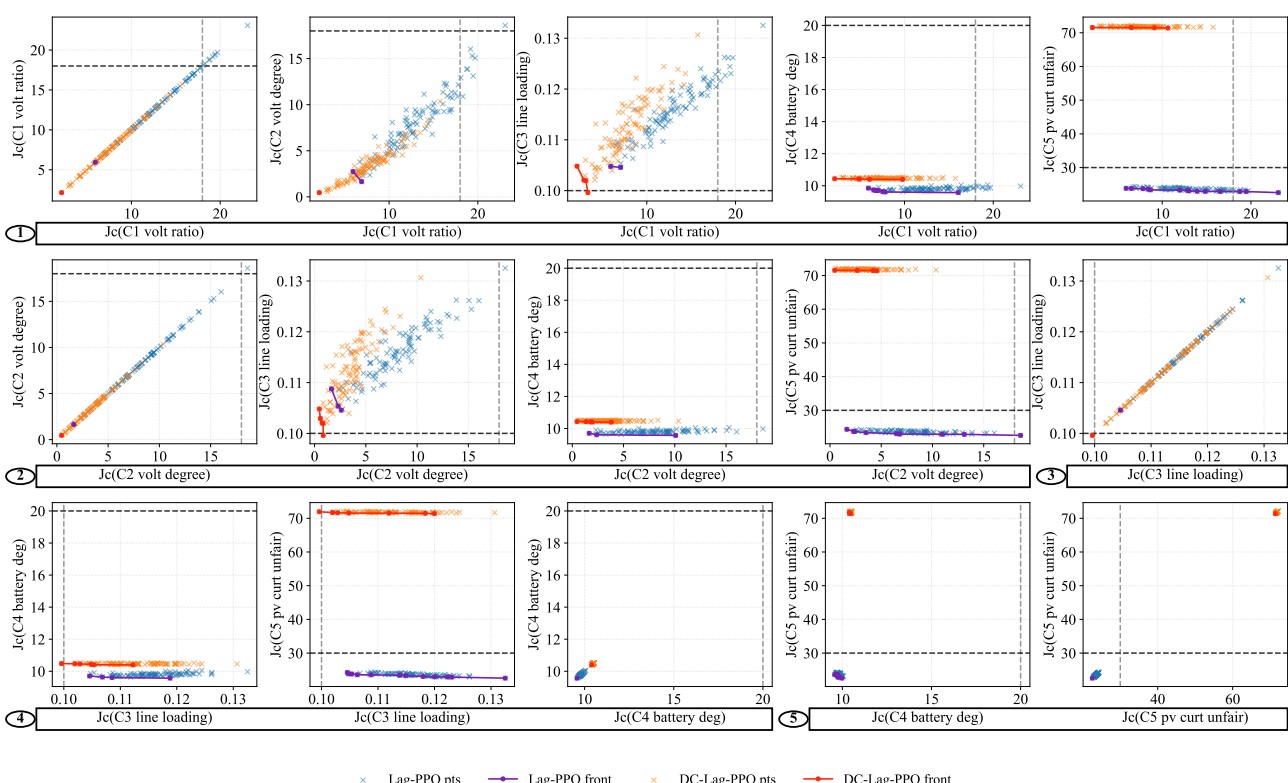

*Figure 31.* Pareto fronts from the test results of $J_c$ on Lagrangian cost threshold set [18,18,0.1,20,30], where the black dotted lines are the thresholds $d$.

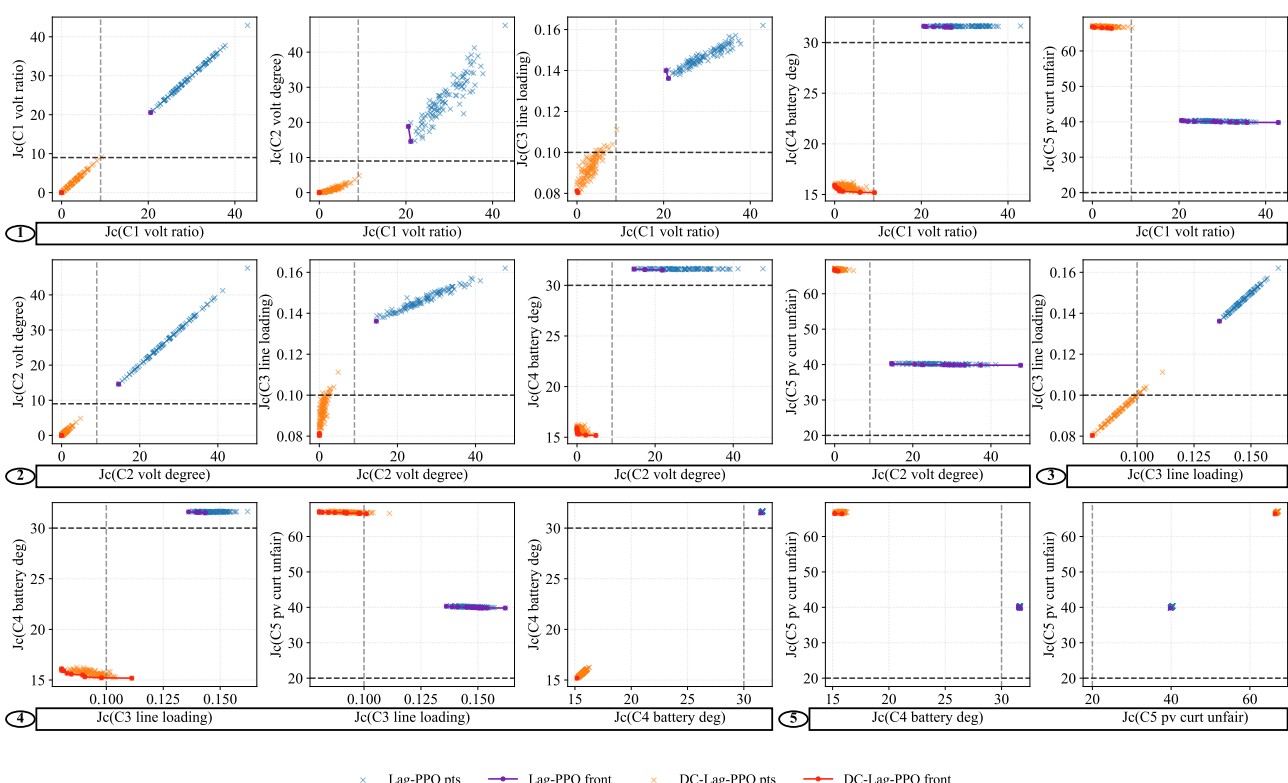

*Figure 32.* Pareto fronts from the test results of $J_c$ on Lagrangian cost threshold set [9,9,0.1,30,20], where the black dotted lines are the thresholds $d$.

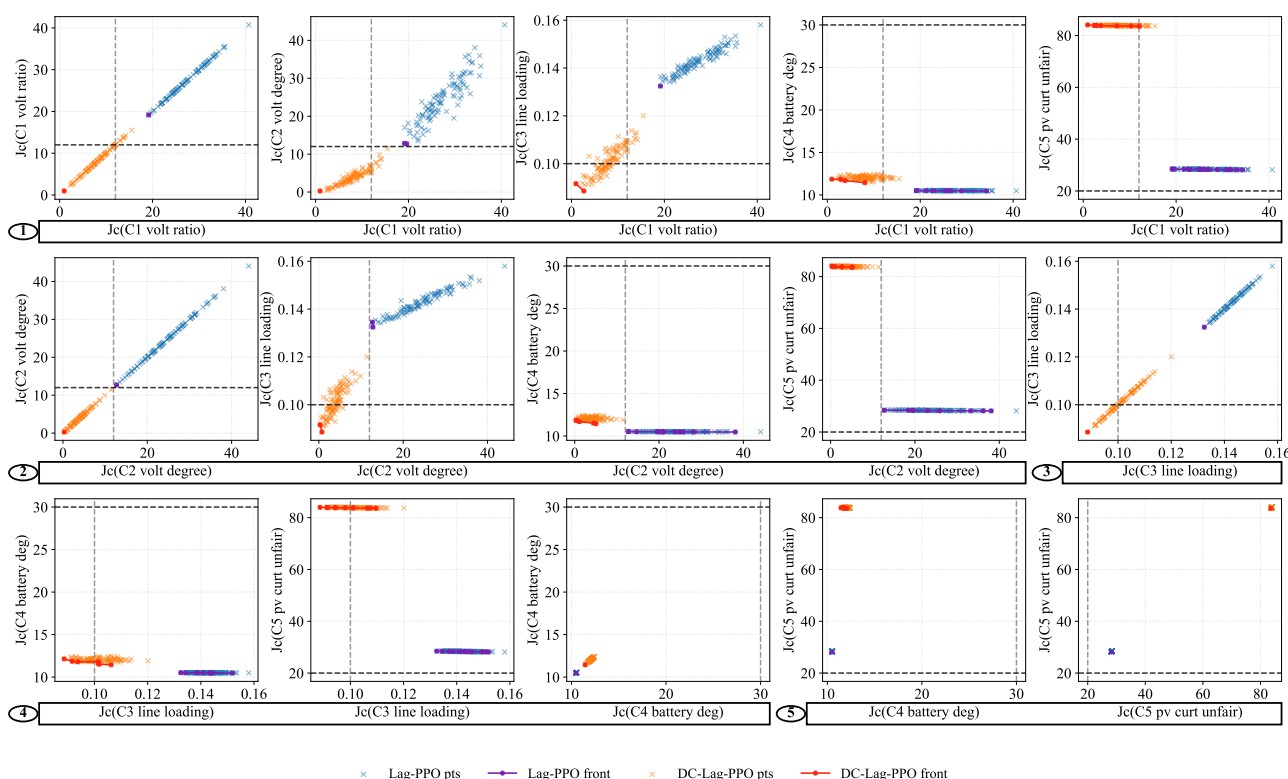

*Figure 33.* Pareto fronts from the test results of $J_c$ on Lagrangian cost threshold set [12,12,0.1,30,20], where the black dotted lines are the thresholds $d$.

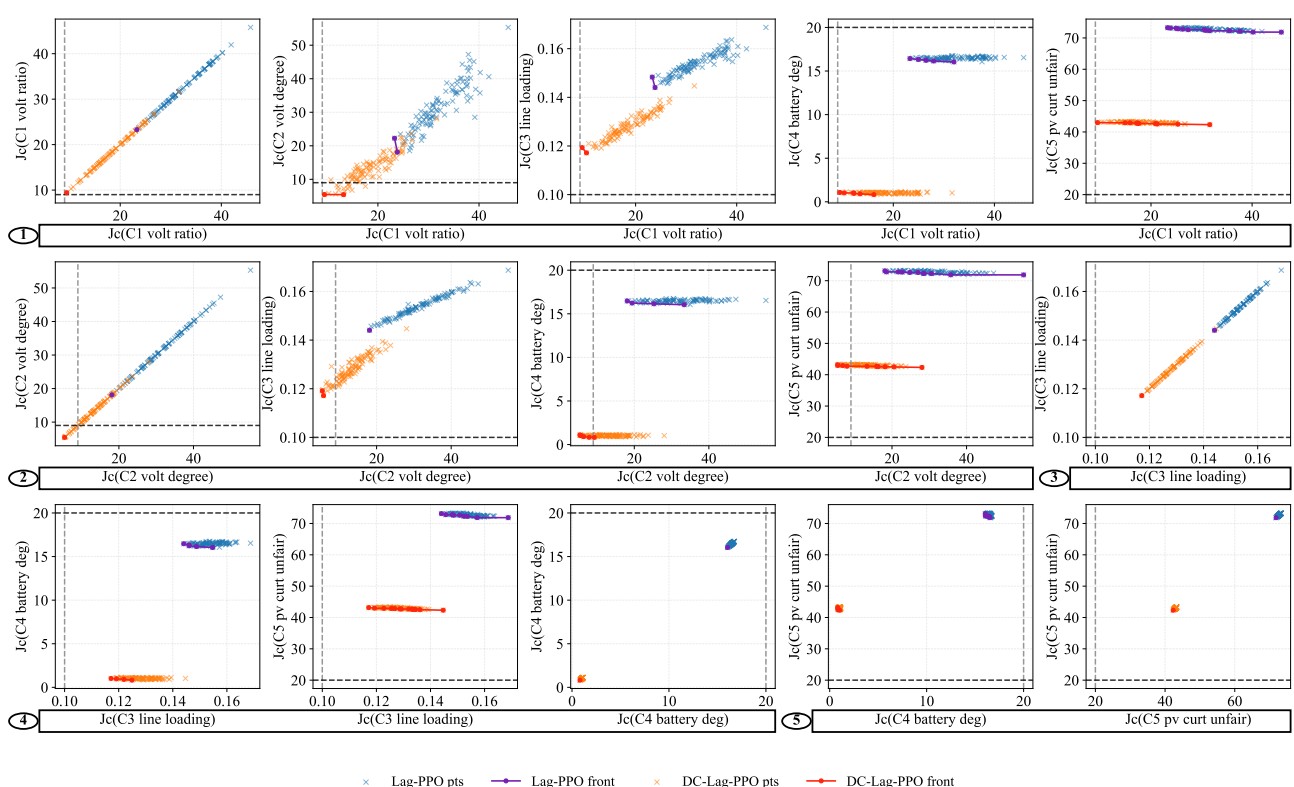

*Figure 34.* Pareto fronts from the test results of $J_c$ on Lagrangian cost threshold set [9,9,0.1,20,20], where the black dotted lines are the thresholds $d$.

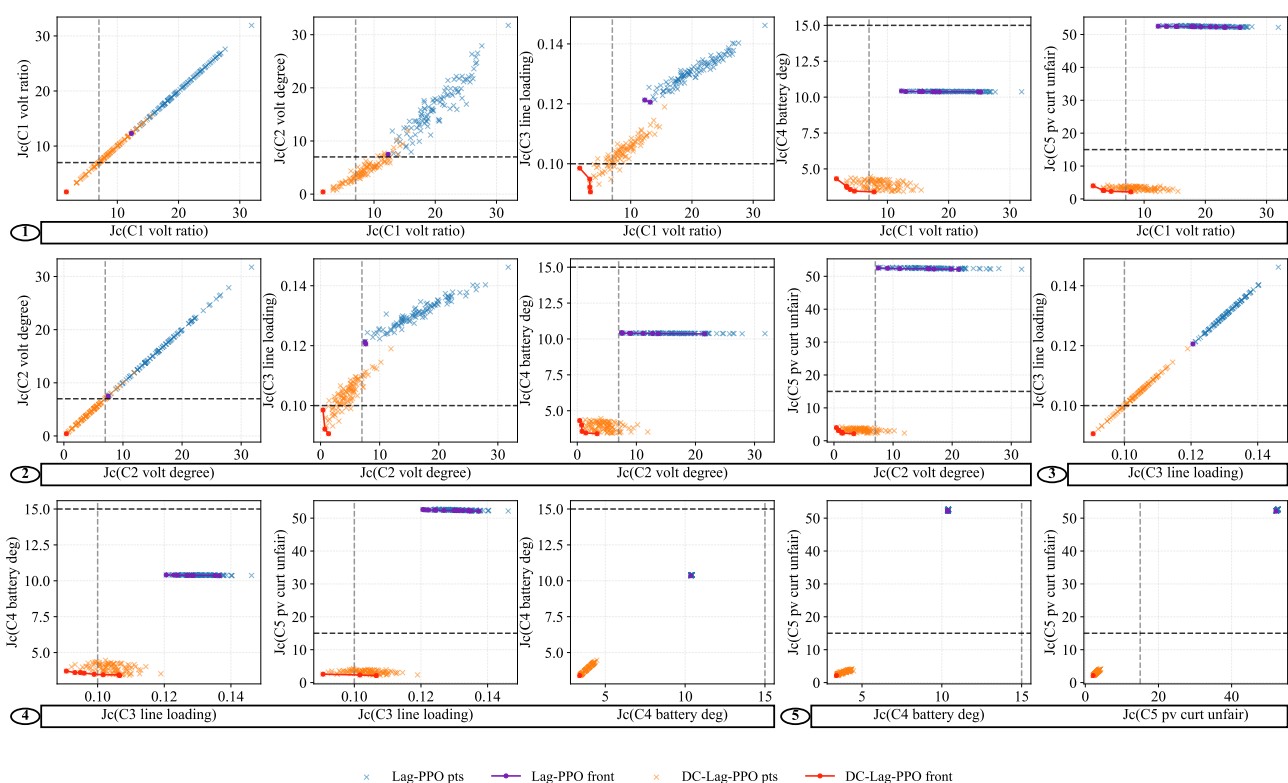

*Figure 35.* Pareto fronts from the test results of $J_c$ on Lagrangian cost threshold set [7,7,0.1,15,15], where the black dotted lines are the thresholds $d$.

# L. Experiment Details - Case 2

We extend our experiments to a more complex electric vehicle charging station (EVCS) problem to further evaluate how well the dedicated-critic approach scales and whether its benefits persist in realistic multi-constraint settings (Power system environment → observation dimension: 105, action dimension: 24; Electric Vehicle Charging → Observation dimension: 219; action dimension: 40). This environment models coordination problems, where the goal is to minimize charging costs while enforcing multiple charging related constraints, including voltage limits, EV battery degradation, and charging demand satisfaction. In contrast to the power system problem, this setting involves coordinating multiple EVCSs, each operating dozens of chargers and responding to highly stochastic and heterogeneous EV behaviours (arrival and departure times, charging demands, battery capacities, etc.). As a result, both the state and action spaces are substantially higher-dimensional, and the additional uncertainty introduced by EV dynamics makes the EVCS coordination task considerably more challenging than community battery scheduling.

## L.1. System Description

### L.1.1. PDN

The power distribution network (PDN) consists of a set of buses $\mathcal{N} = \{1, \ldots, N\}$ interconnected via distribution lines $\mathcal{L} \subseteq \mathcal{N} \times \mathcal{N}$. The system evolves over a discrete time horizon $\mathcal{T} = \{1, \ldots, T\}$. At each bus $i \in \mathcal{N}$ and time $t \in \mathcal{T}$, let $p_{i,t}$ and $v_{i,t}$ denote the net active power injection and voltage magnitude, respectively.

All nodes must satisfy the standard voltage bounds:

$$V^{\min} \leq v_{i,t} \leq V^{\max}, \qquad \forall i \in \mathcal{N}, \, t \in \mathcal{T}. \tag{69}$$

### L.1.2. EVCS DEPLOYMENT AND NEIGHBORHOOD STRUCTURE

The DSO manages a set of EV charging stations (EVCSs) $\mathcal{K} = \{1, \ldots, K\}$, each equipped with rooftop PV generation and a set of chargers $\mathcal{C}_k = \{1, \ldots, C_k\}$. EVCS $k$ is placed at exactly one PDN bus, represented by the binary deployment matrix $\mathbf{K} \in \{0,1\}^{N \times K}$:

$$K_{ik} = 1 \text{ if EVCS } k \text{ is located at bus } i, \quad K_{ik} = 0 \text{ otherwise,}$$

with the physical constraint that no two EVCSs colocate:

$$\sum_{k \in \mathcal{K}} K_{ik} \leq 1, \qquad \forall i \in \mathcal{N}.$$

For each bus $i$, its one-hop neighborhood is defined as

$$\mathcal{N}_i^{(1)} = \{j \in \mathcal{N} \setminus \{i\} \mid (i,j) \in \mathcal{L}\}.$$

If EVCS $k$ is located at bus $i$ ($K_{ik} = 1$), its accessible neighborhood is

$$\mathcal{N}_k^{(1)} = \mathcal{N}_i^{(1)},$$

representing all physically adjacent buses whose aggregate voltage and load information is available.

### L.1.3. EV CHARGING MODEL

Each charger serves EVs that arrive, park for a duration, and leave with a required energy level. Let $T_{c_k}^{\mathrm{arr}}$ and $T_{c_k}^{\mathrm{dep}}$ denote the arrival and departure times of EV $c_k$, and let $\mathrm{SoC}_{c_k}^{\mathrm{arr}}$ and $\mathrm{SoC}_{c_k}^{\mathrm{dep}}$ be the corresponding SoC levels. Their target SoC trajectory is modeled via linear interpolation:

$$\mathrm{SoC}_{c_k}^{\mathrm{target}}(t) = \mathrm{SoC}_{c_k}^{\mathrm{arr}} + \frac{t - T_{c_k}^{\mathrm{arr}}}{T_{c_k}^{\mathrm{dep}} - T_{c_k}^{\mathrm{arr}}} \left( \mathrm{SoC}_{c_k}^{\mathrm{dep}} - \mathrm{SoC}_{c_k}^{\mathrm{arr}} \right).$$

Each charger must satisfy:

$$0 \leq p_{c_k}^{\text{ch}}(t) \leq P_{c_k}^{\text{ch,max}}, \tag{70a}$$

$$0 \leq p_{c_k}^{\text{dis}}(t) \leq P_{c_k}^{\text{dis,max}}, \tag{70b}$$

$$p_{c_k}^{\text{ch}}(t)\, p_{c_k}^{\text{dis}}(t) = 0, \tag{70c}$$

$$\text{SoC}_{c_k}^{\text{min}} \leq \text{SoC}_{c_k}(t) \leq \text{SoC}_{c_k}^{\text{max}}, \tag{70d}$$

$$\Delta \text{SoC}_{c_k}(t) = \eta^{\text{ch}} p_{c_k}^{\text{ch}}(t) - \frac{1}{\eta^{\text{dis}}} p_{c_k}^{\text{dis}}(t). \tag{70e}$$

### L.1.4. VOLTAGE VIOLATION METRICS

In addition to total voltage violation, we also consider the number of voltage-violating buses:

$$f^{\text{NV}}(t) = \sum_{i \in \mathcal{N}} \mathbf{1}\{v_{i,t} \notin [V^{\text{min}}, V^{\text{max}}]\}.$$

This discrete stability metric counts the extent of widespread voltage deviations across the PDN.

### L.1.5. DEMAND SATISFACTION VIOLATION RATE

Let $N^{\text{EV}}$ denote the total number of EVs served during the horizon. Define a violation indicator for each EV:

$$\delta_c^{\text{DS}} = \mathbf{1}\Big\{ \text{SoC}_c(T_c^{\text{dep}}) < 0.95\, \text{SoC}_c^{\text{dep}} \Big\},$$

i.e., the EV fails to achieve at least $95\%$ of its desired departure SoC. The demand satisfaction violation rate is then

$$f_{\text{DS}}^{\text{VR}} = \frac{1}{N^{\text{EV}}} \sum_{c=1}^{N^{\text{EV}}} \delta_c^{\text{DS}}.$$

### L.1.6. OPERATIONAL COST FUNCTIONS

At each EVCS $k$, the DSO controls charging and discharging powers $\{p_{c_k}^{\text{ch}}(t), p_{c_k}^{\text{dis}}(t)\}_{c_k \in \mathcal{C}_k}$. The cost components are:

$$f_k^{\text{TD}}(t) = \begin{cases} \lambda_t^{\text{buy}}\, p_k^{\text{TD}}(t), & p_k^{\text{TD}}(t) > 0, \\ \lambda_t^{\text{sell}}\, p_k^{\text{TD}}(t), & \text{otherwise}, \end{cases}$$

$$f_k^{\text{DG}}(t) = \alpha_e \sum_{c_k \in \mathcal{C}_k} \left( [p_{c_k}^{\text{ch}}(t)]^2 + [p_{c_k}^{\text{dis}}(t)]^2 \right),$$

$$f^{\text{VT}}(t) = \sum_{i \in \mathcal{N}} \left( [v_{i,t} - V^{\text{max}}]^+ + [V^{\text{min}} - v_{i,t}]^+ \right),$$

$$f_k^{\text{DS}}(t) = \sum_{c_k \in \mathcal{C}_k} \left[ \text{SoC}_{c_k}^{\text{target}}(t) - \text{SoC}_{c_k}(t) \right]^+,$$

with traded power

$$p_k^{\text{TD}}(t) = \sum_{c_k \in \mathcal{C}_k} \left( p_{c_k}^{\text{ch}}(t) - p_{c_k}^{\text{dis}}(t) \right) - p_{k,t}^{\text{PV}}.$$

L.1.7. OBJECTIVE

The DSO seeks to minimize aggregated operational costs and violation penalties:

$$\min_{\mathbf{p}^{\text{ch}}, \mathbf{p}^{\text{dis}}} \sum_{t \in \mathcal{T}} \left[ \sum_{k \in \mathcal{K}} \left( \beta_1 f_k^{\text{TD}}(t) + \beta_2 f_k^{\text{DG}}(t) + \beta_3 f_k^{\text{DS}}(t) \right) + \beta_4 f^{\text{VT}}(t) + \beta_5 f^{\text{NV}}(t) \right] + \beta_6 f_{\text{DS}}^{\text{VR}}. \tag{71}$$

This objective captures energy costs, degradation, voltage safety, spatial extent of voltage violations, and global demand satisfaction reliability.

## L.2. CMDP Formulation with Dedicated-Critic Lagrangian RL

The EVCS coordination problem is modeled as a constrained Markov decision process (CMDP)

$$\left( \mathcal{S}, \mathcal{A}, P, r, \{c_i\}_{i=1}^m, \gamma, \{d_i\}_{i=1}^m \right),$$

where $\mathcal{S}$ and $\mathcal{A}$ denote the state and action spaces, $P(\cdot|s,a)$ the transition kernel, $r(s,a)$ the reward signal, $c_i(s,a)$ the cost signal for constraint $i$ with threshold $d_i$, and $\gamma \in (0,1)$ the discount factor.

For a policy $\pi_\theta(a|s)$, define the discounted returns:

$$J_r(\pi_\theta) = \mathbb{E}_\pi \left[ \sum_{t=0}^\infty \gamma^t \, r(s_t, a_t) \right], \tag{72}$$

$$J_{c_i}(\pi_\theta) = \mathbb{E}_\pi \left[ \sum_{t=0}^\infty \gamma^t \, c_i(s_t, a_t) \right], \quad i = 1, \dots, m. \tag{73}$$

The CMDP objective is

$$\max_\theta \ J_r(\pi_\theta) \qquad \text{s.t.} \qquad J_{c_i}(\pi_\theta) \le d_i, \ \ i = 1, \dots, m. \tag{74}$$

**Reward and cost signals derived from the system model.** Let the instantaneous cost components from the system description be:

- $f_t^{\text{VT}}$: total voltage violation magnitude,

- $f_t^{\text{NV}}$: number of voltage-violating buses,

- $f_t^{\text{LL}}$: line-loading stress,

- $f_t^{\text{DG}}$: battery degradation,

- $f_t^{\text{TD}}$: energy trading cost,

- $f_t^{\text{DS}}$: per-step EV dissatisfaction,

- $f_{\text{DS}}^{\text{VR}}$: demand satisfaction violation rate (episodic).

A practical reward–cost decomposition aligning with operational goals is:

$$r(s_t, a_t) = -\left( \alpha_{\text{TD}} \, f_t^{\text{TD}} + \alpha_{\text{DG}} \, f_t^{\text{DG}} + \alpha_{\text{DS}} \, f_t^{\text{DS}} \right), \tag{75}$$

$$c_1(s_t, a_t) = f_t^{\text{VT}}, \qquad \text{(voltage violation magnitude)} \tag{76}$$

$$c_2(s_t, a_t) = f_t^{\text{NV}}, \qquad \text{(number of violating buses)} \tag{77}$$

$$c_3(s_t, a_t) = f_t^{\text{LL}}, \qquad \text{(line loading)} \tag{78}$$

$$c_4(s_t, a_t) = f_t^{\text{DG}}, \qquad \text{(battery degradation)} \tag{79}$$

Additionally, the demand-satisfaction violation rate $f_{\text{DS}}^{\text{VR}}$ is an episodic cost:

$$C_5(\tau) \triangleq f_{\text{DS}}^{\text{VR}}, \qquad \mathbb{E}_\pi[C_5(\tau)] \le d_5, \tag{80}$$

where $\tau$ denotes a full episode. If preferred, $f_{\text{DS}}^{\text{VR}}$ can be distributed as a per-step cost $c_5(s_t, a_t)$ such that its discounted sum recovers the same episodic value.

**Lagrangian relaxation with per-constraint critics.** Introduce dual multipliers $\lambda = (\lambda_1, \ldots, \lambda_m) \succeq 0$ and form the Lagrangian:

$$\mathcal{L}(\theta, \lambda) = J_r(\pi_\theta) - \sum_{i=1}^m \lambda_i \left( J_{c_i}(\pi_\theta) - d_i \right). \tag{81}$$

Primal–dual updates follow:

$$\nabla_\theta \mathcal{L}(\theta, \lambda) = \nabla_\theta J_r(\pi_\theta) - \sum_{i=1}^m \lambda_i \nabla_\theta J_{c_i}(\pi_\theta), \tag{82}$$

$$\lambda_i \leftarrow \Pi_{[0, \lambda_{\max}]} \left( \lambda_i + \beta \left( \widehat{J}_{c_i} - d_i \right) \right), \tag{83}$$

with $\Pi$ denoting projection for stability.

**Value functions and signal-specific advantages.** For each signal $x \in \{r, c_1, \ldots, c_m\}$, define:

$$Q_\pi^x(s, a) = \mathbb{E}_\pi \left[ \sum_{t=0}^\infty \gamma^t x(s_t, a_t) \mid s_0 = s, a_0 = a \right], \tag{84}$$

$$V_\pi^x(s) = \mathbb{E}_{a \sim \pi}[Q_\pi^x(s, a)], \tag{85}$$

$$A_\pi^x(s, a) = Q_\pi^x(s, a) - V_\pi^x(s). \tag{86}$$

The actor gradient becomes:

$$\nabla_\theta \mathcal{L}(\theta, \lambda) = \mathbb{E}_\pi \left[ \nabla_\theta \log \pi_\theta(a|s) \left( A_\pi^r(s, a) - \sum_{i=1}^m \lambda_i A_\pi^{c_i}(s, a) \right) \right]. \tag{87}$$

**Dedicated critics for each signal.** Each signal $x \in \{r, c_1, \ldots, c_m\}$ is assigned a separate critic $Q_{\omega_x}$:

$$\delta_t^x = x_t + \gamma Q_{\omega_x}(s_{t+1}, a_{t+1}) - Q_{\omega_x}(s_t, a_t), \tag{88}$$

and the critic minimizes $\mathbb{E}[(\delta_t^x)^2]$. Advantage estimates (e.g., GAE) are computed per signal and combined through the Lagrangian structure.

**PPO-style actor update.** Let $r_t(\theta) = \pi_\theta(a_t|s_t)/\pi_{\theta_{\text{old}}}(a_t|s_t)$ and

$$\widetilde{A}_t = A_t^r - \sum_{i=1}^m \lambda_i A_t^{c_i}.$$

The clipped surrogate is

$$\mathcal{J}_{\text{PPO}}(\theta) = \mathbb{E} \left[ \min \left( r_t(\theta) \widetilde{A}_t, \; \text{clip}(r_t(\theta), 1 - \epsilon, 1 + \epsilon) \widetilde{A}_t \right) \right] + \eta \, \mathbb{E}[\mathcal{H}(\pi_\theta(\cdot|s_t))], \tag{89}$$

where $\mathcal{H}$ denotes policy entropy.

**Instantiated constraints for this problem.** With the reward and cost mapping above, we typically have

$$m = 5,$$

corresponding to:

- $c_1$: voltage violation magnitude $f_t^{\mathrm{VT}}$,

- $c_2$: violating node count $f_t^{\mathrm{NV}}$,

- $c_3$: line loading $f_t^{\mathrm{LL}}$,

- $c_4$: battery degradation $f_t^{\mathrm{DG}}$,

- $c_5$: demand-satisfaction violation rate $f_{\mathrm{DS}}^{\mathrm{VR}}$ (episodic or densified).

The corresponding critics are:

$$Q_{\omega_r}, \ Q_{\omega_{c_1}}, Q_{\omega_{c_2}}, Q_{\omega_{c_3}}, Q_{\omega_{c_4}}, Q_{\omega_{c_5}}.$$

The combined advantage is:

$$\widetilde{A}_t = A_t^r - \lambda_1 A_t^{c_1} - \lambda_2 A_t^{c_2} - \lambda_3 A_t^{c_3} - \lambda_4 A_t^{c_4} - \lambda_5 A_t^{c_5}.$$

Dual multipliers update via equation 83.

**Practical considerations.** To stabilize learning: (i) use target networks or Polyak averaging for each critic; (ii) normalize each advantage $A_t^{c_i}$ before aggregation; (iii) constrain multipliers via projection or softplus parameterization; (iv) for episodic costs, update multipliers once per episode; stepwise costs update per batch.

### L.3. Experimental Parameters

**Symbols.** $\phi_t^{\mathrm{buy}}, \phi_t^{\mathrm{sell}}$: buy/sell electricity prices; $[x]^+ = \max(x, 0)$; $\mathbf{1}\{\cdot\}$: indicator; $v_{i,t}$: voltage at bus $i$; $V^{\min}, V^{\max}$: voltage bounds; $\ell_{ij,t}$: loading of line $(i, j)$; $\tau^{\mathrm{line}}$: overload threshold; $p_{c_k,t}^{\mathrm{ch}}, p_{c_k,t}^{\mathrm{dis}}$: charging/discharging powers; $\gamma_{i,t}$: PV curtailment ratio; $\Delta t$: step duration (5 min); $N$: number of buses; $|\mathcal{C}_k|$: chargers at EVCS $k$.

### L.4. Two-Tiered Statistics

See Table 4 for details. This table summarizes the two-tiered evaluation statistics, reporting the mean and standard deviation across three independent training runs. Overall, DC-Lag-PPO consistently outperforms Lag-PPO across all constraint metrics while also achieving better economic performance, confirming that decomposing the critics alleviates the interference between heterogeneous constraints and stabilizes the dual updates.

DC-Lag-PPO reduces the economic cost substantially, outperforming Lag-PPO by 19.3 units on average, despite the inherently high variance of cost signals. The improvement of -109% (negative because higher reward is better) indicates that DC-Lag-PPO not only avoids the reward degradation often observed in constrained RL, but actually discovers more cost-efficient charging strategies while still satisfying the operational constraints.

For both voltage violation ratio (c1) and degree (c2), DC-Lag-PPO achieves consistent and significant reductions: -23.14 in violation ratio (+16.9% improvement), -44.67 in violation degree (+22.8% improvement). These gains validate the core motivation of the dedicated-critic design: each voltage-related critic captures its own risk landscape, preventing the single-critic baseline from being dominated by a few severe constraints. The larger improvement on violation degree (c2) suggests that DC-Lag-PPO not only reduces the frequency of violations but also suppresses their severity, producing safer voltage profiles across the entire PDN.

Battery throughput and degradation drop from 41.43 to 25.62, yielding the largest improvement among all instantaneous constraints (+38.2%). This indicates that DC-Lag-PPO is better at distributing the charging/discharging workload across EV chargers, avoiding the overuse of individual chargers or time windows. The result also aligns with DC-Lag-PPO's smoother dual updates, which prevent oscillatory behaviors commonly seen in single-multiplier methods.

*Table 3.* Key hyperparameters, reward structure, and CMDP constraints for the EVCS coordination case study.

| Category | Term / Parameter | Value | Definition / Description |
|---|---|---|---|
| **Reward & CMDP constraints** | | | |
| Reward $r_t$ | $-\big(\alpha_{\mathrm{TD}} f_t^{\mathrm{TD}} + \alpha_{\mathrm{DG}} f_t^{\mathrm{DG}} + \alpha_{\mathrm{DS}} f_t^{\mathrm{DS}}\big)$ | – | Trading, degradation, and dissatisfaction penalties |
| Constraint $c_1$ (Voltage magnitude violation) | $f_t^{\mathrm{VT}} = \sum_i \big([v_{i,t} - V^{\max}]^+ + [V^{\min} - v_{i,t}]^+\big)$ | – | Voltage violation magnitude across buses |
| Constraint $c_2$ (Count of violating buses) | $f_t^{\mathrm{NV}} = \sum_i \mathbf{1}\{v_{i,t} \notin [V^{\min}, V^{\max}]\}$ | $[0, N]$ | Number of buses violating voltage limits |
| Constraint $c_3$ (Line loading) | $f_t^{\mathrm{LL}} = \sum_{(i,j)} [\ell_{ij,t} - \tau^{\mathrm{line}}]^+$ | – | Thermal overload above permissible threshold |
| Constraint $c_4$ (Battery degradation) | $f_t^{\mathrm{DG}} = \alpha_e \sum_{c_k} ([p_{c_k,t}^{\mathrm{ch}}]^2 + [p_{c_k,t}^{\mathrm{dis}}]^2)$ | – | Throughput-based quadratic degradation |
| Constraint $c_5$ (Demand satisfaction violation rate) | $f_{\mathrm{DS}}^{\mathrm{VR}}$ | $[0, 1]$ | Fraction of EVs leaving with SoC $<$ $0.95\,\mathrm{SoC}^{\mathrm{dep}}$ |
| *Lag-PPO baseline constraint* | $\sum_{i=1}^{5} c_i$ | – | Single aggregated constraint in standard Lag-PPO |
| **General training hyperparameters** | | | |
| Learning rate | – | $3 \times 10^{-4}$ | For both actor and critics |
| PPO clip $\epsilon$ | – | 0.2 | Ratio clipping: $[1 - \epsilon, 1 + \epsilon]$ |
| Target KL | – | 0.015 | Early stopping threshold |
| Value loss coefficient | – | 0.5 | Weight for critic loss |
| Entropy coefficient | – | 0.0 | Entropy regularization |
| Gradient norm clip | – | 0.5 | Global clipping limit |
| Hidden sizes | – | (256, 256) | MLP layers for all networks |
| Init log-std | – | $-0.5$ | Gaussian policy initialization |
| Discount $\gamma$ / GAE $\lambda$ | – | 0.99 / 0.95 | Returns and advantage estimation |
| Dual learning rate | – | $5 \times 10^{-3}$ | Step size for multiplier update |
| $\lambda$ init / max | – | $0.0 / 10^4$ | Multiplier projection range |
| **Training schedule & environment** | | | |
| Episodes | – | 2000 | Total PPO training episodes |
| Steps per episode | – | 288 | One full day (5 min resolution) |
| Environment step | – | 5 min | Sampling interval |

The dissatisfaction volume is reduced from 41.07 to 36.72 (+12.2%), demonstrating that DC-Lag-PPO better supports EV users' charging requirements. The relatively low variance of DC-Lag-PPO also implies improved training stability and more consistent performance across runs.

Among all constraints, the dedicated-critic method yields one of the most significant improvements on dissatisfaction number from 53.13 to 35.44 (+33.9%). This confirms that DC-Lag-PPO not only reduces instantaneous dissatisfaction but also lowers the probability of EVs failing to meet their departure SoC requirement, complementing the improvement in dissatisfaction volume.

*Table 4.* Two-tiered test statistics, where across-run mean $\pm$ across-run std; The higher reward is better, while the lower constraints are better. $\Delta = (\text{DC-Lag-PPO} - \text{Lag-PPO})$. Positive improvement % is computed as $(\text{Lag-PPO} - \text{DC-Lag-PPO})/\text{Lag-PPO} \times 100\%$, except reward where the value is negated because higher is better.

| Metric | Lag-PPO (n=3) | DC-Lag-PPO (n=3) | $\Delta$ | Improvement % |
|---|---|---|---|---|
| Economic cost (reward) | $17.66 \pm 40.88$ | $-1.61 \pm 10.02$ | **-19.27** | **-109.18%** |
| Volt. vio. ratio (c1) | $136.67 \pm 6.61$ | $113.53 \pm 10.72$ | **-23.14** | **+16.93%** |
| Volt. vio. degree (c2) | $196.41 \pm 14.06$ | $151.74 \pm 47.41$ | **-44.67** | **+22.75%** |
| Battery deg. (c3) | $41.43 \pm 4.86$ | $25.62 \pm 7.81$ | **-15.82** | **+38.18%** |
| Dissat. vol. (c3) | $41.07 \pm 9.01$ | $36.72 \pm 3.16$ | **-9.89** | **+12.17%** |
| Dissat. num. (c5) | $53.13 \pm 8.25$ | $35.44 \pm 5.22$ | **-11.68** | **+33.87%** |

## L.5. Training Curves

See Figure 36 and Figure 37 for detail. It can be seen that DC-Lag-PPO learns faster, stabilizes earlier, satisfies constraints better, and avoids the late-stage divergence exhibited by Lag-PPO.

First, DC-Lag-PPO achieves lower and more stable economic cost, whereas Lag-PPO exhibits large oscillations and late-stage degradation. Second, for all constraint metrics, including voltage ratio/degree, line loading, battery degradation, and demand dissatisfaction, DC-Lag-PPO maintains lower violation levels throughout training, with clearly reduced variance. In contrast, Lag-PPO's curves drift upward or fluctuate heavily, indicating unstable constraint handling under the aggregated-critic formulation.

The $J_c$ curves further confirm this advantage: every DC-Lag-PPO constraint return steadily moves toward its target threshold $d$ crosses it, and eventually stabilizes near or below the limit. Lag-PPO, however, shows a single aggregated $J_c$ that quickly rises above the feasibility threshold and fails to recover, demonstrating an inability to control multiple constraints simultaneously.

In summary, DC-Lag-PPO learns faster, stabilizes earlier, satisfies constraints more reliably, and avoids the divergence observed in Lag-PPO, showing clear benefits of using dedicated critics for multi-constraint RL.

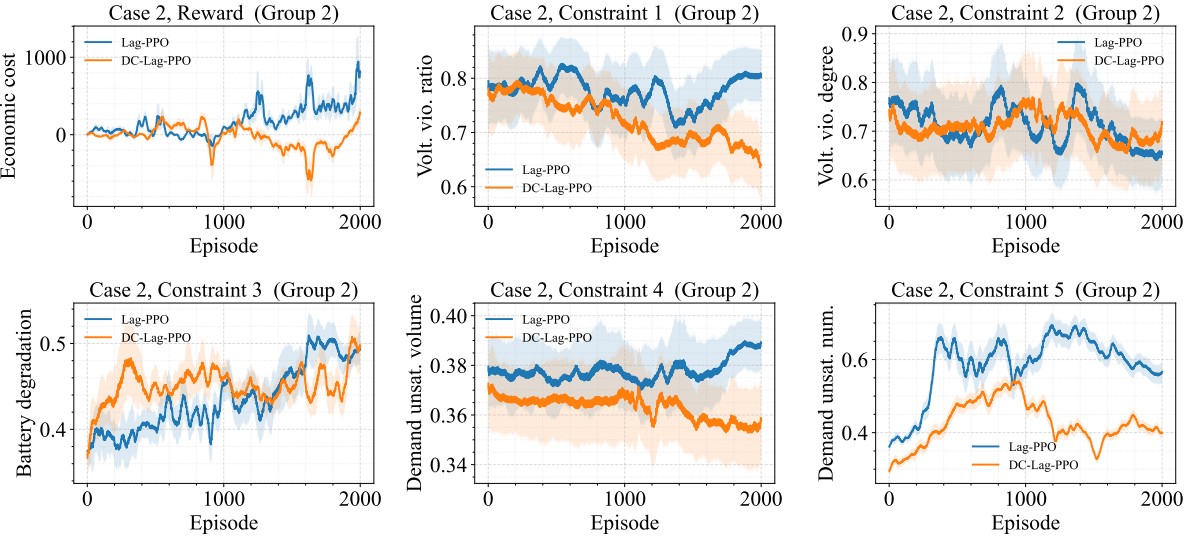

*Figure 36.* Training curves on Lagrangian cost threshold set: [30,39,20,20,30].

## L.6. Lagrangian Multiplier Learning Curves

See Figure 38a.

## L.7. Pareto Fronts

See Fig. 39. Across all reward-constraint and constraint-constraint pairs in this figure, DC-Lag-PPO exhibits consistently superior Pareto fronts. Its frontier lies uniformly closer to the lower-left region, indicating lower violations for comparable (or better) economic cost. Reward vs. all constraint: DC-Lag-PPO achieves strict dominance, producing solutions with both lower cost and lower violations, whereas Lag-PPO spreads widely and lacks a coherent frontier. Voltage-related pairs (c1-c2): DC-Lag-PPO forms a tighter and clearly improved front, showing better joint voltage safety. Battery degradation & dissatisfaction (c3-c4-c5): DC-Lag-PPO consistently pushes the front downward, reducing both degradation and unmet demand simultaneously. Lag-PPO fronts are often fragmented or upward-sloping, reflecting unstable trade-offs caused by aggregated-critic interference. Overall, the DC-Lag-PPO front either envelops or strictly improves upon Lag-PPO across all dimensions, confirming its ability to maintain safer and more efficient trade-offs under multi-constraint settings.

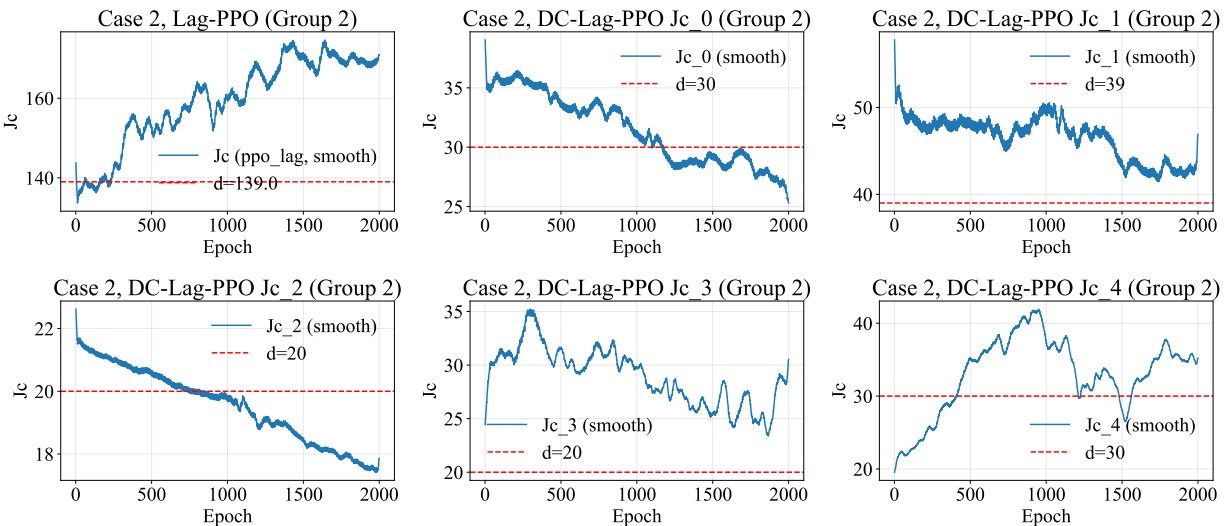

*Figure 37.* Training curves of $J_c$ on Lagrangian cost threshold set: [30,39,20,20,30].

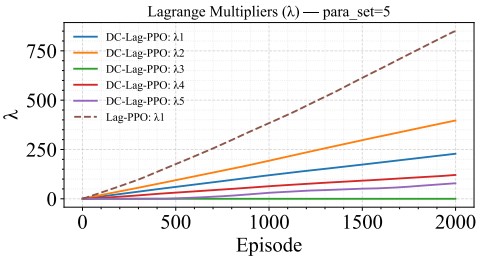

*(a)* Lagrangian cost threshold set [30,39,20,20,30].

*Figure 38.* Lagrangian multiplier $\lambda$ learning curves.

# M. Notation Summary

We summarize the key notations used throughout the paper.

| Symbol | Meaning |
|---|---|
| $\mathcal{S}$ | State space of the CMDP. |
| $\mathcal{A}$ | Action space of the CMDP. |
| $P(\cdot\|s,a)$ | Transition kernel, probability of next state given $(s,a)$. |
| $\gamma \in (0,1)$ | Discount factor. |
| $r(s,a)$ | Reward function. |
| $c_i(s,a)$ | Cost function for constraint $i \in \{1,\dots,m\}$. |
| $d_i$ | Threshold for constraint $i$. |
| $\pi_\theta$ | Stochastic policy parameterized by $\theta$. |
| $\nabla_\theta \log \pi_\theta(a\|s)$ | Policy score function. |
| $J_x(\pi_\theta)$ | Expected discounted return of signal $x$. |
| $\mathcal{L}(\theta,\lambda)$ | Lagrangian objective. |
| $\lambda = (\lambda_1,\dots,\lambda_m)$ | Vector of Lagrange multipliers. |
| $Q_\pi^x(s,a)$ | State–action value function for signal $x$ under $\pi$. |
| $T_\pi^x$ | Bellman operator for signal $x$. |
| $\phi(s,a) \in \mathbb{R}^d$ | Feature vector for linear function approximation. |
| $\Phi$ | Feature matrix stacking $\phi(s,a)$ for all $(s,a)$. |

| | |
|---|---|
| $Q_\omega(s,a) = \phi(s,a)^\top \omega$ | Linear critic parameterized by $\omega$. |
| $D$ | Diagonal weighting matrix with stationary distribution $d_\pi(s,a)$. |
| $A(\theta)$ | System matrix $\Phi^\top D(I - \gamma P_\pi)\Phi$. |
| $b^x(\theta)$ | Right-hand side vector $\Phi^\top D\, x$. |
| $\omega^x(\theta)$ | PBE solution for signal $x$: $A(\theta)\omega^x(\theta) = b^x(\theta)$. |
| $\omega^\lambda(\theta, \lambda)$ | Mixed critic solution. |
| $\eta_t$ | Critic stepsize. |
| $\alpha_t$ | Actor stepsize. |
| $\beta_t$ | Dual stepsize. |
| $\theta_t$ | Actor parameters at iteration $t$. |
| $\lambda_t$ | Dual variables at iteration $t$. |
| $\omega_t$ | Critic parameters at iteration $t$. |
| $\omega_t^\star$ | Instantaneous mixed-critic target at iteration $t$, |
| $e_t$ | Mixed critic error: $e_t = \omega_t - \omega_t^\star$. |
| $e_t^x$ | Dedicated critic error: $e_t^x = \omega_t^x - \omega^x(\theta_t)$. |
| $\zeta_t, \zeta_t^x$ | Martingale-difference noise terms in critic updates. |
| $\Delta_t^\theta$ | Variation from changes in $\theta$, eq. equation 4. |
| $\Delta_t^{\theta,x}$ | Drift term for dedicated critic $x$. |
| $g_t^\star$ | True actor gradient at iteration $t$. |
| $\widehat{g}_t$ | Actor gradient estimate using mixed critic. |
| $\widehat{g}_t^{\mathrm{multi}}$ | Actor gradient estimate using dedicated-critic, eq. equation 63. |
| $B_t$ | Actor-gradient bias (mixed critic): $\widehat{g}_t - g_t^\star$. |
| $B_t^{\mathrm{multi}}$ | Actor-gradient bias (dedicated-critic): $\widehat{g}_t^{\mathrm{multi}} - g_t^\star$. |
| $G$ | Uniform bound on $\|\nabla_\theta \log \pi_\theta(a|s)\|$. |
| $L_\phi$ | Uniform bound on $\|\phi(s,a)\|$. |
| $\mu$ | Uniform lower bound on eigenvalues of $A(\theta)$. |
| $C_\lambda, C_\theta, \widetilde{C}_\theta$ | Lipschitz / drift constants from error bounds. |

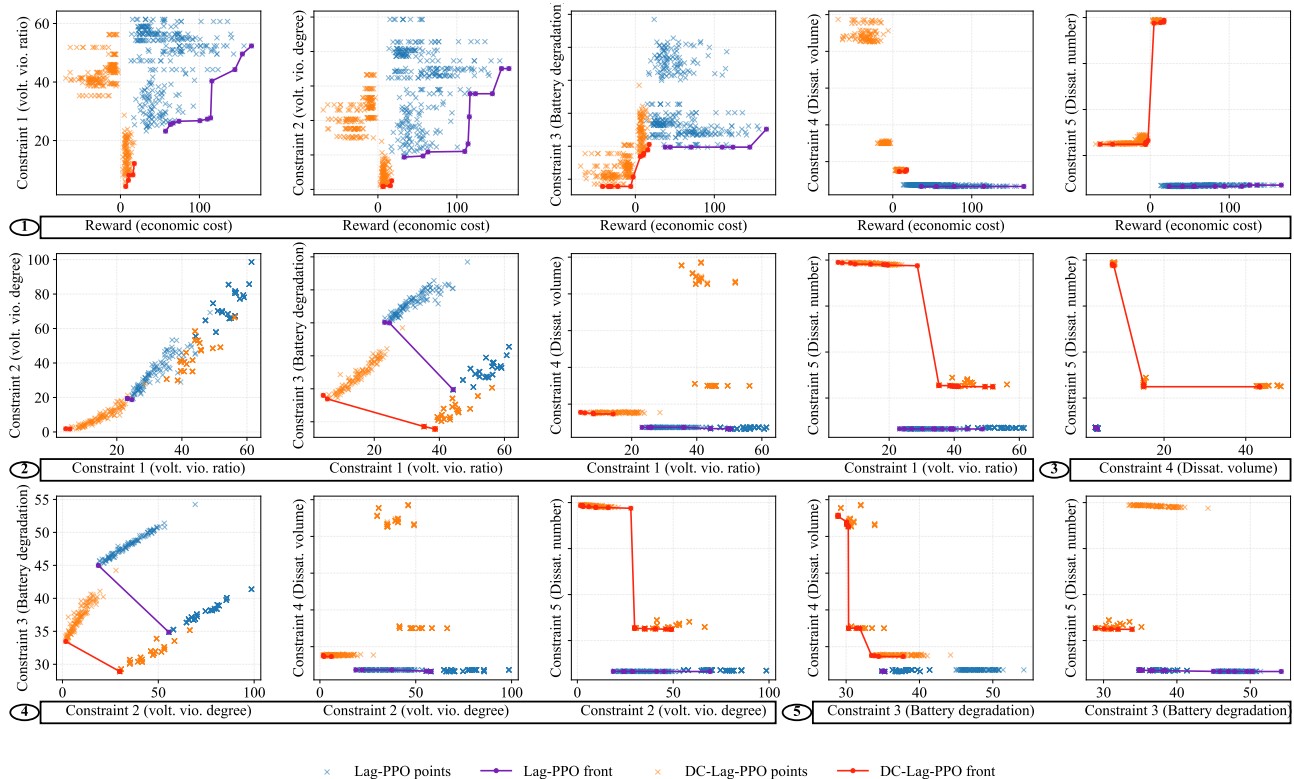

*Figure 39.* Pareto fronts from the test results on Lagrangian cost threshold set [30,39,20,20,30], where the black dotted lines are the thresholds $d$.

## N. Additional Experiments for the Environment with single constraints

To assess the practical impact of critic design on safe RL performance, we compare our dedicated-critic PPO-Lagrangian (separate value functions for reward and constraint cost) against a standard mixed-critic PPO-Lagrangian baseline across a diverse set of Safety-Gymnasium and velocity-control benchmarks. Specifically, we evaluate on the navigation tasks `SafetyCarGoal1-v0`, `SafetyCarButton1-v0`, and `SafetyPointGoal1-v0`, as well as the continuous-control environments `SafetyAntVelocity-v1`, `SafetyHalfCheetahVelocity-v1` (SafetyHalfCheetahV-v1), and `SafetyHopperVelocity-v1`, which together span both sparse goal-reaching rewards with collision costs and dense velocity-tracking settings with safety penalties. For each environment and method, we report the mean ± standard deviation of episodic reward and episodic cost over multiple evaluation rollouts after training, so that higher reward and lower cost indicate a better reward–safety trade-off. As summarised in Table 6, these experiments allow us to directly test whether separating reward and cost critics improves constraint satisfaction and stabilises learning compared to the widely used mixed-critic formulation.

Across all six benchmarks, the dedicated-critic PPO–Lagrangian consistently improves safety and often improves reward relative to the mixed-critic baseline. On the navigation tasks, `SafetyCarGoal1-v0` shows the clearest win: separating reward and cost critics raises the mean return from 1.12 to 14.56 while *also* reducing mean cost from 55.34 to 21.72, and it substantially shrinks the very large cost variance of the mixed critic. A similar pattern appears on `SafetyCarButton1-v0`: both methods obtain very low rewards (reflecting task difficulty), but the dedicated critic roughly halves the average cost (51.4 vs. 107.14) and reduces variability, indicating more reliable constraint satisfaction even when the policy is far from optimal. On `SafetyPointGoal1-v0`, the mixed critic achieves slightly higher reward (15.78 vs. 13.03) but at the price of more than double the mean cost (52.81 vs. 23.97), so the dedicated critic offers a strictly safer solution with only a modest reward gap.

The MuJoCo velocity environments highlight the benefit of dedicated critics even more strongly. On `SafetyAntVelocity-v1`, the dedicated-critic agent improves reward from 2821.72 to 3324.67 *and* cuts mean cost by more than half (28.52 to 13.01). On `SafetyHalfCheetahVelocity-v1`, the effect is even more pronounced:

*Table 6.* Performance comparison between dedicated-critic PPO-Lag and mixed-critic PPO-Lag. Values are mean $\pm$ standard deviation over evaluation episodes.

| Env | Dedicated critics | | Mixed critic | |
|---|---|---|---|---|
| | Reward | Cost | Reward | Cost |
| SafetyCarGoal1-v0 | $14.56 \pm 8.97$ | $21.72 \pm 32.06$ | $1.12 \pm 9.23$ | $55.34 \pm 102.32$ |
| SafetyCarButton1-v0 | $0.36 \pm 1.81$ | $51.40 \pm 82.14$ | $1.51 \pm 3.64$ | $107.14 \pm 132.22$ |
| SafetyAntVelocity-v1 | $3324.67 \pm 83.21$ | $13.01 \pm 6.32$ | $2821.72 \pm 201.91$ | $28.52 \pm 8.37$ |
| SafetyHalfCheetahV-v1 | $3035.76 \pm 287.42$ | $4.14 \pm 2.37$ | $2234.245 \pm 345.73$ | $45.82 \pm 7.15$ |
| SafetyHopperVelocity-v1 | $1002.73 \pm 723.64$ | $14.87 \pm 20.74$ | $1238.83 \pm 465.35$ | $17.21 \pm 12.23$ |
| SafetyPointGoal1-v0 | $13.03 \pm 7.15$ | $23.97 \pm 33.16$ | $15.78 \pm 3.18$ | $52.81 \pm 17.10$ |

reward increases from 2234.25 to 3035.76, while cost drops from 45.82 to 4.14, giving a dramatically better reward–safety trade-off. `SafetyHopperVelocity-v1` is the only case where the mixed critic slightly outperforms in reward (1238.83 vs. 1002.73), but the dedicated critic still attains lower cost (14.87 vs. 17.21) and comparable variance. Overall, these results align with our theoretical claim: by removing the $\lambda$-induced target drift, dedicated critics provide more stable value estimates, which in practice translates into systematically lower constraint violations and, in most tasks, equal or higher task performance than the mixed-critic formulation.

## O. Asymptotic vanishing of mixed–critic bias

*Remark* O.1 (Asymptotic vanishing of mixed–critic bias). Under Assumption 4.1, the step–size ratios satisfy $\alpha_t/\eta_t \to 0$ and $\beta_t/\eta_t \to 0$ as $t \to \infty$. Hence both limsup terms on the right-hand side of Theorem 4.7 are zero, and we obtain $\limsup_{t\to\infty} \mathbb{E}\big[\|B_t\|\big] = 0$. That is, in the idealised linear SA regime of Assumption 4.1, the mixed–critic actor-gradient bias vanishes asymptotically, so mixed–critic Lag-PPO and its dedicated–critic variant coincide in the limit. By contrast, this regime requires Robbins–Monro step sizes and strong timescale separation between critic, actor, and dual, which is *not* how deep PPO–Lagrangian is used in practice with (effectively) constant learning rates and Adam. Our experiments follow this practical setting, where a non-negligible mixed–critic bias can persist, which is why Theorem 4.7 is stated in terms of the ratios $\beta_t/\eta_t$ and $\alpha_t/\eta_t$, keeping the bound informative for realistic, non-asymptotic schedules.

To more directly connect our empirical results to the asymptotic setting in Assumption 4.1 and Theorem 4.7, we also study a Robbins–Monro (RM) variant in a simplified, idealised environment. The goal of these experiments is to approximate the stochastic-approximation regime assumed in the theory and to compare mixed and dedicated critics under those conditions.

**Experimental design.** We consider a small constrained MDP with two constraints and a low-dimensional state and action space, for which we can reliably measure reward, total constraint violation, and dual-variable behaviour over training(same as Appendix H). In this setting we implement two variants:

- **Mixed RM**: a single mixed critic trained on the scalarised signal $r_\lambda = r - \sum_i \lambda_i c_i$.

- **Dedicated RM**: separate critics $V_r$ and $V_{c_i}$ trained on reward and each constraint cost, respectively.

Both variants use the *same* data (trajectories), and differ only in how the value function is parameterised. To align with Assumption 4.1, we use Robbins–Monro learning-rate schedules for the actor, critics, and dual variables:

$$\alpha_t = \frac{\alpha_0}{1 + k_\pi t}, \quad \eta_t = \frac{\eta_0}{1 + k_V t}, \quad \beta_t = \frac{\beta_0}{1 + k_\lambda t},$$

chosen such that $\sum_t \alpha_t = \infty$, $\sum_t \alpha_t^2 < \infty$ (and similarly for $\eta_t, \beta_t$), and with a clear time-scale separation $\eta_t \gg \alpha_t \gg \beta_t$. All other hyperparameters are held fixed across the two variants. We track three metrics over training:

1. expected reward (per step),

2. total constraint violation (per step),

3. the absolute difference between dual variables (to monitor symmetry of the Lagrange multipliers in the two-constraint case).

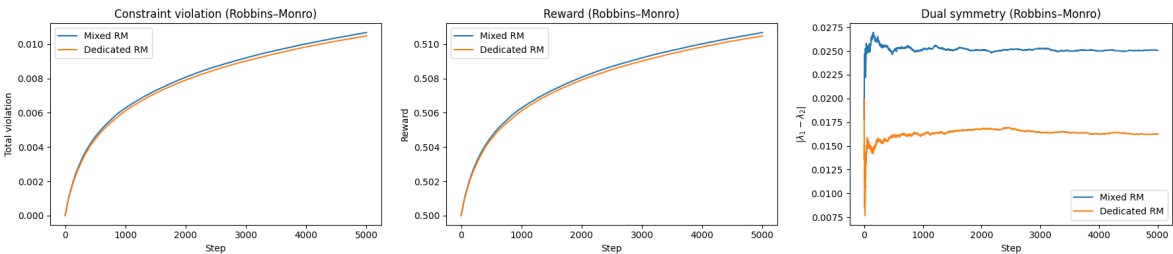

*Figure 40.* Robbins–Monro Lag-PPO in a two-constraint CMDP: comparison between mixed and dedicated critics

Figure 40 shows these three quantities for both Mixed RM and Dedicated RM.

In this idealised, near-linear Robbins–Monro setting, the mixed-critic and dedicated-critic variants behave very similarly. Their reward curves almost overlap, both methods achieve comparable levels of constraint satisfaction, and the dual variables converge to a very similar symmetric configuration. This is consistent with Theorem 4.7: when the learning rates satisfy the stochastic-approximation conditions and the critics can track their targets on the fastest time scale, the additional dual-induced drift term in the mixed critic does not translate into a noticeable difference in the limiting behaviour. In other words, this toy Robbins–Monro experiment can be seen as a direct empirical realisation of the asymptotic linear-theory predictions, and it serves as a sanity check that our finite-sample implementation matches the behaviour analysed in the theoretical section.

## P. Robbins–Monro Lag-PPO

In RM Lag-PPO, we replace Adam with plain SGD and use diminishing step-size schedules for the actor, critic, and dual updates. The data-collection and PPO objective remain unchanged; only the optimiser and step-size schedules differ. We instantiate a *mixed* RM Lag-PPO, Lag-PPO (one critic) and a *Dedicated* RM variant (separate critics), and train them on the complex power system environments. However, as we cannot run for effectively unlimited time, and strict Robbins–Monro schedules cause step sizes to become very small within a realistic training budget, the RM Lag-PPO variant performs quite similarly with the standard constant–stepsize Lag-PPO(Figure 41). This similarity is further reinforced by PPO's ratio clipping (and gradient clipping), which effectively bounds the size of each policy update even under a nominally constant learning rate, making standard Lag-PPO behave in practice like a conservatively damped method whose effective step sizes are not far from those induced by a Robbins–Monro schedule over a finite training horizon.

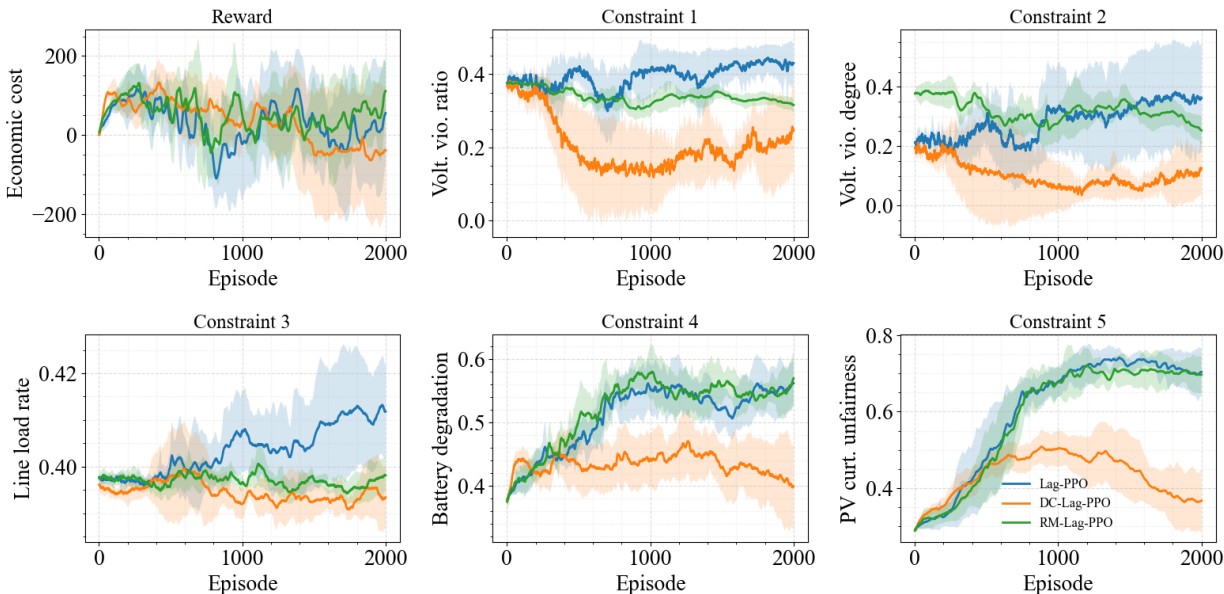

*Figure 41.* Training curves of Lag-PPO.

## Q. Statement on LLM Usage

Large language models (LLMs), such as ChatGPT, were used solely for editorial assistance in this work. Their role was limited to improving grammar, rephrasing sentences, and enhancing clarity and readability of the authors' original text. No LLM was used to generate original scientific content, analysis, or results. The authors take full responsibility for the integrity and validity of the work presented.

