# OpenReview forum: "Why Dedicated Critics: Eliminating Target Drift in Multi-Constraint RL"
_ICML.cc/2026/Conference — ICML 2026 regular_

### Official Review · Reviewer_JWii · 2026-02-21

**Soundness:** 3
**Presentation:** 3
**Significance:** 3
**Originality:** 3
**Overall Recommendation:** 4
**Confidence:** 2

**Summary:**

This paper studies critic architecture in multi-constrain RL and provides a theoretical justification for using dedicated critics—i.e., learning a separate value function for each cost—over a mixed critic that aggregates costs via time-varying Lagrange multipliers and learns a single value on the aggregated cost. It shows that the mixed-critic design suffers from dual-induced target drift: as the multipliers are updated, the critic’s learning target becomes nonstationary, introducing an additional bias term in both the value estimate and the actor’s gradient estimate. In contrast, dedicated critics eliminate this multiplier-driven drift because each critic’s target depends only on the policy, not on the evolving multipliers. The theory is supported empirically on the experiments on a constrained bandit and a multi-constraint power-system control task, where the dedicated-critic approach yields more stable training and better constraint satisfaction than the mixed-critic alternative.

**Compliance With Llm Reviewing Policy:**

Affirmed.

**Key Questions For Authors:**

- I am uncertain whether lambda-based cost aggregation and mixed critics are widely used implementations. In my impression, few works adopt this approach, as one might intuitively anticipate the instability it brings. The cited literature does not make these details clear. I would appreciate it if the authors could clearly list papers/benchmarks that use mixed critics and pinpoint the relevant implementation details. This would deepen the reader’s understanding of the mixed-critic structure.

- What is the exact number of training steps in Figure 3? It seems difficult to judge the final convergence from this figure (e.g., constraints 4 and 5 still appear to be decreasing).

- In Figure 4, "pts" appears in the legend but seems unexplained.

- In Section 2, a citation should be added when WCSAC is mentioned.

**Limitations:**

Yes

**Strengths And Weaknesses:**

### Presentation
- The paper is generally well structured: the motivation is clear, the theoretical claims follow a coherent logic, and the experiments are aligned with the stated mechanism.
- However, several presentation details can be further clarified (See Question 2-4)
### Originality
- The innovation mainly lies in exploring how critic structures affect solving multi-constraint problems. When extending existing constrained RL algorithms to multi-constraint scenarios, how to choose the critic architecture is an somehow unexplored blind spot, which, as the paper states, could be a significant factor affecting performance.
- The main theoretical conclusion of the paper is intuitive and convincing: the cost aggregation determined by lambda is unstable and violates the expected assumptions about the target critic in value updates. The paper extends this intuition into rigorous theoretical results.
### Significance:
- The theoretical analysis provides normative guidance for network architecture choices in practice.
- However, the discussion on the existing usage of mixed critics could be strengthen—see Question 1.
### Soundness:
- The experimental results are relatively clear, illustrating the effectiveness from multiple metrics and supporting the theoretical conclusions.
- However, the constrained bandit environment is simple and intuitive for comparison but does not represent complex environments, while the applied scenario experiments may be difficult to reproduce without data. Standard Safe-RL benchmarks are ideal for reproduction, but the paper only considers single constraints (though many environments inherently only provide single constraints), which is inconsistent with the paper’s core multi-constraint viewpoint. This might raise some concerns regarding reproducibility.

---

> ### Author Rebuttal · Authors · 2026-03-31
>
> ***Question regarding reproducibility.***
>
> **Response:** We thank the reviewer for raising this concern. We clarify that we provide all code in the supplementary material covering the energy-system experiments, including the environment setup, training pipeline, and evaluation scripts.
>
> We agree that these application-driven environments are more complex than standard benchmarks. To support reproducibility, our implementation builds on publicly available and widely used simulators, namely pandapower [1], together with well-defined experimental configurations. We provide the necessary details and code to reproduce the reported results. We also apologise for not citing pandapower in the submitted version; as it is widely used in this area, we inadvertently omitted the citation. We will add the appropriate reference in the revision.
>
> At the same time, we include the CMDP bandit as a fully controlled and lightweight setting to ensure that the core mechanism can be easily reproduced and verified. The application experiments are intended to demonstrate practical relevance rather than replace standard benchmarks.We will also make the complete codebase publicly available upon acceptance to ensure full reproducibility.
>
> [1] L. Thurner, A. Scheidler, et al, pandapower - an Open Source Python Tool for Convenient Modeling, Analysis and Optimization of Electric Power Systems, in IEEE Transactions on Power Systems, 2018.
>
> ***Question 2: list papers/benchmarks that use mixed critics and pinpoint the relevant implementation details.***
>
> **Response:** Our work is motivated by the gap between common safe-RL practice and the limited theoretical understanding of critic design in multi-constraint settings. To our knowledge,while Lag-based safe RL method has been widely used and cited in recent literature, there has been very little theoretical understanding of which critic architecture should be used in multi-constraint settings, even though such settings are far more realistic in real-world applications. In standard safe-RL practice, Lagrangian PPO-style baselines are widely used, and public benchmark/toolkit implementations typically default to a single critic setting, often associated with a single Lagrange multiplier. This pattern appears in widely used frameworks such as Safety Gym [2] and OmniSafe [7].
>
> However, practical systems usually involve multiple safety, resource, or operational constraints simultaneously. In these settings, many works simply carry over the single-constraint formulation into multi-constraint scenarios as one of the baseline [3][4][5], without explicitly reconsidering whether that critic design remains appropriate. Other works adopt multi-critic design without explicitly analysing their theoretical implications behind the design [6]. A closer look at existing toolkit implementations of PPO-Lagrangian for multi-constraint problems suggests that the critic design is often chosen in a fairly ad hoc manner, with limited explicit justification. Our paper is motivated by this gap between standard safe-RL practice and real deployment needs and aims to provide both theoretical analysis and empirical clarity.
>
> We will add this discussion to the main text to better highlight this practical and conceptual gap.
>
> [2] Ray, Alex, et al. "Benchmarking safe exploration in deep reinforcement learning." arXiv:1910.01708 7.1 (2019): 2.
>
> [3] Yao, Yihang, et al. "Gradient shaping for multi-constraint safe reinforcement learning." 6th annual learning for dynamics & control conference. PMLR, 2024.
>
> [4] Zhang, Linrui, et al. "Penalized proximal policy optimization for safe reinforcement learning." arXiv:2205.11814 (2022).
>
> [5] Ma, Jianmina, et al. "Adversarial Constrained Policy Optimization: Improving Constrained Reinforcement Learning by Adapting Budgets." arXiv:2410.20786 (2024).
>
> [6] Zheng, Jun, et al. "Safe reinforcement learning for industrial optimal control: A case study from metallurgical industry." Information Sciences 649 (2023): 119684.
>
> [7] Ji, Jiaming, et al. "Omnisafe: An infrastructure for accelerating safe reinforcement learning research." Journal of Machine Learning Research 25.285 (2024): 1-6.
>
> ***Question regarding the training steps and performance of Figure 3***
>
> **Response:** The current training step is 2000 episodes. To further validate our proposed method, we extend training to 10000 episodes, and the results are provided in Figure 1 through the following link: https://anonymous.4open.science/r/RebuttalPage_Why_Dedicated_Critic-5FE0
>
> ***Unexplained "pts".***
>
> **Response:** “pts” in Figure 4 denotes “points,” and we will spell this out explicitly in the revision for clarity.
>
> ***In Section 2, a citation should be added when WCSAC is mentioned***
>
> **Response:**  We will add the appropriate citation in the revision.
>
> [8] Yang, Q., et al. WCSAC: Worst-case soft actor critic for safety constrained reinforcement learning. AAAI 2021 (Already cited in the submitted manuscript.)

---

> > ### Author Rebuttal · Reviewer_JWii · 2026-04-02
> >
> > Thanks for the authors’ response, which addresses most of my concerns about the experiments. I will keep my current score.

---

> > > ### Author Response · Authors · 2026-04-06
> > >
> > > Thank you for your follow-up and for taking the time to carefully review our responses. We are pleased to hear that our reply has addressed all of your concerns. We sincerely appreciate your consideration and thank you for your continued support for the paper.

---

### Official Review · Reviewer_f2NF · 2026-03-09

**Soundness:** 2
**Presentation:** 3
**Significance:** 2
**Originality:** 3
**Overall Recommendation:** 4
**Confidence:** 3

**Summary:**

This paper studies the design of critic architectures in multi-constraint safe reinforcement learning. The authors identify a "target drift" issue in mixed-critic methods, which arises from dual variable updates. They propose a dedicated-critic architecture that theoretically eliminates this drift. Theoretical analysis under linear function approximation provides error bounds showing the mixed critic's bias depends on dual step sizes, while the dedicated critic's bias does not. Empirical evaluations on a diagnostic bandit, a complex power system simulator, and an EV charging problem demonstrate that dedicated critics lead to more stable training, lower constraint violations, and better Pareto frontier trade-offs compared to PPO-Lagrangian baselines.

**Compliance With Llm Reviewing Policy:**

Affirmed.

**Final Justification:**

Thank you for the additional experiments, which partially address my first concern. However, my second concern about gradient alignment as a meaningful diagnostic signal remains unresolved.

**Key Questions For Authors:**

1.  My comments are provided in the weaknesses section.
2. There is a typo on Page 4, Paragraph 3: ``sfol-lows'' should be ``follows''.

**Limitations:**

yes

**Strengths And Weaknesses:**

Strengths:
1. The paper tackles a design choice in multi-constraint safe RL that is often treated as an afterthought in the literature. Identifying and formalizing the "target drift" phenomenon is a valuable contribution.
2. The paper provides a clean theoretical analysis within a linear function approximation setting, clearly deriving the source of the mixed critic's bias and contrasting it with the dedicated critic's error bound. This provides a solid conceptual foundation for the proposed method.
3. The experiments on power systems and EV charging simulators, and the use of multiple metrics, provide a comprehensive view of the method's benefits.

Weaknesses:

1. The claim of ``independence'' is an idealization. Dedicated critics remain implicitly coupled---they share the same policy dynamics, on-policy data, and are combined via $\lambda$. The analysis masks this with a coarse norm bound that ignores error direction and correlation ($\|\sum \lambda_i e^{(c_i)}\| \le \sum \|\lambda_i\| \|e^{(c_i)}\|$). While dedicated critics eliminate dual-induced target drift, the analysis assumes worst-case error bounds and does not characterize how approximation errors from multiple critics interact. In practice, correlated critic errors might affect the resulting gradient bias.

2. Furthermore, while the dedicated critic removes the dual-induced drift term, it replaces a single coupled estimation problem with multiple independently trained critics. The resulting actor gradient aggregates errors from each constraint critic, potentially leading to error accumulation that scales with the number of constraints. The theoretical analysis bounds this interaction only in a worst-case norm sense and does not characterize how the total bias scales with the number of constraints $K$. As $K$ increases, it remains unclear whether the dedicated critic yields strictly better asymptotic behavior or merely redistributes the estimation error across multiple sources.

3. While higher alignment with the Lagrangian gradient is presented as evidence of superiority, this metric implicitly assumes that the instantaneous Lagrangian gradient provides a reliable optimization direction. In non-convex saddle-point settings with noisy dual updates, improved gradient fidelity does not necessarily imply improved global optimization behavior. A deeper dynamical analysis would strengthen this claim.

---

> ### Author Rebuttal · Authors · 2026-03-31
>
> ***Question regarding approximation errors from multiple critics interact or accumulate as the number of constraints increases***
>
> **Response:** To address this concern, we provide additional empirical results showing that performance remains stable as the number of constraints increases, indicating that error accumulation is not severe in the regimes we study.
> The table below reports EVCS results under different critic configurations. The first column lists evaluation metrics, including five constraint-related metrics (C1–C5). The columns labeled 1–5 indicate the number of critics adopted using our method. The value in each cell corresponds to the violation for each constraint of different numbers of critics. As the number of critics increases from 1 to 5, constraint violations generally decrease, often substantially, indicating that error accumulation is not severe in the regimes we study and instead pointing to the mixed-critic drift term as the main cause of the poor performance.
>
> | Metric | 1 | 2 | 3 | 4 | 5 |
> |--------|---:|---:|---:|---:|---:|
> | C1 | 62.96 | 43.61 | 45.85 | 43.87 | 42.17 |
> | C2 | 54.77 | 31.53 | 32.21 | 29.08 | 26.47 |
> | C3 | 0.405 | 0.400 | 0.374 | 0.361 | 0.358 |
> | C4 | 32.23 | 35.59 | 30.00 | 19.20 | 17.35 |
> | C5 | 210.95 | 227.61 | 195.86 | 150.42 | 99.82 |
>
> (Note: Columns 1–5 indicate how many critics are used: 1 means a single critic handles all five constraints, 2 means one critic is dedicated to C1 and the other handles C2–C5 together, and higher numbers continue this progressive split until 5 means one dedicated critic per constraint.)
>
> ***Question regarding the claim of independence***
>
> **Response:** We thank the reviewer for this point. We show in theory that each dedicated critic’s target is independent of the evolving multipliers $\lambda$, whereas the mixed critic’s target explicitly depends on $\lambda$, which introduces an extra dual-induced drift term. This is exactly the distinction formalized in Section 4.2, where each dedicated critic’s fixed point depends on $\theta$ and the signal $x$, but not on $\lambda$.
>
> In the revised manuscript, we will revise the wording to make clear that our “independence” claim refers specifically to independence of the critic target from the evolving multipliers, rather than independence of all learning dynamics.
>
> ****Question regarding the global optimization behavior in non-convex saddle-point settings with noisy dual updates.***
>
> **Response:** Thank you for this insightful comment. We will include this point in the limitations discussion. We agree that in non-convex stochastic saddle-point settings with noisy dual updates, improved local gradient alignment does not necessarily guarantee better global optimisation behavior. This limitation is not specific to our method, but is shared by gradient-based primal–dual approaches more broadly.
>
> To isolate the effect studied in our theory, the submitted manuscript includes a controlled CMDP bandit experiment (section 5.1). In this setting, the true instantaneous Lagrangian gradient is available in closed form, so gradient alignment serves as a mechanistic diagnostic of whether the actor update tracks the ideal current update direction more faithfully.
>
> This aligns with the scope of our theory: Theorems 4.7 and 4.9 characterize actor-gradient bias relative to the ideal instantaneous Lagrangian gradient, rather than establishing stronger global convergence guarantees for general non-convex primal–dual dynamics. We therefore interpret gradient alignment together with outcome metrics such as constraint violation and dual oscillation, rather than as a standalone claim of superiority.
>
> ***Question regarding the typos***
>
> **Response:**  Thanks for pointing this out. We appreciate the suggestion and will carefully proofread the manuscript and correct all typos in the revision.

---

> > ### Author Rebuttal · Reviewer_f2NF · 2026-04-03
> >
> > 1 First, thank the authors for providing the additional experiment regarding approximation errors. However, I still have concerns. First, the experiment lacks critical details. For example, was the network architecture fixed across configurations? Without a controlled experiment, the claimed mechanism remains unverified.
> > 2 For gradient alignment, the authors claimed that gradient alignment serves as a diagnostic for whether the actor update tracks the ideal direction, not as a guarantee of global convergence. In Section 5.1, higher gradient alignment is presented as evidence of superiority (being more reliable). However, in the main experiments (5.2, 5.3), gradient alignment is not reported—only constraint violations are. The bandit experiment shows a correlation between higher alignment and lower violations, but in that setting, the final policies are near-feasible. In contrast, the main experiments show policies that remain infeasible (e.g., Table 2, C5: 99.82 vs threshold 30). It is unclear whether gradient alignment is really a meaningful diagnostic signal.

---

> > > ### Author Response · Authors · 2026-04-06
> > >
> > > Thank you for this thoughtful comments. We sincerely appreciate the time and effort you have invested in helping us improve the paper. Below, we address your remaining questions point by point.
> > >
> > > First, in our submitted manuscript, both the mixed and dedicated methods have used the same PPO backbone and the same MLP design. The actor architecture is identical across methods, and each critic also uses the same two-layer MLP architecture. The dedicated-critic only replaces the single mixed critic with multiple parallel critics of the same architecture (See Table 1 for detailed results).
> > >
> > > To further isolate the effect of architecture, we add three controlled experiments to demonstrate that observed performance differences are not primarily driven by architectural changes (see Figures 3, 4, and 5 in https://anonymous.4open.science/r/RebuttalPage_Why_Dedicated_Critic-5FE0):
> > > - We run a **parameter-matched comparison**, where the total number of critic parameters is kept the same between the dedicated and mixed/shared setups, while keeping the same network depth. The performance gap remains, indicating that the advantage is not simply due to increased model capacity (Figure 3 in the link).
> > >
> > > - We test a **shared-trunk multi-head ablation** (one head per constraint). Its performance is substantially better than the original mixed critic but still not as good as the fully dedicated critic, as we have discussed in the limitation section, a shared-backbone multi-head critic may still suffer from cross-signal interference even when each head is per-signal (Figure 4 in the link).
> > > - We test a **stronger constraint-scaling baseline** using penalty ×10, which also provides only partial improvement: some constraints become better, but others remain poor. The overall performance again lies between the original mixed critic and the fully dedicated critic. This indicates that simple reweighting does not remove the underlying issue that the critic is still fitting a single aggregated signal (Figure 5 in the link).
> > > - We already included a dual-timescale intervention in Appendix P. When the dual updates are slowed or controlled, the mixed–dedicated gap is reduced, which is consistent with the proposed dual-induced target-drift mechanism (Appendix P).
> > >
> > > Regarding the gradient alignment (see Figures 6 in https://anonymous.4open.science/r/RebuttalPage_Why_Dedicated_Critic-5FE0), in the bandit setting, the true gradient is available in closed form, so gradient alignment can be measured directly. In the main control experiments, however, the exact gradient is unavailable. To address this, we define a **proxy gradient alignment metric** to quantify whether the actor update produced by the current method is directionally consistent with a higher-fidelity reference Lagrangian gradient. At a checkpoint $t$, let $g_t^{\mathrm{train}}$ denote the actual actor gradient used by the algorithm, and let $g_t^{\mathrm{ref}}$ denote a high-sample reference gradient estimated under the frozen pair $(\theta_t,\lambda_t)$ using additional on-policy rollouts. The proxy alignment is then defined as $\mathrm{Align}^{\mathrm{proxy}}_t = \frac{\langle g_t^{\mathrm{train}}, g_t^{\mathrm{ref}}\rangle} {\|g_t^{\mathrm{train}}\|_2\,\|g_t^{\mathrm{ref}}\|_2+\varepsilon}$. This metric serves as a local diagnostic of actor-update fidelity: if dedicated critics indeed reduce the actor-gradient bias predicted by the theory, then they should exhibit higher alignment with the reference Lagrangian direction. Our design achieves substantially better gradient alignment than the mixed-critic design, as shown in Figure 6.
> > >
> > > We also introduce **dual oscillation magnitude**, which shows that under a mixed-critic design, updates of the dual variables $\lambda$ change the critic target itself and thereby introduce additional instability into actor learning (Figure 7 in the link). A simple proxy is to track the volatility of the dual variables during training. In analogy to the bandit analysis, one may also report a sliding-window oscillation magnitude such as$\mathrm{Osc}_t = \sqrt{ \max\!\left( 0,\; \mathbb{E}_t[\Delta^2] - \bigl(\mathbb{E}_t[\Delta]\bigr)^2 \right)}$, where $\Delta$ denotes a scalar summary of dual variation within the window. The dedicated critic is much more stable in terms of oscillation than the mixed critic.
> > >
> > > Overall, these additional controls suggest that the advantage of dedicated critics is not mainly explained by architectural changes alone. Rather, the fully dedicated design consistently performs best. This is most consistent with our theoretical argument that the central issue is the structural coupling in the mixed critic, especially through the dual-induced target-drift mechanism.

---

### Official Review · Reviewer_3F7m · 2026-03-13

**Soundness:** 3
**Presentation:** 3
**Significance:** 2
**Originality:** 3
**Overall Recommendation:** 4
**Confidence:** 4

**Summary:**

Lagrangian-based methods are widely used in safe reinforcement learning for constrained Markov decision processes (CMDPs). As discussed in the paper, prior approaches differ in how they parameterize value estimation: some use a mixed critic, where constraint signals are aggregated into a single scalarized cost estimate (possibly together with reward), while others use dedicated critics with separate estimators for reward and for each constraint. The paper argues that this architectural choice has received limited explicit theoretical attention and then investigates that question in a linear stochastic approximation setting. The paper shows that the use of mixed critics leads to an additional bias term that scales with the ratio between the dual step size, $\beta_t$, and the critic step size, $\eta_t$. Beyond that linear setting used for the theoretical analysis, the paper compares the performance of mixed and dedicated critics on constrained reinforcement learning problems including a multi-constraint power system application problem, an EV charging problem, and safe RL benchmarks with single constraints (appendix). Numerical experiments show that the use of dedicated critics can lead to lower constraint violations. At the same time, the paper also notes that under diminishing step sizes that slow the dual updates, the mixed-critic bias vanishes.

**Compliance With Llm Reviewing Policy:**

Affirmed.

**Final Justification:**

My main concern about the theoretical framing has been substantially alleviated: the revised version now motivates the theory more clearly, makes clear that the asymptotic gap disappears under the stated assumptions, and points readers to the relevant appendix discussion immediately after the main theorem. I also found the new empirical ablations helpful, especially the parameter-matched and shared-trunk multi-head comparisons. To me, these results suggest that representational power accounts for part of the observed gap, but that the dedicated critic still performs best overall. My remaining reservation is that the constraint-scaling ablation only partially addresses the normalization issue I had in mind: a global penalty multiplier is not necessarily the same as per-constraint normalization. Still, all in all, I think the revised experiments make the empirical case materially stronger, and I am updating my assessment accordingly.

**Key Questions For Authors:**

1. Theorem 4.7 shows an additional mixed-critic bias term depending on the ratio between the learning rates, but Assumption 4.1 requires these ratios to vanish, and Appendix O states that mixed and dedicated critics coincide asymptotically under this regime. Can the authors clarify earlier in the paper that the theory seems to motivate mainly finite-timescale settings, and point to that discussion in appendices O and P?

2. In several application plots and tables, both architectures still appear to violate constraints, with dedicated critics mostly showing lower violations rather than clear satisfaction of the CMDP constraints. Do the final policies in the power-system and EV-charging experiments satisfy the constraints approximately, or are the results mostly looking at fixed training horizons? What happens in longer training settings?

3. Since actor-critic methods in practice often already use heuristic timescale separation, e.g. slower dual updates, how sensitive are the mixed-critic results to that? Could stronger timescale separation reduce the observed gap without changing the architecture?

**Limitations:**

Yes.

**Strengths And Weaknesses:**

Strengths

1. The paper focuses on an architectural question that is practically relevant but usually under-discussed in the safe RL literature.

2. The central argument is intuitive: when constraint signals are aggregated through the current Lagrange multipliers, dual updates induce an additional source of target drift for the critic, whereas dedicated critics avoid this direct dependence.

3. The empirical results are consistent with the paper’s main intuition, in the sense that dedicated critics often reduce violations relative to the mixed-critic baseline in the reported results.

Weaknesses:

1. The paper’s theory is developed in a linear stochastic-approximation regime with diminishing step sizes and timescale separation (Assumption 4.1). That assumption is needed for the analysis of the induced bias term, but under the same assumption the mixed-critic bias term also vanishes asymptotically, and the appendix states that mixed and dedicated critics coincide in the limit. I believe that that reference to the results presented in the Appendix should be stated earlier in the text, e.g., immediately after the main theorem.

2. The analysis is carried out for a linear stochastic approximation scheme with diminishing step sizes and explicit timescale separation, whereas the experiments use PPO training with nonlinear function approximation. This does not invalidate the theoretical intuition, but it also means that the theory is not directly analyzing the empirical regime of interest.

3. In practice, actor-critic methods often already rely on heuristic timescale separation, with, e.g., slower dual updates. While this is not the same as the regime assumed in the theory, it raises the question of whether some of the mixed-critic degradation could be reduced through better tuning of the learning rates.

4. Since the paper is formulated in a Lagrangian CMDP framework, I would expect the empirical section to provide some evidence that training is approaching approximately feasible policies, rather than showing only that one architecture violates constraints less than another within a fixed number of training steps. In several plots and tables, both mixed and dedicated critics appear to remain significantly above the feasibility threshold. This makes it unclear whether the methods are approaching feasible solutions, converging to different infeasible solutions, or simply have not converged yet.

5. The paper seems to mostly attribute the empirical gains of dedicated critics to eliminating the additional bias that arises with mixed critics. However, dedicated critics also change the parameterization and training problem in other ways, including representational capability, interference between different constraint signals. As a result, the observed gains may reflect better training setup rather than only the specific mechanism emphasized in the theory.

6. I am confused by the single-constraint Safe-RL experiments in the appendix. In Section 5.2, the key comparison is between one reward critic plus one aggregated cost critic, and one reward critic plus one per-constraint critic. However, in the single-constraint setting, these two architectures appear to coincide, or at least the distinction becomes much less meaningful. If that is the case, then the appendix results do not really test the proposed multi-constraint target-drift mechanism? If not, the paper should clarify more explicitly what architectural difference remains in the single-constraint setup.

Minor comments:
1. Some typos (e.g., space after the comma in line 189, Page 4; “sfollows” in line 190 on Page 4; “We work” in line 197, Page 4) should be addressed during the review process.

Recommendation:

The paper raises a very interesting question, but I do not think the current version of the manuscript convincingly answers it. The theory is developed in an idealized linear stochastic approximation setting under which the mixed-critic bias vanishes asymptotically, so the main result does not establish a persistent theoretical advantage for dedicated critics. The experiments then move to a deep reinforcement learning regime not covered by the analysis, but do not isolate that target-drift mechanism from other explanations such as increased capacity or easier tracking of specific constraints, and they do not clearly show convergence to feasible policies either. As such, I currently lean reject.

---

> ### Author Rebuttal · Authors · 2026-03-31
>
> ***Clarify that the theory mainly motivates finite-timescale settings (Appendices O and P)***
>
> **Response:** We will point readers to Appendices O and P immediately after Theorem 4.7. Our focus is the practical finite-horizon regime, where safety violation during training matters. Accordingly, Theorem 4.7 is not intended to claim a persistent asymptotic separation under Assumption 4.1, but to explain that the mixed critic introduces an additional dual-induced drift term unless the critic is nearly exact or tracks much faster than the dual in infinite time. Appendix O clarifies the asymptotic-coincidence phenomenon under Robbins–Monro conditions, while Appendix P shows that the idealized asymptotic regime can differ substantially from practical finite-horizon deep RL.
>
> ***Question regarding linear function approximation assumptions***
>
> **Response:** We’d like to clarify that **our submitted manuscript provides both theoretical and empirical analysis in linear and nonlinear settings**.
>
> We use a linear critic class in the theory because it is the canonical setting for TD-style convergence analysis and allows closed-form characterisation of target drift and actor-gradient bias. Importantly, **the submitted manuscript also extends the analysis to the nonlinear setting in Appendix G**, which considers a general smooth population loss with a locally convex minimum and shows that dual-induced drift arises generically: for any smooth mixed-critic loss $L_{\text{mix}}(\omega;\theta,\lambda)$ whose targets depend on the scalarized signal $r_\lambda = r - \sum_i \lambda_i c_i$, the population optimum $\omega^* $ typically satisfies $\partial \omega^*/\partial \lambda \neq 0$, so changes in $\lambda$ inevitably alter the critic’s target.
>
> Empirically, our paper covers both theory-aligned and practical deep-RL settings: Appendix O provides an idealized linear setting, while the main PV and EVCS experiments under nonlinear setting with PPO.
>
> ***Question regarding stronger timescale separation.***
>
> **Response:** Heuristic timescale separation is not equivalent to a principled timescale-separation assumption, especially in deep RL. Appendix P shows that even under Robbins–Monro schedules designed to approximate such separation, the mixed critic still performs poorly relative to the dedicated critic. In practical deep RL, training is finite and critic tracking is governed by optimisation dynamics, feature drift, and noise, not by $\eta_t$ alone, while in the mixed-critic setting the target is non-stationary due to both policy and dual updates. Thus, better tuning may alleviate the issue, but does not eliminate it, as supported by our main results and Appendix P. We will clarify this in the revision.
>
> ***Question regarding feasibility***
>
> **Response:** We clarify our claim is that dedicated critics reduce constraint violations, variance, and improve safety during training, rather than guaranteeing exact feasibility. The submitted manuscript already includes feasibility-oriented evidence in Figs. 28–35, where the black dotted lines denote the thresholds d. The $J_c-d$ Pareto views show that dedicated-critic solutions cluster much closer to, and often slightly below, the constraint thresholds than mixed-critic solutions.
> To make this more explicit, we additionally evaluate the total feasibility gap $\sum_i [J_{c_i}-d_i]_+$ and a near-feasible rate in the CMDP bandit. The dedicated critic significantly improves feasibility: the mean total gap drops from 0.4592 ± 0.0825 to 0.1255 ± 0.0793, and the near-feasible fraction (0.05) increases from 0.1700 to 0.5500. These results show that the dedicated critic is not merely “less violating,” but moves policies materially closer to the feasible region. We will revise the paper to clarify this distinction between violation reduction and feasibility.
>
> ***Questions regarding empirical gain and better training setup.***
>
> **Response:** In realistic finite-training settings, we proved that the mixed critic incurs additional bias from dual-induced target drift, while the dedicated critic removes this source of error. Although stronger timescale separation may reduce the gap in idealized asymptotic regimes (Appendices O and P), it does not remove the issue in deep RL setting. For longer training, we extend the run in Figure 3 (power-system) and Figure 36 (EVCS)  to 5× training length, and the results remain consistent. (https://anonymous.4open.science/r/RebuttalPage_Why_Dedicated_Critic-5FE0).
>
> ***Question regarding single-constraint Safe-RL in the appendix***
>
> **Response:** Appendix N is not intended as the main mechanism-isolation experiment for our multi-constraint theory. As stated in the main text, it’s included only as a supplementary special-case evaluation to test whether the dedicated-vs.-mixed critic design still offers practical benefits when reward and constraint are treated as two performance metrics. If the reviewer finds these results distracting, we are happy to remove them.

---

> > ### Author Rebuttal · Reviewer_3F7m · 2026-04-03
> >
> > Thank you for the detailed rebuttal. The clarification regarding the intended interpretation of the theory is helpful, especially the decision to move the discussion of the practical finite-horizon regime and the reference to Appendices O and P immediately after Theorem 4.7. I also appreciate the clarification that Appendix N is intended only as a supplementary special case.
> >
> > However, my main empirical concerns are only partially addressed. The rebuttal improves the framing of the theoretical contribution, but I still find the connection between that analysis and the PPO experiments somewhat indirect. Moreover, I do not think the rebuttal fully resolves the possibility that part of the empirical gain of dedicated critics may come from more general training, constraint scaling, or parameterization effects, rather than only from the specific dual-induced target-drift mechanism emphasized in the paper.

---

> > > ### Author Response · Authors · 2026-04-06
> > >
> > > Thank you for this thoughtful comment. We sincerely appreciate the time and effort you have invested in helping us improve the paper.
> > >
> > > In the nonlinear setting, the theoretical analyses show that the same structural distinction analyzed in the linear setting persists when the signal estimators are implemented via neural networks (e.g., PPO’s case). To more directly address your concern regarding the theory and empirical gains, we add five further ablation studies and diagnostics (covering aspects from general training, constraint scaling, and parameterization efforts; see figures at https://anonymous.4open.science/r/RebuttalPage_Why_Dedicated_Critic-5FE0 for detailed results), together with two existing studies, making a total of seven analyses:
> > > - 1. In our submitted manuscript, both mixed and dedicated methods have used the same PPO backbone and the same MLP design. The actor architecture is identical across methods, and each critic uses the same two-layer MLP architecture. The dedicated-critic only replaces the single mixed critic with multiple parallel critics of the same architecture (Table 1 in the submitted manuscript).
> > > - 2. We run a **parameter-matched comparison**, where the total number of critic parameters is kept the same between the dedicated and mixed/shared setups, while keeping the same network depth. The performance gap remains, indicating that the advantage is not simply due to increased model capacity (Figure 3 in the link).
> > > - 3. We test a **shared-trunk multi-head ablation** (one head per constraint). Its performance is substantially better than the original mixed critic but still not as good as the fully dedicated critic, as we have discussed briefly in the limitation, a shared-backbone multi-head critic may still suffer from cross-signal interference even when each head is per-signal. (Figure 4 in the link).
> > > - 4. We test a **stronger constraint-scaling baseline** using penalty ×10, which also provides only partial improvement: some constraints become better, but others remain poor. The overall performance again lies between the original mixed critic and the fully dedicated critic. This indicates that simple reweighting does not remove the underlying issue that the critic is still fitting a single aggregated signal (Figure 5 in the link).
> > > - 5. We already included a **dual-timescale intervention** in Appendix P. When the dual updates are slowed or controlled, the mixed-dedicated gap is reduced, which is consistent with the proposed dual-induced target-drift mechanism (Appendix P).
> > > - 6. We also define a **proxy gradient alignment metric** to quantify whether the actor update produced by the current method is directionally consistent with a higher-fidelity reference Lagrangian gradient. At a checkpoint $t$, let $g_t^{\mathrm{train}}$ denote the actual actor gradient used by the algorithm, and let $g_t^{\mathrm{ref}}$ denote a high-sample reference gradient estimated under the frozen pair $(\theta_t,\lambda_t)$ using additional on-policy rollouts. The proxy alignment is then defined as $\mathrm{Align}^{\mathrm{proxy}}_t = \frac{\langle g_t^{\mathrm{train}}, g_t^{\mathrm{ref}}\rangle} {\|g_t^{\mathrm{train}}\|_2\,\|g_t^{\mathrm{ref}}\|_2+\varepsilon}$. This metric serves as a local diagnostic of actor-update fidelity: if dedicated critics indeed reduce the actor-gradient bias predicted by the theory, then they should exhibit higher alignment with the reference Lagrangian direction. The results show that our dedicated-critic design achieves substantially better gradient alignment than the mixed-critic design (Figure 6 in the link).
> > > - 7. We introduce **dual oscillation magnitude**, which shows that under a mixed-critic design, updates of the dual variables $\lambda$ change the critic target itself and thereby introduce additional instability into actor learning. A simple proxy is to track the volatility of the dual variables during training. In analogy to the bandit analysis, one may also report a sliding-window oscillation magnitude, such as $\mathrm{Osc}_t = \sqrt{ \max\!\left( 0,\; \mathbb{E}_t[\Delta^2] - \bigl(\mathbb{E}_t[\Delta]\bigr)^2 \right)}$, where $\Delta$ denotes a scalar summary of dual variation within the window. The results show that the dedicated critic is much more stable in terms of oscillation than the mixed critic (Figure 7 in the link).
> > >
> > > Overall, these additional controls systematically rule out alternative explanations such as model capacity, architecture, or simple reweighting. We also explicitly show the alignment with gradients. These results consistently support the advantage of dedicated-critic design performing the best. This is consistent with our theoretical finding that the central issue is the structural coupling in the mixed critic, especially through the dual-induced target-drift mechanism, rather than from superficial design differences.

---

### Official Review · Reviewer_g8dr · 2026-03-13

**Soundness:** 3
**Presentation:** 3
**Significance:** 3
**Originality:** 3
**Overall Recommendation:** 5
**Confidence:** 3

**Summary:**

This paper investigates critic architecture design in multi-constraint safe reinforcement learning, specifically comparing mixed critics that aggregate constraints versus dedicated critics with per-constraint estimators. The authors provide theoretical analysis demonstrating that mixed critics introduce an additional bias term proportional to dual variable updates, causing target drift during training. In contrast, dedicated critics isolate constraint signals and avoid this λ-induced drift. Experiments on a simple bandit problem, complex power system management, and electric vehicle charging coordination validate the theoretical claims, showing that dedicated critics achieve lower constraint violations, more stable training, and better Pareto frontier trade-offs compared to mixed critic baselines like PPO-Lagrangian.

**Compliance With Llm Reviewing Policy:**

Affirmed.

**Final Justification:**

I will keep my rating.

**Key Questions For Authors:**

Refer to Weaknesses

**Limitations:**

Requires separate critics for each constraint which increases memory footprint and computational cost linearly with the number of constraints

Static one-to-one mapping between constraints and critics may be inefficient when constraints have varying relevance or activate sparsely during training

Provides asymptotic error bounds rather than finite-time guarantees for deep RL settings with neural networks and practical stabilisation techniques

Fairness constraint performance shows higher variance across runs in some experiments, suggesting room for additional stabilisation mechanisms

**Strengths And Weaknesses:**

Strengths:

Provides rigorous theoretical analysis with clear mathematical derivations showing the fundamental difference between mixed and dedicated critic architectures, including explicit error bounds and bias characterizations

Validates theory through multiple complementary experiments ranging from a minimal diagnostic bandit to complex real-world inspired power system applications with heterogeneous constraints

Demonstrates consistent empirical advantages across multiple metrics including constraint satisfaction, training stability, gradient alignment, and Pareto optimality

Addresses an important overlooked design choice in safe RL literature and provides actionable guidance for practitioners

Weaknesses:

The improved constraint satisfaction comes at a significant cost to reward performance (87% reduction in the power system case and 109% improvement negative in EVCS case), raising questions about practical applicability when economic objectives matter

Theoretical results rely on linear function approximation assumptions while experiments use neural networks, leaving a gap between theory and practice

Increased computational and memory requirements with dedicated critics may limit scalability to problems with very large numbers of constraints

---

> ### Author Rebuttal · Authors · 2026-03-31
>
> ***Question regarding practical applicability when economic objectives matter.***
>
> **Response:** We thank the reviewer for highlighting this. We agree that economic performance is important in many applications, and that there is often a natural trade-off between safety and economic return. However, motivated by practical needs of real-world applications (e.g., power-grid control and EV charging), constraints represent safety requirements that must be carefully respected, rather than optional performance objectives. In safety-critical settings, policies are expected to maintain violations at low levels. As a result, a policy with lower reward but consistently better constraint satisfaction may be preferable to those attaining higher reward by violating constraints more often.
> Our experiments also demonstrate this trade-off: the first row of Figures 20–27 illustrates the Pareto frontier between reward and constraint satisfaction. When the safety threshold is relatively lenient, our method achieves comparable, and in some cases even better, reward than the mixed critic, while maintaining substantially lower constraint violation. When the threshold becomes tighter, the reward naturally decreases. This is due to the nature of the underlying problem: under stricter safety requirements, the feasible policy space becomes smaller, so achieving high reward requires sacrificing less on safety, and vice versa. In this regime, lower reward of our method should be interpreted as the result of adhering more closely to the desired constraint levels, rather than inferior optimisation. We will revise the manuscript to clarify this point more. In practice, the reward–safety balance can be further adjusted through the choice of constraint thresholds and cost design, depending on the application.
>
> ***Question regarding linear function approximation assumptions***
>
> **Response:** We’d like to further clarify the use of a linear critic class in the theoretical development. Because linear critic is the canonical setting for TD-style convergence analysis and yields closed-form normal equations, allowing us to quantify target drift and actor-gradient bias explicitly.
>
> Importantly, our conclusions are not limited to linear settings. In Appendix G of the submitted manuscript, we explicitly present the analysis in non-linear settings. Appendix G treats a general smooth population loss with a locally convex minimum and shows that dual-induced drift also arises generically. Using an implicit-function argument, we show that for any smooth mixed-critic loss $L_{\text{mix}}(\omega;\theta,\lambda)$ whose targets depend on the scalarized signal $r_\lambda = r - \sum_i \lambda_i c_i$, the population optimum $\omega^* $ typically satisfies $\partial \omega^*/\partial \lambda \neq 0$, implying that changes in $\lambda$ inevitably alter the critic’s target.
> Empirically, this paper also includes both theory-aligned and deep-RL settings. In Appendix O, we provide an idealised Robbins–Monro experiment specifically designed to approximate the regime assumed in the theory, as a near-linear sanity check and a direct empirical realisation of the asymptotic linear-theory predictions. In contrast, our main PV and EVCS results use nonlinear deep-RL regime of practical interest.
>
> ***Question regarding Increased computational and memory requirements with dedicated critics***
>
> **Response:** We thank the reviewer for this point and agree that dedicated critics incur additional computational and memory cost, which can become a scalability concern for very large numbers of constraints. This issue is not only acknowledged qualitatively; we also include a constraint-count ablation in Appendix H of the submitted manuscript. Specifically, we compare runs with $m \in {1,2,3,5}$ constraint critics, all sharing a common backbone architecture for the PV experiment. The per-update cost scales roughly linearly in $m$, but the observed overhead is modest in the regime we target, with wall-clock time increasing by only 71% (five critic vs. one critic).
>
> To make this clearer, we additionally report a plot of computation time per training epoch as a function of the number of Lagrangian critics. The results follow a clean linear trend, captured by the fitted regression y=0.00169 x+0.00776, indicating stable and predictable scaling as $m$ increases. The variance bars are small across all settings, suggesting that training remains relatively stable even with more critics. We therefore agree that scalability is a limitation in large-constraint regimes, but the empirical evidence suggests the overhead is controlled rather than prohibitive over the tested range. We will discuss this issue more explicitly in the limitations section.
>
> ***Question regarding limitation discussion.***
>
> **Response:** Thank you very much for raising these valuable points. We agree that these limitations are worth discussing more explicitly, and we appreciate the reviewer’s constructive perspective.

---

> > ### Author Rebuttal · Reviewer_g8dr · 2026-04-03
> >
> > The authors have solved my problems and I remained positive.

---

> > > ### Author Response · Authors · 2026-04-06
> > >
> > > Thank you for your follow-up and for carefully reviewing our responses. We are very pleased to hear that our reply has addressed all of your concerns. We sincerely appreciate your consideration and continued support for the paper.

---

### Decision · Program_Chairs · 2026-04-30

**Decision:**

Accept (regular)

**Comment:**

The reviewers found that this paper makes a clear contribution to the field of constrained RL. In particular, the importance of having separate critics when handling multi-constraint problems is an issue sometimes overlooked by safe RL practitioners. The paper provides a theoretical analysis and an empirical evaluation of these issues, augmented by a clear demonstration through a minimalist example. Although the empirical evaluation departs from the usual presentation, the rebuttal showed that the supplemental material presents the results in terms of constraint violations, including the constraint bounds. Finally, the rebuttal was sufficient to address the issues raised in the reviews regarding the paper's theory. Therefore, I recommend accepting the paper.